# DNMT3A clonal hematopoiesis-driver mutations induce cardiac fibrosis by paracrine activation of fibroblasts

Mariana Shumliakivska[1,2,3], Guillermo Luxán [1,2,3], Inga Hemmerling[4,5], Marina Scheller [6], Xue Li[4,5], Carsten Müller-Tidow [6], Bianca Schuhmacher[1], Zhengwu Sun[7], Andreas Dendorfer [7], Alisa Debes[1], Simone-Franziska Glaser [1,2,3], Marion Muhly-Reinholz[1], Klara Kirschbaum [8], Jedrzej Hoffmann [2,9], Eike Nagel[2,9], Valentina O. Puntmann[2,9], Sebastian Cremer[1,2,3,8], Florian Leuschner[4,5], Wesley Tyler Abplanalp [1,2,3], David John [1,3], Andreas M. Zeiher [1,2,3,10] & Stefanie Dimmeler [1,2,3,10] ✉

Hematopoietic mutations in epigenetic regulators like DNA methyltransferase 3 alpha (DNMT3A), play a pivotal role in driving clonal hematopoiesis of indeterminate potential (CHIP), and are associated with unfavorable outcomes in patients suffering from heart failure (HF). However, the precise interactions between CHIP-mutated cells and other cardiac cell types remain unknown. Here, we identify fibroblasts as potential partners in interactions with CHIP-mutated monocytes. We used combined transcriptomic data derived from peripheral blood mononuclear cells of HF patients, both with and without CHIP, and cardiac tissue. We demonstrate that inactivation of DNMT3A in macrophages intensifies interactions with cardiac fibroblasts and increases cardiac fibrosis. DNMT3A inactivation amplifies the release of heparin-binding epidermal growth factor-like growth factor, thereby facilitating activation of cardiac fibroblasts. These findings identify a potential pathway of DNMT3A CHIP-driver mutations to the initiation and progression of HF and may also provide a compelling basis for the development of innovative anti-fibrotic strategies.

Clonal hematopoiesis of indeterminate potential, or CHIP, is a common age-related condition, in which acquired mutations in hematopoietic stem cells result in the expansion of a genetically distinct subpopulation of blood cells[1]. The incidence of clonal hematopoiesis increases with age and was previously shown to associate with a variety of cardiovascular diseases, including heart failure[2–7]. Mutations in the epigenetic modifiers DNA methyltransferase 3 A (*DNMT3A*) and ten eleven translocation 2 (*TET2*) are the most common CHIP-driver mutations[2]. Importantly, previous studies have shown that individuals harboring either DNMT3A or TET2 CHIP-driver mutations in circulating blood cells experience increased mortality in the presence of heart failure or aortic valve stenosis[8–12].

Mechanistically, both experimental as well as clinical single-cell RNA sequencing studies have revealed that DNMT3A and TET2 CHIP-driver mutations influence the inflammatory potential of circulating immune cells, potentially contributing to the promotion of diffuse cardiac fibrosis and consequently impacting long-term outcome[4,12–16]. Several studies have shown that the heart is continuously replenished with monocyte-derived macrophages expressing proinflammatory mediators, which play a role in adverse left ventricular remodeling,

heart failure pathogenesis, and progression[17–19]. Notably, experimental models simulating DNMT3A-mediated clonal hematopoiesis have shown enhanced infiltration of cardiac tissue with macrophages after heart failure induction, alongside increased gene expression of markers of activated monocytes and T cells, indicating heightened myocardial inflammation[14].

Nevertheless, a fundamental question remains unresolved: how can such a minor fraction of circulating blood cells defined as CHIP with a variant allele frequency of at least 2% resulting in at least 4% of circulating myeloid cells harboring the mutation[1] activate inflammatory processes in cardiac tissue and influence the prognosis of patients harboring these mutations. We hypothesized that circulating mutant cells, which are continuously recruited to the heart, may interact with cardiac cells to promote heart failure.

To test this hypothesis, we utilized in silico tools capable of analyzing intercellular communications by predicting ligand-receptor interactions from single-cell or single-nuclei RNA-sequencing data (scRNA-seq or snRNA-seq). Several methods have been recently developed to infer cell–cell communication from scRNA-seq, such as CellPhoneDB[20] or CellChat[21], and have been successfully applied to the analysis of cardiac tissue[22]. In this study, we used CellChat to predict interactions between monocytes deriving from patients carrying DNMT3A CHIP mutations and cells of the human cardiac tissue. The analysis revealed an enriched potential for interaction between monocytes from DNMT3A CHIP carriers and cardiac fibroblasts. These predictions were subsequently validated, demonstrating that DNMT3A silencing augments cardiac fibroblast activation via the heparin-binding epidermal growth factor-like growth factor (HB-EGF) - epidermal growth factor (EGFR) axis. In murine models and in humans carrying DNMT3A CHIP mutations, we observed increased cardiac fibrosis, providing evidence that the interaction between circulating mutant cells, which are recruited to the heart, and cardiac fibroblasts plays a prominent role in exacerbating diffuse cardiac fibrosis. This process may promote disease progression and result in higher mortality rates among patients with established heart failure.

## Results

### Interaction of DNMT3A CHIP monocytes with cardiac fibroblasts

To investigate potential interactions of monocytes with cardiac cell types, we integrated single-cell RNA-sequencing data from circulating monocytes derived from patients with chronic heart failure with reduced ejection fraction (HFrEF), who either carried DNMT3A CHIP-driver mutations ($n = 5$) or were No-CHIP carriers ($n = 4$)[15], with publicly available single-nuclei RNA-sequencing data of human healthy heart tissue ($n = 14$)[23] and from heart failure with reduced ejection fraction (HFrEF) patients ($n = 3$) (Fig. 1a, b and Supplementary Table 1). The integration was performed using the IntegrateData function within the Seurat package, as described in the method section. Quality controls for the integrated object are provided in Supplementary Fig. 1. The number of expressed genes was higher in circulating monocytes, which were obtained from scRNA-seq, in comparison to the snRNA-seq data from cardiac tissue. However, DNMT3A CHIP and No-CHIP carriers showed similar features (Supplementary Fig. 2a–c). This similarity allowed for a meaningful comparative analysis between these two groups of circulating monocytes and the cells of the cardiac tissue.

Unsupervised clustering of the combined data sets revealed 21 distinct clusters, which were annotated to the major cell types of the heart (Fig. 1c, Supplementary Fig. 2a). Briefly, cardiomyocytes expressed genes encoding troponin (TNNT2), ryanodine receptor type 2 (RYR2), endothelial cells expressed genes encoding vascular endothelial (VE)-cadherin (CDH5) and CD31 (PECAM1), fibroblasts were characterized by decorin (DCN) expression, pericytes expressed platelet-derived growth factor receptor beta (PDGFRB), and smooth muscle cells were identified by expression of myosin heavy chain 11 (MYH11) (Fig. 1d, Supplementary Fig. 2b, c). Immune cells of the cardiac

tissue (named "Immune cells") were characterized by CD163, which is primarily expressed by tissue-resident macrophages, whereas the cluster of circulating monocytes (named "Monocytes") showed higher expression of the monocyte markers CD14 and PTPRC, as well as CCR2, which mediates homing to the cardiac tissue[24] (Supplementary Fig. 2d).

Pooled analysis of cellular interactions by CellChat[21] demonstrated that monocytes derived from patients carrying DNMT3A CHIP-driver mutations showed the highest numbers of outgoing signals between the monocyte cluster to healthy tissue-derived cardiac fibroblasts (Fig. 1e). Additionally, a higher number of interactions was predicted for monocytes derived from DNMT3A CHIP carriers compared to No-CHIP carriers to most cell types (Fig. 1e). When comparing the outgoing monocyte interactions with the cardiac cells of the HFrEF tissue, we found a generally lower absolute number of interactions, which is explained by the lower number of patients in the HFrEF compared to healthy control samples (Fig. 1e, f). However, the patterns of interactions of DNMT3A CHIP versus No-CHIP with the HFrEF tissue-derived cells were similar (Fig. 1f). Again, DNMT3A CHIP-derived monocytes showed a higher number of predicted interactions compared to No-CHIP carriers (Fig. 1f). Next, we analyzed the specifically outgoing interactions, which were enriched in DNMT3A CHIP compared to No-CHIP-derived monocytes to cardiac cell types. The highest number of DNMT3A CHIP monocyte-enriched interactions was observed between the monocytes and fibroblasts of the HFrEF samples (Fig. 1g). These results predict that DNMT3A CHIP monocytes, after being recruited to the heart, may interact predominantly with cardiac fibroblasts.

To test this hypothesis, we silenced DNMT3A in PMA-activated THP-1 monocytes by siRNAs, which efficiently reduced DNMT3A mRNA and protein expression compared to control siRNAs (Fig. 2a, b). The silencing of DNMT3A with siRNA mimics the most common loss-of-function CHIP mutations[25]. Subsequently, we added the supernatants from these transfected monocytes to cardiac fibroblasts, assessing their impact on fibroblast gene expression and myofibroblast activation (Fig. 2c). Remarkably, treatment of cardiac fibroblasts with supernatants from DNMT3A-silenced monocytes led to an increase in α-smooth muscle actin (αSMA) expression, a well-known marker of myofibroblast activation[26,27], at both mRNA and protein level (Fig. 2d, e). Moreover, the supernatants of siRNA DNMT3A-treated monocytes increased the expression of TGF-β1 in fibroblasts (Fig. 2f), which is the prototypical fibrogenic factor in cardiac tissue[28,29]. In addition, this treatment stimulated contraction and migration of cardiac fibroblasts (Fig. 2g, h), recognized features of myofibroblast activation[30,31]. The conditioned medium from DNMT3A-silenced monocytes also augmented collagen expression (Supplementary Fig. 3a), and induced a trend towards enhanced proliferative capacity in treated fibroblasts (Supplementary Fig. 3b).

To gain further insights into how monocytes derived from patients harboring DNMT3A CHIP-driver mutations affect fibroblasts and other cardiac cells in a more physiological environment, we used 3D cardiac tissue mimetics (cardiospheres)[32], which comprise cardiomyocytes, fibroblasts and endothelial cells (Fig. 3a). Functionally, supernatants derived from DNMT3A-silenced monocytes reduced the frequency of cardiosphere contractions (Fig. 3b, c), while the size of cardiospheres remained unchanged (Fig. 3b, d). This suggests that DNMT3A silencing does not exert a direct effect on cardiomyocyte hypertrophy in this specific context. Moreover, the area positive for PDGFRα was significantly increased (Fig. 3e), while vascularization remained unaffected (Supplementary Fig. 3c). Consistent with the results of the 2D culture, supernatants from DNMT3A-silenced monocytes also increased αSMA expression (Fig. 3f). Given the observed effect on contractility by supernatants from DNMT3A silenced monocytes, and considering the predicted interaction between CHIP monocytes and cardiomyocytes, we conducted additional tests on cardiomyocytes using a recently established human

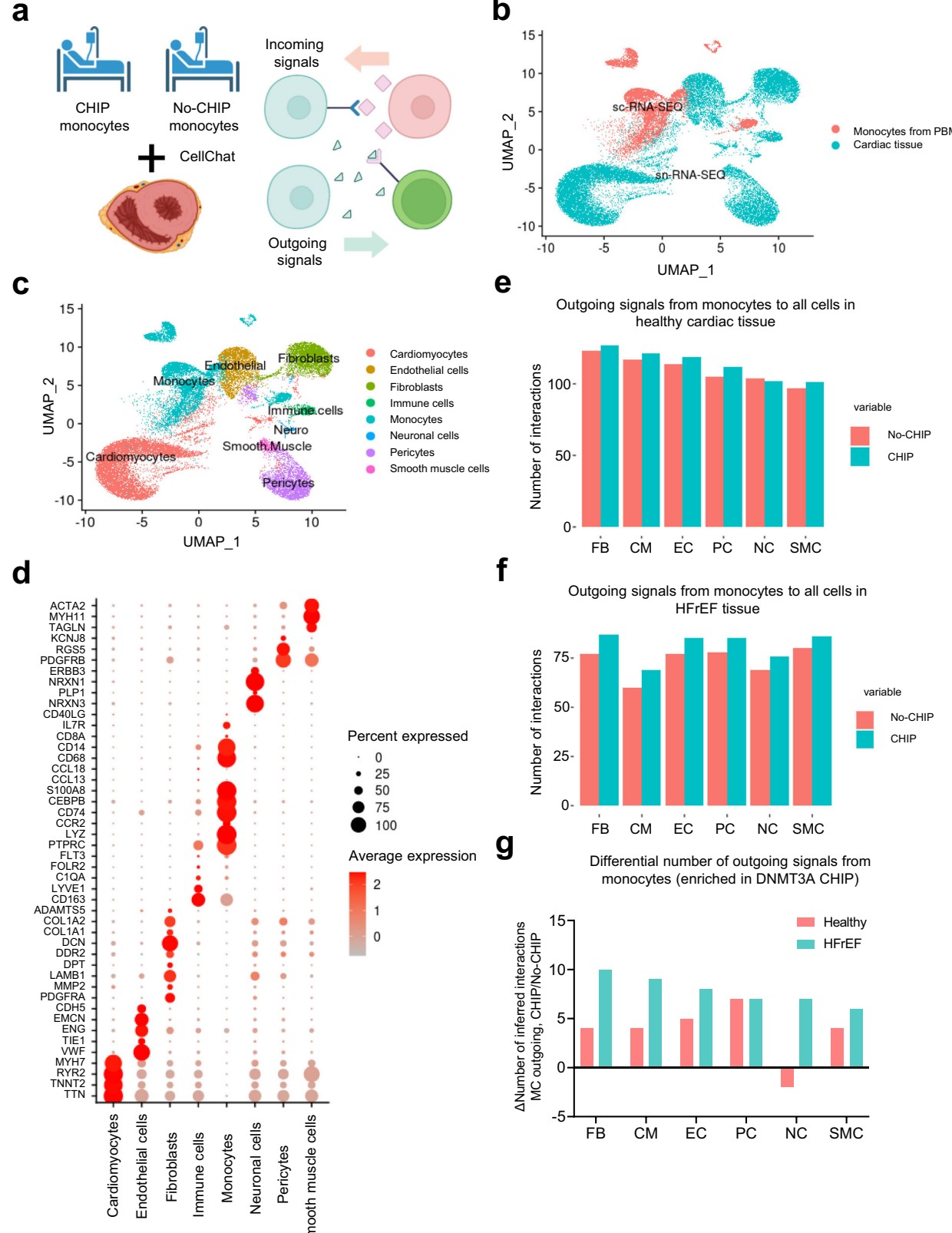

ventricular cardiomyocyte cell line[33]. However, we did not observe cytotoxic effects in this context (Supplementary Fig. 3d).

Taken together, these data demonstrate that the reduction of DNMT3A in monocytes can induce cardiac fibroblast activation and reduce contraction frequency of cardiac tissue mimetics in a paracrine manner.

## DNMT3A CHIP driver mutations promote diffuse cardiac fibrosis in mice

To investigate whether DNMT3A CHIP-driver mutations in hematopoietic cells stimulate cardiac fibroblasts in vivo, we utilized a humanized *DNMT3A-R882H* knock-in mouse model that expresses hDNMT3A[R882H] in a Cre-dependent manner[34]. Bone marrow cells

**Fig. 1 | DNMT3A CHIP monocytes interact with cardiac fibroblasts in healthy and diseased heart. a** Schematic representation of the experimental design. Bioinformatic analysis combining monocytes derived from scRNA-seq dataset of PBMC from DNMT3A CHIP and No-CHIP patients, snRNA-seq dataset of cardiac tissue from the septum of control hearts ($n = 14$ biologically independent samples) and snRNA-seq dataset of cardiac tissue from patients with heart failure with reduced ejection fraction (HFrEF) (n = 3 biologically independent samples) **b** Uniform manifold approximation and projection (UMAP) plots after integration of monocytes from PBMC and cardiac data sets. The source of data in the integrated object is color coded. **c** Representation of the different cell type clusters identified after integration **d** Dot plot of representative marker gene expression in each cell type cluster **e** Analysis of the integrated objects for cellular interactions by CellChat. Total number of outgoing paracrine signals from monocytes to all cells of the healthy heart is shown. **f** Analysis of the integrated objects for cellular interactions by CellChat. Total number of outgoing paracrine signals from monocytes to all cells of the HFrEF tissue. **g** Differential number of outgoing signals enriched in DNMT3A CHIP-carrier monocytes to the individual clusters of cardiac cells of healthy and HFrEF tissues as indicated. FB Fibroblasts. CM Cardiomyocytes. EC Endothelial cells. PC Pericytes. NC Neurons. SMC Smooth muscle cells. Source data are provided as a Source Data file.

---

from pI:pC treated donor mice with monoallelic *DNMT3A-R882H* expression (*Mx1-Cre+:DNMT3A WT/R882H*) and control wild-type donor (*Mx1-Cre−:DNMT3A WT/WT*) (both Ly5.2+) mice were transplanted into congenic wildtype recipients (Ly5.1+). Myocardial infarction was induced by permanent ligation of the left anterior descending artery (LAD)[35], and hearts were analyzed four weeks after surgery (Fig. 4a). Comparable cardiac troponin T levels 24 h post-LAD ligation indicated similar infarct sizes in mice carrying a human DNMT3A$^{R882H}$ mutation (DNMT3A$^{R882H}$) in bone marrow-derived hematopoietic cells and the control group (WT) (Fig. 4b). Histopathological analysis of cardiac tissue after four weeks showed an increase in cardiac interstitial fibrosis in the remote zone of recipients, expressing *DNMT3A$^{R882H}$* cells in comparison to control mice, with no change in infarct size (Fig. 4c–f). Cardiac fibrosis was also increased by 1.42-fold in *DNMT3A$^{R882H}$*-carrying mice at baseline, but this increase was not statistically significant (Supplementary Fig. 4a). Furthermore, mutant *DNMT3A$^{R882H}$*-carrying mice exhibited an increase in CD68-positive inflammatory cells in the remote zone after infarction, while no change was detected in non-injured mice at baseline (Supplementary Fig. 4b, c). These data suggest that mimicking *DNMT3A$^{R882H}$* in mice augments the presence of inflammatory cells in the remote zone and induces diffuse interstitial cardiac fibrosis, particularly in response to injury.

To determine the effect of the *DNMT3A$^{R882H}$* mutation on the cardiac tissue on a cellular level, we performed single-nuclei RNA-sequencing of cardiac tissue. After annotating all major cardiac cell types (Fig. 4g, Supplementary Fig. 5a–d), we analyzed the expression of fibrosis-related genes in cardiac fibroblasts (Fig. 4h). Expression of pro-fibrotic and myofibroblast activation genes *Col3a1, Postn, Pdgfra* and others were upregulated in fibroblasts (Fig. 4h, i, Supplementary Fig. 5e). Additionally, we did observe an elevated expression of *Ccr2* and increased numbers of *Ccr2* positive cells in the macrophages subclusters specifically in *DNMT3A$^{R882H}$* mice (Fig. 4j, Supplementary Fig. 5f–h), suggesting an enrichment of bone marrow-derived macrophages.

## DNMT3A CHIP driver mutations promote diffuse cardiac fibrosis in heart failure patients

To further investigate the impact of harboring DNMT3A CHIP-driver mutations on diffuse cardiac fibrosis in humans, we determined cardiac fibrosis using cardiac magnetic resonance imaging with myocardial mapping[36–38] in patients with heart failure (Fig. 5a, Supplementary Table 2). 6 of the patients were harboring DNMT3A CHIP-driver mutations (Fig. 5b–d, Supplementary Fig. 6), but did not differ with respect to sex, age or co-morbidities from No-CHIP patients (Supplementary Table 2). We performed native T1 mapping, which specifically determines excessive extracellular matrix deposition associated with diffuse fibrosis[39]. In addition, we measured T2, which primarily detects myocardial edema[36,38]. Carriers of DNMT3A CHIP-driver mutations had significantly increased native T1, but showed no differences in native T2 measurements, indicating the presence of diffuse myocardial fibrosis in these patients (Fig. 5e–g).

## EGF signaling contributes to CHIP monocyte-mediated cardiac fibroblast activation

Next, we assessed the mechanism by which monocytes from DNMT3A CHIP-driver mutation carriers stimulate fibroblast activation. Therefore, we predicted receptor-ligand interactions between monocytes derived from DNMT3A CHIP-driver mutation carriers versus No-CHIP patients, and cardiac tissue by using CellChat. When we specifically analyzed interactions between circulating monocytes and fibroblasts within the healthy heart tissue, we found a significant enrichment of signaling pathways including natriuretic peptide receptor 2 (NPR2), epidermal growth factor (EGF), interleukin 1 (IL1), resistin (RESISTIN), insulin-like growth factor (IGF), and others, in DNMT3A CHIP monocyte interactions (Fig. 6a). In HFrEF hearts, enriched interactions showed some overlaps, but also distinct pathways including prolactin (PRL), epidermal growth factor (EGF), activin (ACTIVIN), resistin (RESISTIN), and insulin-like growth factor (IGF) signaling between DNMT3A CHIP monocytes and cardiac fibroblasts (Fig. 6b). Common enriched pathways for DNMT3A CHIP condition in both healthy and diseased heart tissue include EGF, RESISTIN and IGF, with EGF dominating the absolute information flow (Fig. 6a–c). Given that EGF was predicted to mediate interactions of CHIP monocytes with healthy and disease heart and showed the highest information flow, we focused our attention on this pathway.

EGF signaling was characterized by enriched ligand-receptor pairing of heparin-binding epidermal growth factor (EGF)-like growth factor (HB-EGF) and amphiregulin (AREG) expressed by monocytes of DNMT3A CHIP-carriers with the receptors EGFR, ERRB2 or ERRB4 in cardiac cell types in healthy and heart failure conditions (Fig. 6d). Reactome analysis of interactions specific to monocytes obtained from DNMT3A CHIP-driver mutation carriers to cardiac fibroblasts revealed an enrichment of EGFR interactions with known EGF downstream signaling pathways like phospholipase C-gamma, GRB2, SHC1 activity (Fig. 6e, f). Moreover, analysis of the specific interactions between monocytes and fibroblasts revealed a specific enrichment of EGF-family members to EGFR crosstalks in DNMT3A CHIP carriers (Fig. 6g).

To test the hypothesis that EGFR signaling is involved in the CHIP-driven activation of fibroblasts, we determined the expression of predicted EGF family member ligands in monocytes and the EGF receptors on cardiac fibroblasts at the single-cell level. We found that the expression of both *HBEGF* and *AREG* was significantly higher in circulating monocytes of heart failure patients carrying DNMT3A CHIP-driver mutations compared to No-CHIP carriers (Fig. 6h, Supplementary Fig. 7a). Among the EGF receptor family members, EGFR exhibited the highest expression in cardiac fibroblasts, both in the healthy and HFrEF hearts (Fig. 6i, Supplementary Fig. 7b, c). Moreover, HB-EGF protein levels were elevated in plasma samples obtained from patients with HFrEF harboring DNMT3A CHIP-driver mutations compared to No-CHIP HFrEF patients, as assessed by ELISA (Fig. 6j; Supplementary Table 3). Furthermore, the percentage of *Hbegf* positive cells in the macrophage cluster was increased by 7,87-fold in DNMT3A$^{R882H}$ mice, whereas *Areg* expression was similar between the groups (Supplementary Fig. 7d–g).

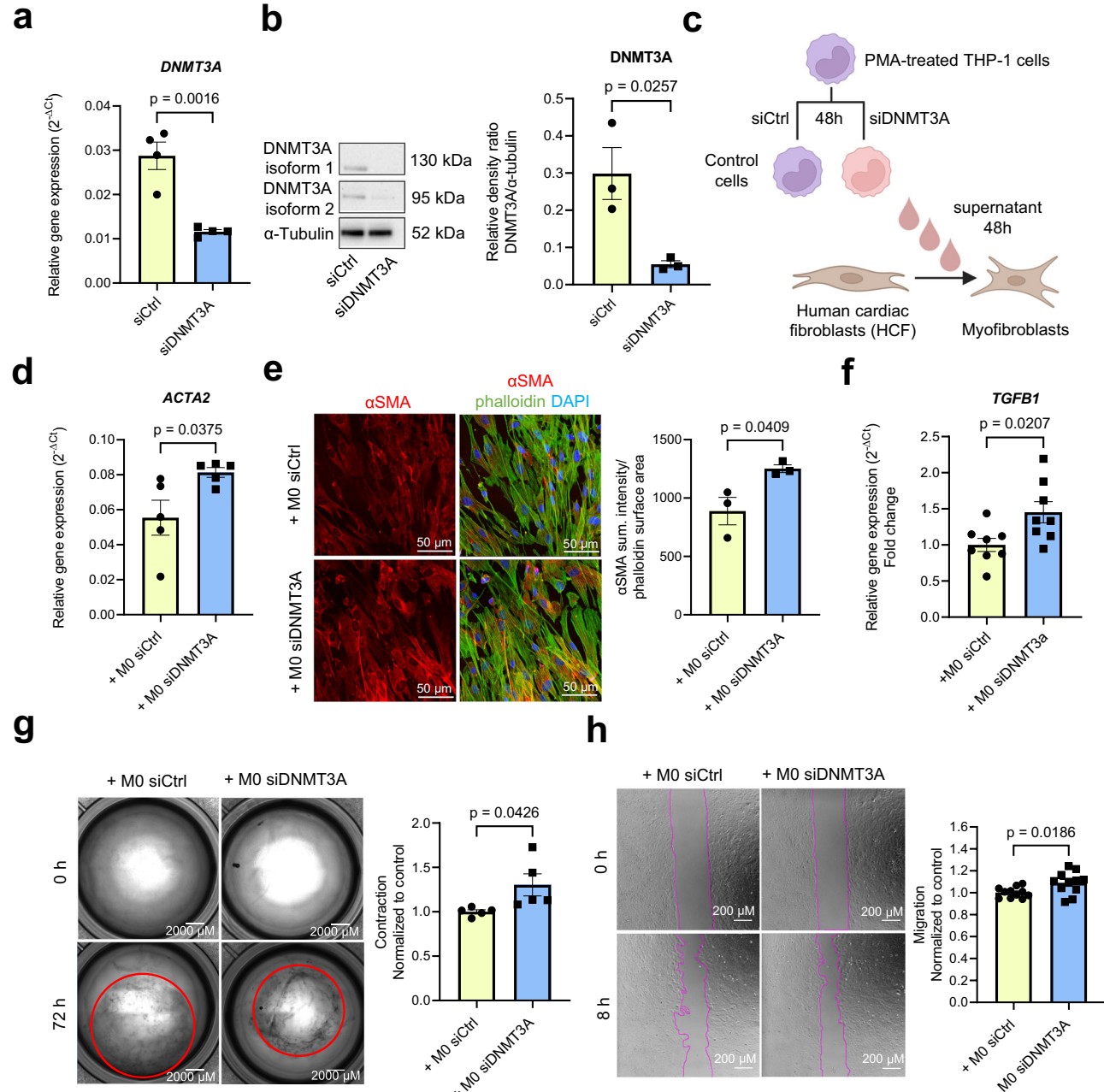

**Fig. 2 | DNMT3A-silenced monocytes activate cardiac fibroblasts in a paracrine manner. a** *DNMT3A* mRNA expression in PMA-activated THP-1 cells after siRNA silencing of *DNMT3A* normalized to *RPLPO* mRNA expression ($n = 4$ biologically independent samples). Source data are provided as a Source Data file. **b** DNMT3A protein expression (isoform 1 (DNMT3A1) and 2 (DNMT3A2)) in PMA-activated THP-1 cells after siRNA silencing of DNMT3A ($n = 3$ biologically independent samples). Representative Western blots are shown on the left. Source data are provided as a Source Data file. **c** Schematic representation of the indirect co-culture experiment. **d** Relative *ACTA2* gene expression normalized to *RPLPO* mRNA expression in human cardiac fibroblasts (HCF) after incubation with supernatants from PMA-activated THP-1 cells for 48 h after siRNA silencing of *DNMT3A* relative or negative control ($n = 5$ biologically independent samples). Source data are provided as a Source Data file. **e** Immunofluorescence analysis of αSMA protein expression in stimulated HCF (n = 3 biologically independent experiments).

Representative examples are shown left. Source data are provided as a Source Data file. **f** Relative *TGFB1* mRNA expression normalized to *RPLPO* mRNA expression analyzed by qPCR in stimulated HCF ($n = 8$ biologically independent samples). Source data are provided as a Source Data file. **g** Collagen gel contraction analysis in stimulated HCF ($n = 5$ biologically independent experiments) after treatment with supernatants. Source data are provided as a Source Data file. **h** Migration assay of stimulated HCF ($n = 11$ biologically independent experiments) after treatment with supernatants. Source data are provided as a Source Data file. Data are shown as mean ± SEM (**a**, **b**, **d**–**h**). Normal distribution was assessed using the Shapiro−Wilk test (**a**, **b**, **d**–**h**). Statistical analysis was performed using unpaired, two-sided Student's *t* tests (**a**,**b**, **d**–**h**). $t = 5.471$, 6 degrees of freedom (**a**); $t = 3.464$, 4 degrees of freedom (**b**); $t = 2.490$, 8 degrees of freedom (**d**); $t = 2.977$, 4 degrees of freedom (**e**); $t = 2.607$, 14 degrees of freedom (**f**); $t = 2.408$, 8 degrees of freedom (**g**); t = 2.562, 20 degrees of freedom (**h**).

Given that EGFR signaling is a well-known mediator of cardiac fibroblast activation[40,41], these data suggest that monocytes obtained from DNMT3A CHIP-driver mutation carriers may release factors of the EGF family, inducing cardiac myofibroblast activation. This is

particularly noteworthy because cardiac myofibroblasts express high levels of *EGFR* in the human heart.

The regulation of HB-EGF expression by DNMT3A was further supported by the detection of increased expression of HB-EGF mRNA

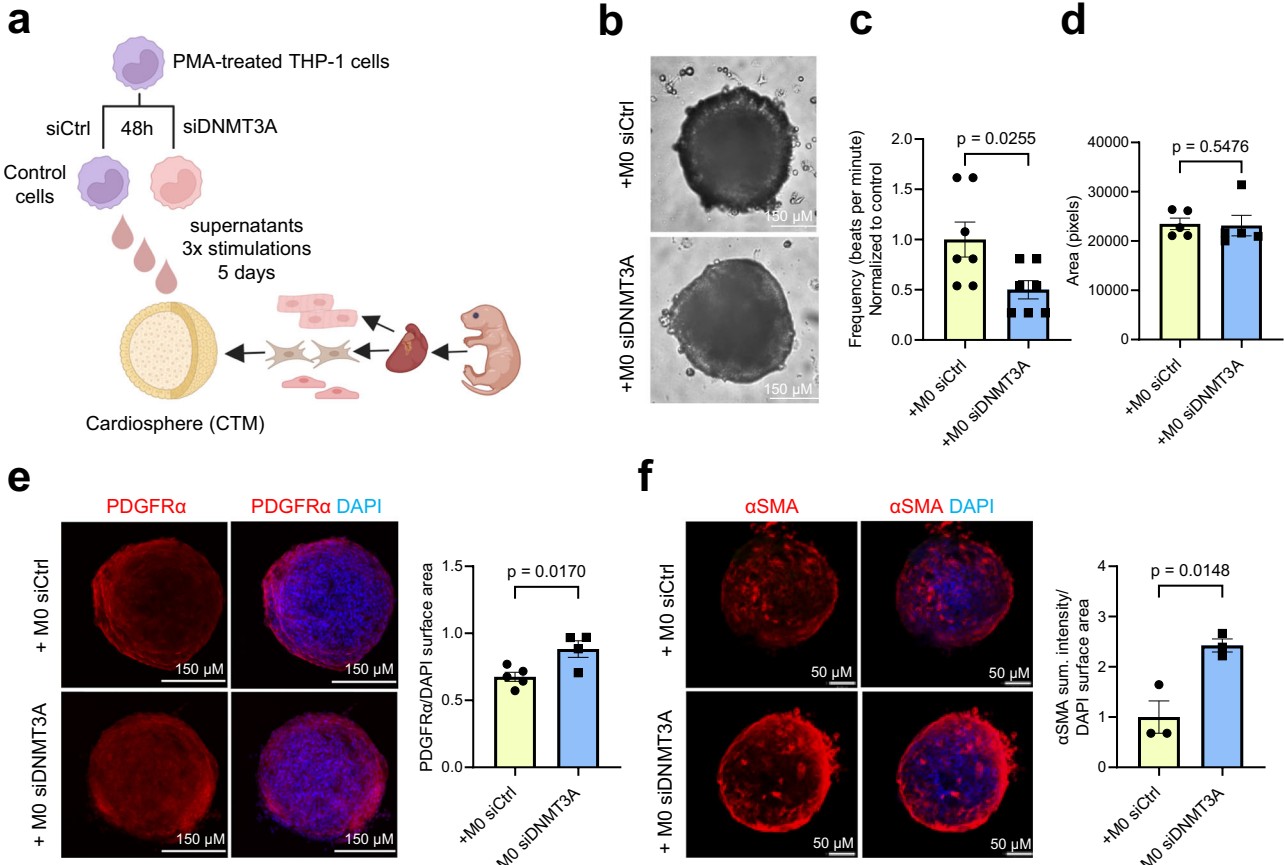

**Fig. 3 | DNMT3A-silenced monocytes reduce contractility and lead to fibrosis in cardiospheres. a** Schematic representation of the experimental design **b** Representative brightfield images of cardiospheres upon indirect co-culture (5 days, 3 stimulations) **c** Beating frequency of cardiospheres stimulated with THP-1 supernatants ($n = 7$ biologically independent samples). Source data are provided as a Source Data file. **d** Cardiospheres area upon stimulation with THP-1 supernatants ($n = 5$ biologically independent samples). Source data are provided as a Source Data file. **e** Immunofluorescence analysis of PDGFRα protein expression in stimulated cardiospheres ($n = 4$–5 biologically independent samples). Source data are provided as a Source Data file. **f** Immunofluorescence analysis of αSMA protein expression in stimulated cardiospheres ($n = 3$ biologically independent samples). Source data are provided as a Source Data file. Data are shown as mean ± SEM. **c**–**f** Normal distribution was assessed using the Shapiro–Wilk test. **c**–**f** Statistical analysis was performed using unpaired, two-sided Student's $t$ tests (**c**, **e**–**f**). $t = 2.550$, 12 degrees of freedom (**c**); $t = 3.464$, 4 degrees of freedom (**e**); $t = 3.113$, 7 degrees of freedom (**e**); $t = 2.977$, 4 degrees of freedom (**f**); $t = 4.105$, 4 degrees of freedom. Statistical analysis was performed using two-tailed Mann–Whitney test (**d**).

and protein levels in DNMT3A-silenced monocytes in vitro (Fig. 7a–c). In addition, we observed increased expression of metalloproteases such as *ADAM8* and *ADAM9*, which are known to shed HB-EGF[42], in monocytes of DNMT3A CHIP carrying patients (Supplementary Fig. 8a, b) and after *DNMT3A* silencing in vitro (Supplementary Fig. 8c). Collectively, these data demonstrate that DNMT3A inactivation leads to an increased expression and possibly activation of HB-EGF in cell culture, in mice and in humans.

To gain insights into the functional consequences of the increased HB-EGF expression, we tested the effects of recombinant HB-EGF on cardiac fibroblast monocultures and cardiospheres containing cardiac fibroblasts. Recombinant HB-EGF increased αSMA expression and proliferation of cardiac fibroblasts (Fig. 7d, e). HB-EGF treatment additionally reduced the beating frequency of cardiospheres without altering their size (Fig. 7f, g). The effect on contractility was further assessed using human heart slices[43] showing a decline in contractility after HB-EGF treatment (Supplementary Fig. 9a, b). Additionally, HB-EGF treatment led to an increase in PDGFRα positive fibroblast area and collagen deposition in cardiospheres (Fig. 7h, i). Overall, HB-EGF induced a similar phenotype in cardiac fibroblasts and cardiospheres as compared to supernatants of *DNMT3A*-silenced monocytes shown previously in Fig. 3, suggesting that it might mediate cardiac fibroblast

activation. Indeed, treatment of fibroblasts with supernatants of DNMT3A-silenced monocytes was not only associated with autophosphorylation of the EGFR, but also induced phosphorylation of Akt, the downstream kinase of EGFR signaling, in fibroblasts (Fig. 8a–c). Thus, we can conclude that silencing of DNMT3A in monocytes induces activation of the EGFR signaling pathway.

Finally, to establish a potential causal involvement of EGF signaling in monocyte-fibroblast crosstalk, we inhibited EGFR signaling in cardiac fibroblasts using the small-molecule EGFR kinase inhibitor gefitinib[44] or neutralizing antibodies directed against HB-EGF (Fig. 8d). While gefitinib did not affect the basal expression of αSMA, it prevented the induction of αSMA by supernatants of DNMT3A-silenced monocytes (Fig. 8e, f), all without exhibiting cytotoxic effects in human cardiac fibroblasts (Fig. 8f). Moreover, EGFR inhibition with gefitinib in cardiospheres inhibited the induction of αSMA and reversed the reduced contraction of cardiac tissue mimetics induced by the treatment with the supernatant from DNMT3A-silenced monocytes (Fig. 8g, h). Finally, blocking HB-EGF by neutralizing antibodies reduced the paracrine activation of fibroblasts by DNMT3A-silenced monocytes (Fig. 8i, j). Collectively, these results demonstrate that the secretome of DNMT3A silenced monocytes induces fibroblasts activation, in part, through EGFR signaling.

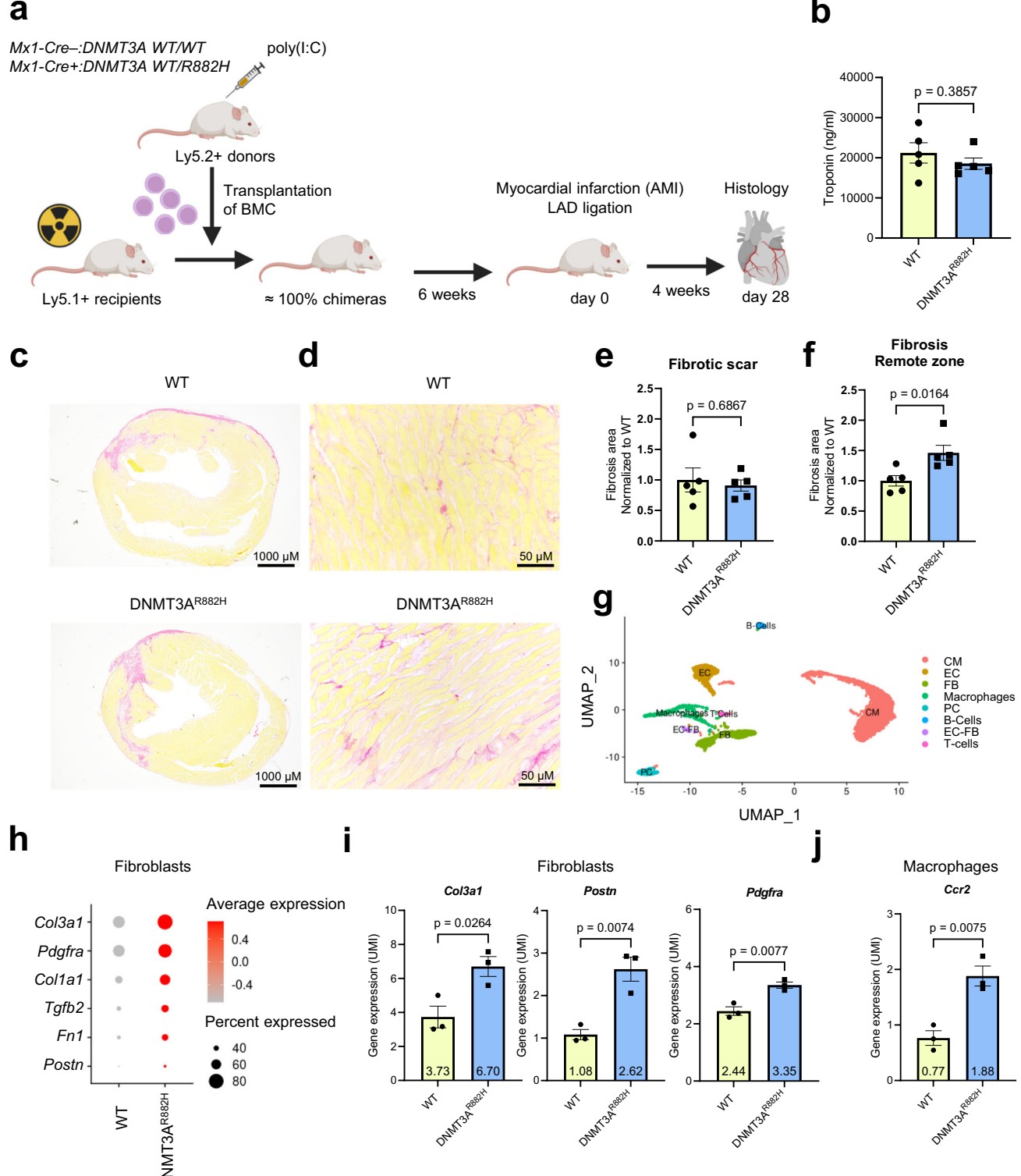

**Discussion**

In this study, we demonstrate that monocytes derived from patients with heart failure carrying DNMT3A CHIP-driver mutations may contribute to the progression of cardiac fibrosis. Building upon our in silico predictions of increased interaction between monocytes derived from patients harboring DNMT3A CHIP-driver mutations with cardiac fibroblasts, we conducted experiments to investigate the paracrine cross-talk. Our findings show that the supernatant from DNMT3A-silenced monocytes effectively activates cardiac fibroblasts in 2D and 3D cultures. Consequently, we confirmed an increased diffuse cardiac fibrosis in mice carrying hematopoietic DNMT3A CHIP-driver mutations subjected to myocardial infarction, as well as in patients with established heart failure. Collectively, these data disclose a mechanism by which DNMT3A CHIP-driver mutations contribute to heart failure: in addition to the well-known pro-inflammatory activation induced by DNMT3A CHIP-driver mutations, our findings suggest a direct interaction of DNMT3A-mutant monocytes with cardiac fibroblasts, as well as other cardiac cells.

Cardiac fibroblasts represent a heterogeneous population of cells, particularly after myocardial infarction[30,45,46]. Our observed interaction

**Fig. 4 | DNMT3A CHIP promotes diffuse cardiac fibrosis in mice with DNMT3A^R882H bone marrow cells. a** Schematic illustration of the experimental design of the in vivo study. **b** ELISA-based quantification of cardiac troponin in serum of wild-type mice transplanted with wild-type bone marrow cells (WT) and wild-type mice transplanted with DNMT3A^R882H bone marrow cells (DNMT3A^R882H) ($n = 5$ biologically independent samples). Source data are provided as a Source Data file. **c, d** Picrosirius red staining of murine cardiac cross sections. **c** Infarct fibrotic scar in WT and DNMT3A^R882H mice. **d** Remote zone in WT and DNMT3A^R882H mice. **e** Quantification of the fibrotic score for the scar and (**f**) for the remote zone ($n = 5$ biologically independent samples). Source data are provided as a Source Data file. **g** Representative uniform manifold approximation and projection (UMAP) plots of scRNA-seq sequenced WT and DNMT3A^R882H murine hearts after AMI (day 75) showing different cell clusters identified after annotation. CM Cardiomyocytes; EC Endothelial cells; FB Fibroblasts; PC Pericytes ($n = 3$ biologically independent

samples). **h** Dot plot depicting expression of fibrosis-related genes in the cardiac fibroblast cluster in the sequenced WT and DNMT3A^R882H murine hearts after AMI ($n = 3$ biologically independent samples). **i** Mean expression of *Col3a1, Postn, Pdgfra* in WT and DNMT3A^R882H murine hearts after AMI ($n = 3$ biologically independent samples). Source data are provided as a Source Data file. **j** Mean expression of *Ccr2* in WT and DNMT3A^R882H murine hearts after AMI ($n = 3$ biologically independent samples). Source data are provided as a Source Data file. Data are shown as mean ± SEM. **b, e, f, i, j** Normal distribution was assessed using the Shapiro–Wilk test. **b, e, f, i, j.** Statistical analysis was performed using unpaired, two-sided Student's *t* tests. (**b, e, f, i, j.** $t = 0.917$, 8 degrees of freedom (**b**); $t = 0.418$, 8 degrees of freedom (**e**); $t = 3.027$, 8 degrees of freedom (**f**); $t = 3.434$, 4 degrees of freedom; $t = 5.024$, 4 degrees of freedom; $t = 4.966$, 4 degrees of freedom (**i**); $t = 5.003$, 4 degrees of freedom (**j**).

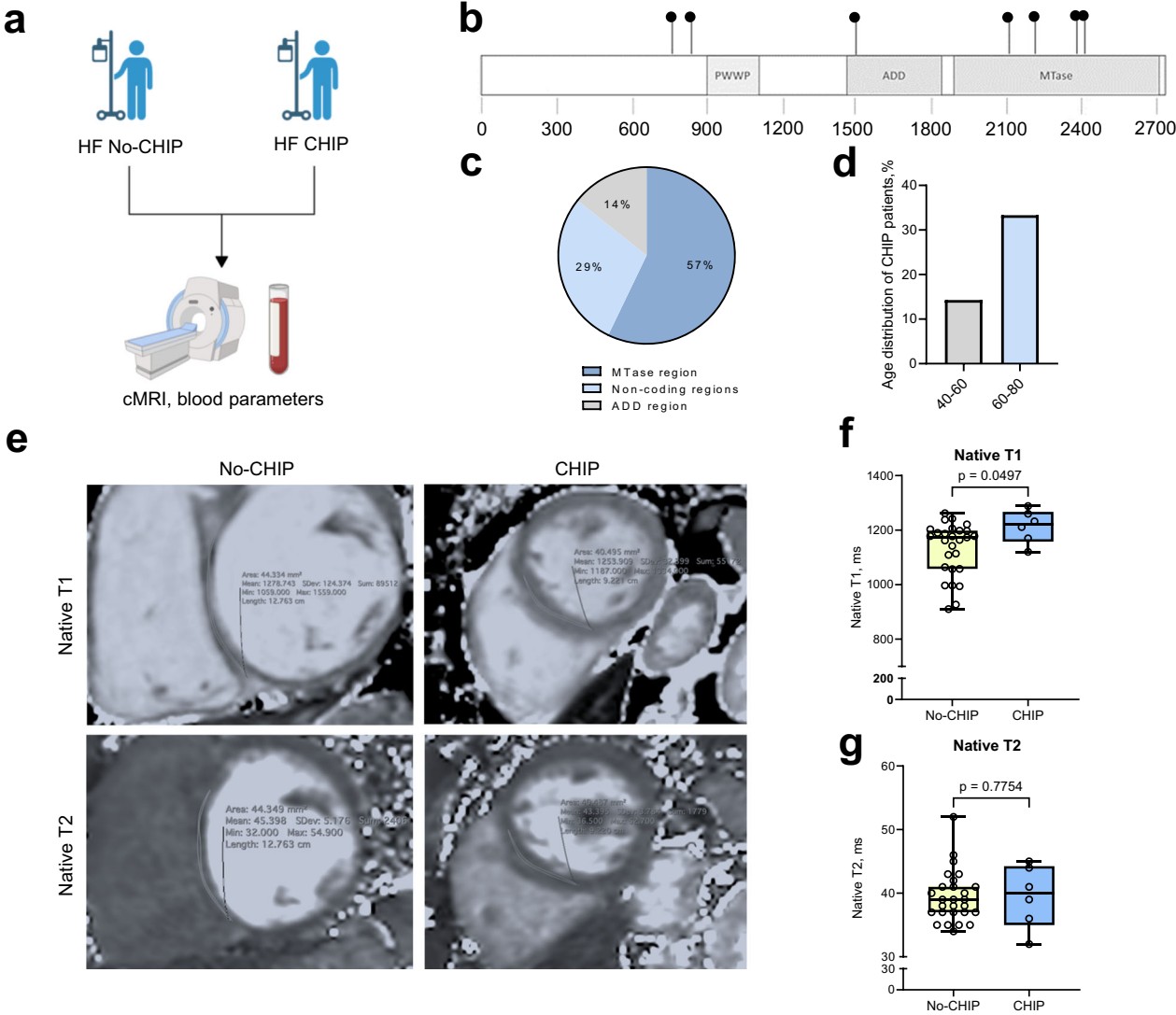

**Fig. 5 | DNMT3A CHIP promotes diffuse cardiac fibrosis in DNMT3A CHIP patients. a** Schematic illustration of the design of the patient study. **b** Distribution of 7 detected mutations of 6 CHIP patients on the *DNMT3A* gene; PWWP - Pro-Trp-Trp-Pro motif domains, ADD - ATRX-DNMT3-DNMT3L domain and MTase—methyltransferase domain. **c** Distribution of the CHIP mutations according to the DNMT3A domains. Source data are provided as a Source Data file. **d** Age distribution of DNMT3A CHIP patients. Source data are provided as a Source Data file. **e** Representative cMRI sequences of native T1 and T2 relaxation times (ms) in HF

patients with DNMT3A CHIP and No-CHIP. **f** Quantification of native T1 and (**g**) T2 relaxation times (ms) in HF patients with DNMT3A CHIP and No-CHIP ($n = 27$ for No-CHIP and $n = 6$ for CHIP). Source data are provided as a Source Data file. Data are shown as a box plot with median in the center and 25th and 75th percentiles as bounds of boxes, with whiskers indicating maximal and minimal values. **f, g** Normal distribution was assessed using the Shapiro–Wilk test. **f, g** Statistical analysis was performed using two-tailed Mann–Whitney test (**f, g**).

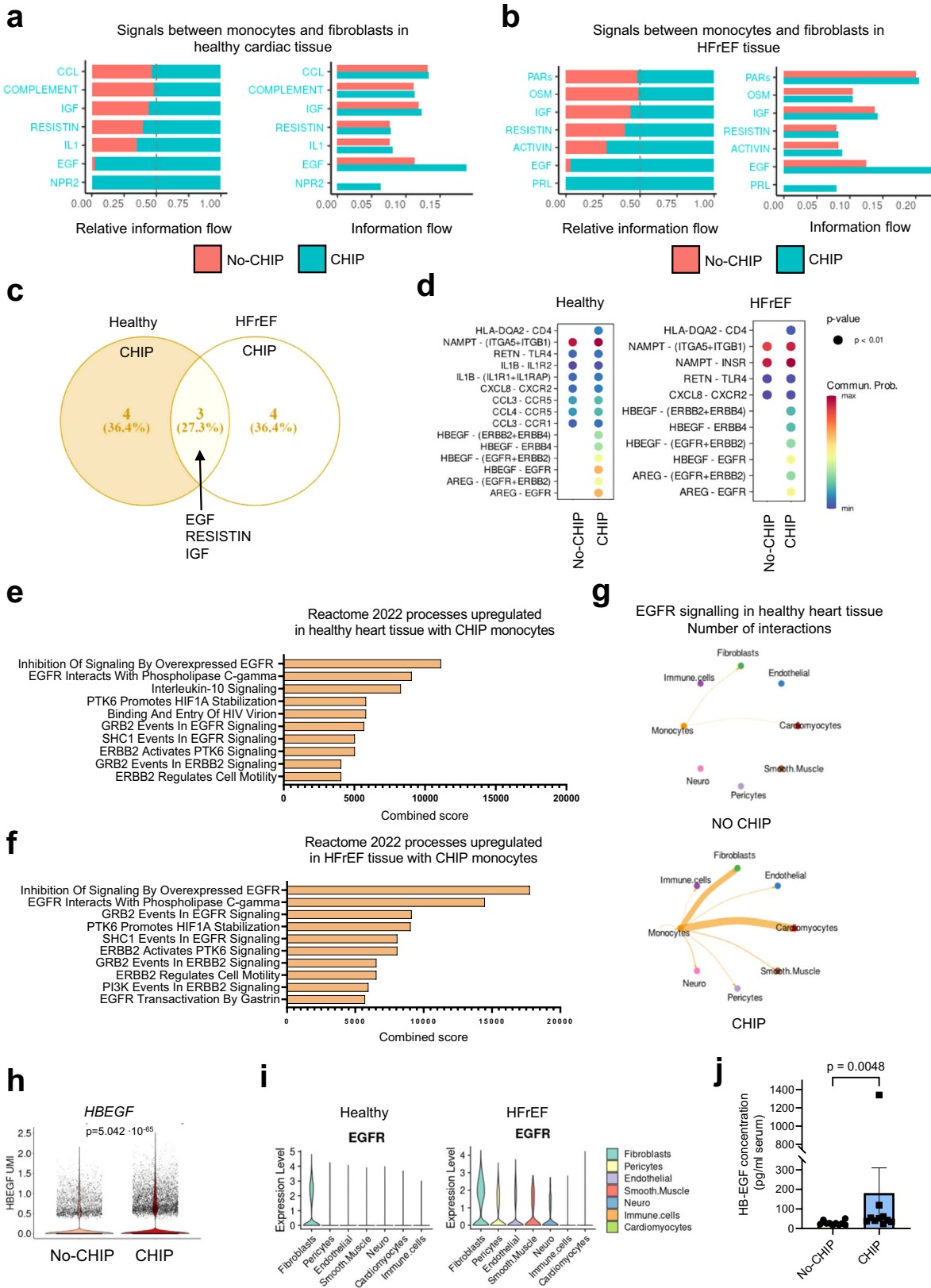

does not appear to be specific to one of the subpopulations, as most fibroblast clusters in the human heart samples showed an interaction with the DNMT3A CHIP-carrying monocytes (Supplementary Fig. 10). The activation of cardiac fibroblasts may lead to the initiation and further progression of diffuse cardiac fibrosis in patients with heart failure carrying DNMT3A CHIP-driver mutations. Since the EGF

receptor is expressed in fibroblasts of normal hearts, our bioinformatics analysis predicts that an interaction would also occur in uninjured hearts. However, in non-injured *DNMT3A^{R882H}* mice, we only observed a trend, but no significant increase in cardiac fibrosis. It is possible that additional myocardial damage resulting from infarction amplifies the recruitment of bone marrow-derived cells into the heart,

**Fig. 6 | EGF signaling contributes to CHIP monocyte-mediated cardiac fibroblast activation. a** CellChat ligand receptor prediction showing signaling pathways mediating relative information flow (left) and total information flow (right) of DNMT3A CHIP patient-derived monocytes with fibroblasts of healthy cardiac tissue. **b** CellChat ligand receptor prediction showing signaling pathways mediating relative information flow (left) and total information flow (right) of DNMT3A CHIP patient-derived monocytes with fibroblasts of HFrEF cardiac tissue. **c** Venn diagram depicting common upregulated signaling pathways in monocytes of DNMT3A CHIP carriers with healthy and HFrEF fibroblasts. **d** Bubble plots representing upregulated ligand-receptor pairs in DNMT3A CHIP monocytes-to-fibroblasts signaling in healthy (left) and HFrEF (right) myocardium. Color encodes communication probability, min. logFC for the interaction depicted is 0.1 and detection in minimum 10% of the cells. **e, f** Gene ontology analysis of differentially expressed genes (logFC for the interaction is 0.1 and detection in minimum 10% of the cells, *p*-value < 0.05) using the Enrichr database. Representation of the ten most significant functional categories in healthy (**e**) and HFrEF (**f**) interactions represented by Enrichr combined score that considers P value and Z score. **g** Circular plots depicting EGFR signaling from No-CHIP and CHIP monocytes in healthy heart tissue. **h** Violin plot depicting *HBEGF* gene expression from scRNA-seq data of monocytes from HF patients with DNMT3A CHIP and No-CHIP. **i** Violin plots showing *EGFR* gene expression in cardiac cell types from snRNA-seq data of cardiac tissue from the septum of 14 control hearts (left) and 3 HFrEF hearts (right). Violin plots represent log2-transformed and normalized UMI counts. Adjusted *p* values are based on Bonferroni correction. **j** ELISA-based quantification of HB-EGF in serum of HF patients with DNMT3A CHIP and No-CHIP (*n* = 10 biologically independent samples for both groups). Source data are provided as a Source Data file. Data are shown as mean ± SEM or box plots with min-max values whiskers. Data are shown as mean ± SEM. **j.** Normal distribution was assessed using the Shapiro–Wilk test. **j.** Statistical analysis was performed using two-tailed Mann–Whitney test (**j**).

as infiltration of circulating monocytes is augmented by endothelial activation and tissue inflammation. We indeed observed an increase in *Ccr2* expression in the macrophage cluster, indicative of recruited bone marrow-derived macrophages[24], and further detected a greater number of CD68-positive immune cells in the remote zone of the infarcted DNMT3A[R882H] mutant mice. Given that the release of inflammatory mediators can activate cardiac fibroblasts, the recruited monocytes may subsequently promote fibrosis not only via HB-EGF but also by releasing interleukins and other cytokines known to activate fibroblasts[47]. Thus, DNMT3A CHIP mutations likely amplify heart failure by pleiotropic effects on inflammation and fibrosis.

Mechanistically, we have identified HB-EGF as one of the mediators of fibroblast activation. HB-EGF plays a role in various physiological and pathological processes. HB-EGF binds to and activates the EGFR through autophosphorylation, subsequently inducing interstitial fibrosis via activation of the Akt/mTor/p70s6k pathway[48,49]. HB-EGF and EGFR families are upregulated under pathological conditions such as cardiac hypertrophy or myocardial infarction[50,51]. Moreover, shedding and activation or adenoviral overexpression of HB-EGF was shown to lead to cardiac hypertrophy and increased fibrosis, as evidenced by larger αSMA positive areas[49,52]. In line with these findings, inhibition of HB-EGF or its receptor has been found to reduce cardiac pathologies in mouse models[52,53].

Furthermore, we showed that the general inhibition of the EGF receptor pathway activation or the direct neutralization of HB-EGF can prevent cardiac fibroblast activation induced by paracrine stimulation with supernatants from DNMT3A-silenced monocytes. These data suggest that inhibition of the EGF pathway, e.g. by available FDA-approved drugs[54], could be an option to therapeutically interfere with heart failure progression in DNMT3A CHIP carriers. It is worth noting that HB-EGF has previously been shown to drive cardiomyocyte hypertrophy and induce the degradation of connexin-43[55], while EGF receptor inhibition may block arrhythmogenic cardiomyopathies[56]. Therefore, such an intervention may offer pleiotropic beneficial effects. However, EGF receptor blockers used for cancer therapies were also shown to exert adverse effects and cardiac toxicity[40]. Hence, the development of adapted, possibly lower dosing strategies might be necessary to implement this therapeutic approach.

## Limitations
Although our study offers valuable insights into the impact of DNMT3A CHIP on cardiac pathologies in mice and humans, the causal contributions of the HB-EGF-EGF receptor interactions to cardiac fibrosis have only been documented through the use of vitro models and cardiac tissue mimetics. Further research is needed to investigate whether inhibition of the EGF pathway reduces DNMT3A CHIP-induced cardiac fibrosis in vivo.

Second, our in silico prediction analysis is limited to the examination of interactions between circulating monocytes and human cardiac tissue. In reality, these interactions would not directly occur, since monocytes only can interact with cardiac fibroblast and cardiomyocytes after entering the heart. Analyzing cardiac tissue biopsies from DNMT3A CHIP mutant carriers would provide a more accurate model. However, such an analysis is hindered by the low number of immune cells in biopsies and difficulties in discriminating between the mutant and wild-type cells. Additionally, our in silico data combines single-cell and single-nuclei RNA sequencing data. Nonetheless, since the No-CHIP and DNMT3A CHIP data show similar quality, a comparative analysis of the interactions of the two groups with the cardiac tissue-derived cells appears valid. We also confirmed the predicted findings with different cell culture models, tissue mimetics and mutant mice in vivo.

Another limitation of our study is that the murine heart failure analysis was conducted in a model of ischemia-induced heart failure, while the clinical data originate from HFrEF patients, potentially with different underlying etiologies. Nevertheless, this mouse model is well-established for HFrEF and supports the casual relationship between DNMT3A CHIP-driver mutation and diffuse cardiac fibrosis.

In conclusion, our data, which suggest enhanced fibroblast activation and diffuse fibrosis in DNMT3A CHIP-driver mutation carriers, may have therapeutic implications. The assessment of DNMT3A mutation could identify patients at high risk for adverse remodeling and cardiac fibrosis, allowing for targeted treatment with new antifibrotic regimens or strategies that specifically interfere with the proposed downstream pathways.

## Methods
### Ethical considerations
Human studies comply with all relevant ethical regulations and were approved by the respective institutional ethics committees (cardiac magnetic resonance (CMR) imaging and post-processing (Fig. 5a–g, Supplementary Table 2), peripheral blood samples analysis (Fig. 6j, Supplementary Table 3) - Ethics Committee of University Hospital of the Johann Wolfgang Goethe University and the Johann Wolfgang Goethe University, Frankfurt am Main, Germany; preparation and treatment of human myocardial slices (Supplementary Fig. 9a, b) - Ethics Committee of the Ruhr-University Bochum and Erich & Hanna Klessmann-Institute, Ruhr University Bochum, Heart & Diabetes Center NRW, Bad Oeynhausen, Germany; Ethics Committee of the Ludwig-Maximilians-University Munich and Walter-Brendel-Centre of Experimental Medicine, University Hospital of the Ludwig-Maximilians-University, Munich, Germany). All participants provided written informed consent. All procedures were performed in concordance with internal standards of the German government, followed institutional guidelines, the Declaration of Helsinki and the International Conference on Harmonization of Good Clinical Practice. No compensation was provided to participants.

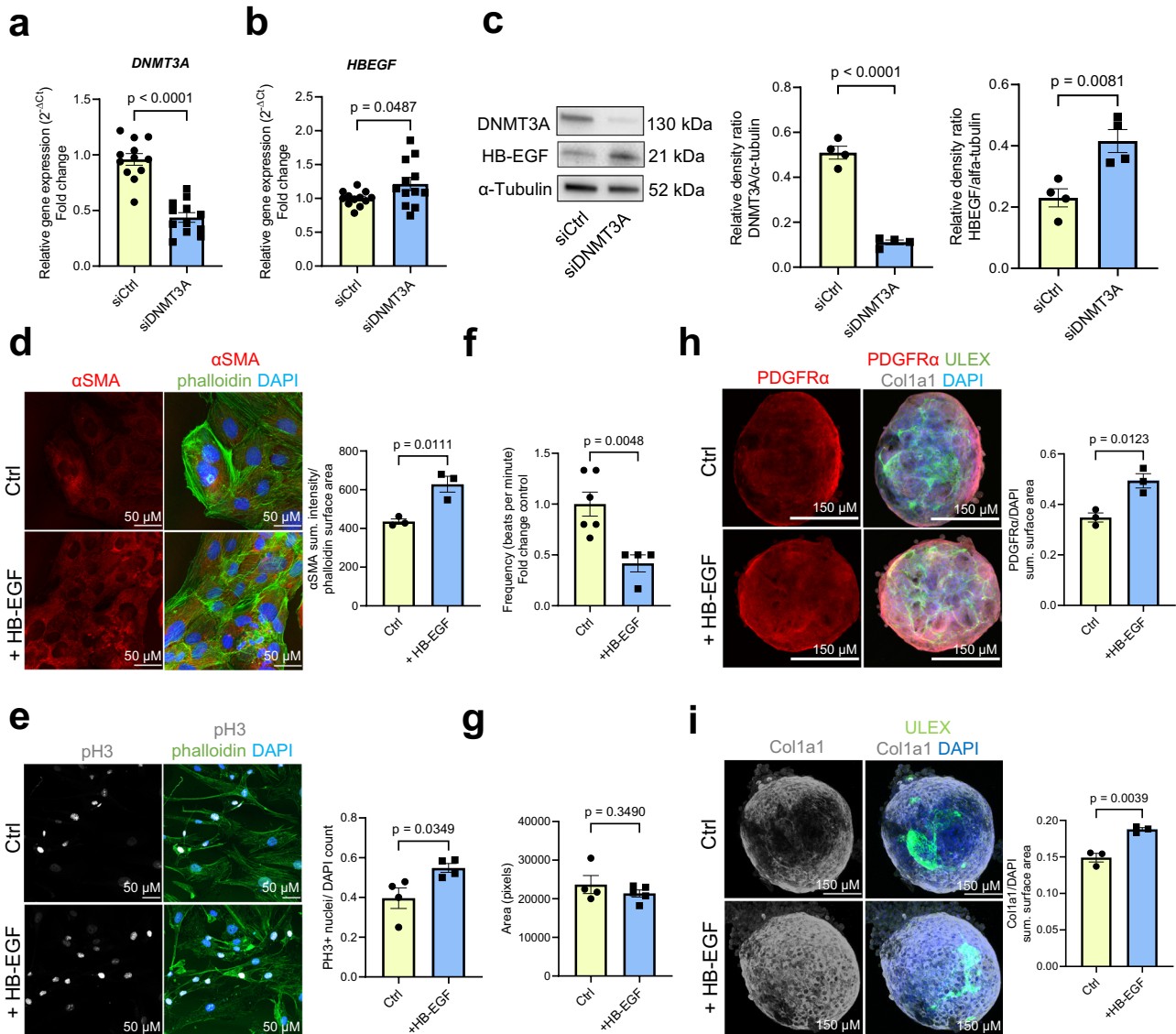

**Fig. 7 | CHIP monocyte-derived HB-EGF induces cardiac fibroblasts activation.**
**a** Relative *DNMT3A* expression and (**b**) *HBEGF* expression in DNMT3A-deficient PMA-activated THP-1 cells normalized to *RPLP0* mRNA expression analyzed by qPCR (*n* = 12 biologically independent samples). Source data are provided as a Source Data file. **c** Relative DNMT3A and HB-EGF protein expression in DNMT3A silenced PMA-activated THP-1 cells (*n* = 4 biologically independent experiments). Source data are provided as a Source Data file. **d** Immunofluorescence analysis of αSMA protein expression in HB-EGF (100 ng/ml, 48 h) stimulated iHCF (*n* = 3 biologically independent experiments). Source data are provided as a Source Data file.
**e** Immunofluorescence analysis images of phospho-histone H3 protein expression in HB-EGF stimulated HCF (*n* = 4 biologically independent experiments). Source data are provided as a Source Data file. **f** Beating frequency of cardiospheres stimulated with HB-EGF (100 ng/ml, 5 days) (*n* = 4 for controls and *n* = 6 for HB-EGF, biologically independent experiments). Source data are provided as a Source Data file.

**g** Cardiospheres area upon stimulation with HB-EGF (*n* = 4 for controls and *n* = 5 for HB-EGF, biologically independent experiments). Source data are provided as a Source Data file. **h** Immunofluorescence analysis of PDGFRα protein expression in HB-EGF stimulated cardiospheres (*n* = 4 biologically independent experiments). Source data are provided as a Source Data file. **i** Immunofluorescence analysis of Col1a1 protein expression in HB-EGF stimulated cardiospheres (*n* = 4 biologically independent experiments). Source data are provided as a Source Data file. Data are shown as mean ± SEM **a**–**i** Normal distribution was assessed using the Shapiro−Wilk test. **a**–**i** Statistical analysis was performed using unpaired, two-sided Student's *t* tests (**a**–**e**, **g**–**i**). *t* = 7.758, 22 degrees of freedom (**a**); *t* = 2.087, 22 degrees of freedom (**b**); *t* = 13.16, 6 degrees of freedom; *t* = 3.887, 6 degrees of freedom (**c**); *t* = 4.464, 4 degrees of freedom (**d**); *t* = 2.713, 6 degrees of freedom (**e**); *t* = 1.004, 7 degrees of freedom (**g**); *t* = 4.333, 4 degrees of freedom (**h**); *t* = 6.004, 4 degrees of freedom (**i**). Statistical analysis was performed using two-tailed Mann−Whitney test (**f**).

All animal experiments were executed with the animal welfare guidelines, German national laws, and EU ethical guidelines (Directive 2010/63/EU). For experiments involving rats (cell isolation for cardiac tissue mimetics, Fig. 3a–f, Fig. 8g) protocols were authorized by the competent authority Regierungspräsidium Darmstadt, Hessen, Germany. Experimental mouse models, bone marrow transplantations, left anterior descending artery (LAD) ligation, murine single-cell nuclei (sn) and single-cell (sc) RNA sequencing (Fig. 4, Supplementary Figs. 4a–c, 5a–h, 7d–g) were approved by the Regierungspräsidium Karlsruhe, Baden-Württemberg, Germany. All laboratory animal experimentation described in the manuscript adhere to ARRIVE guidelines 2.0 (ARRIVE Essential 10) for reporting animal research.

**Human single-cell nuclei (sn) and single-cell (sc) RNA sequencing data sets**
The following three datasets (two published) were used for analysis: snRNA-seq data from healthy septal cardiac tissue of 14 individuals[23], single snRNA-seq from the left ventricle of patients

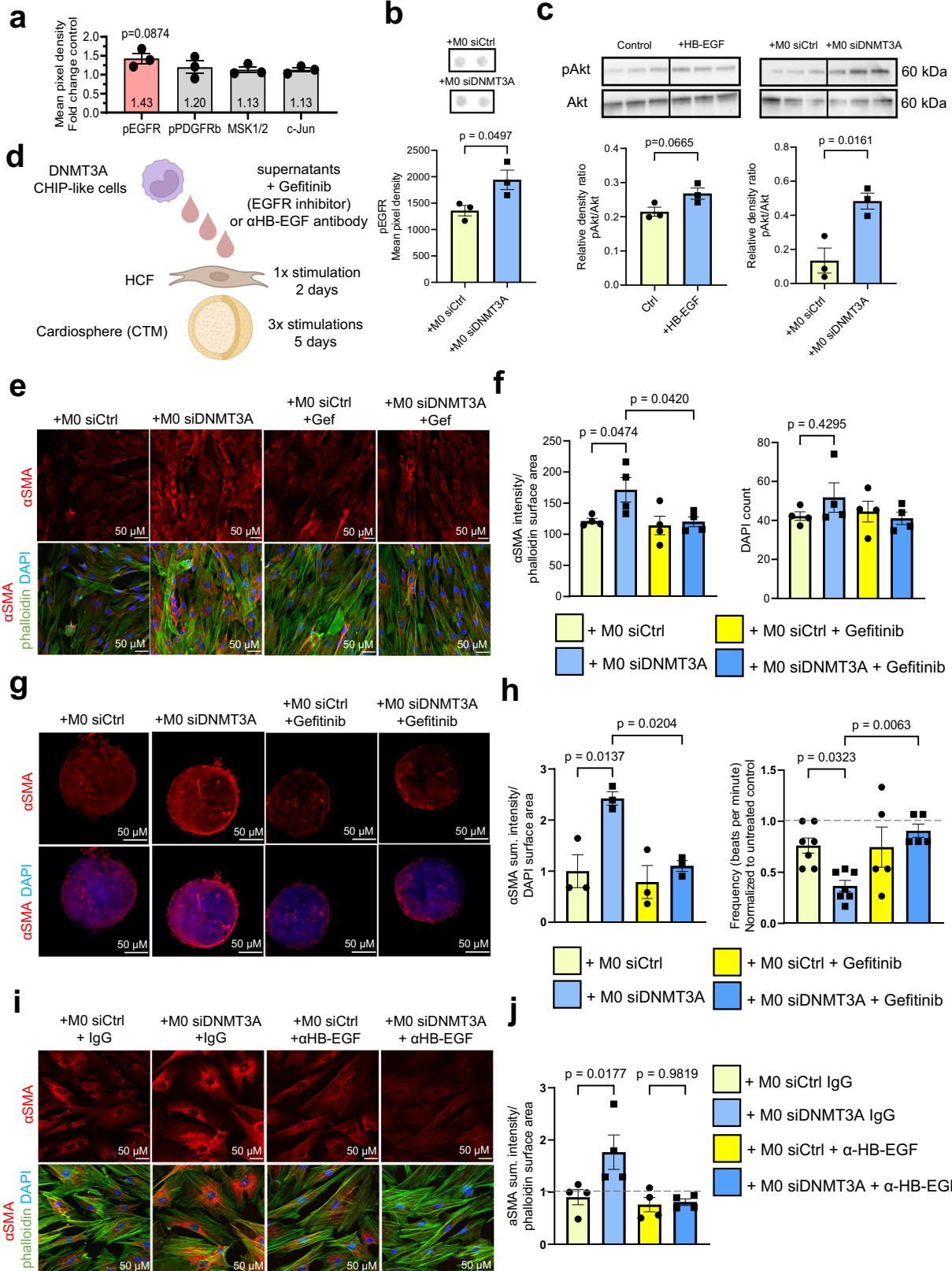

with HFrEF ($n = 3$) and scRNA-seq data from peripheral blood mononuclear cells (PBMCs) of $n = 5$ patients with heart failure (HF) harboring DNA methyltransferase 3A (DNMT3A) clonal hematopoiesis driver mutations (CHIP) and $n = 4$ HF patients without driver mutations (No-CHIP)[15].

**Integration and analysis of single-cell nuclei (sn) and single-cell (sc) RNA-sequencing data**

For integration and analysis of snRNA- and scRNA-sequencing data, we used Seurat (version 4.0.2). Before performing integration, we substracted monocytes via the subset() function of whole PBMC dataset

**Fig. 8 | Inhibition of HB-EGF-EGFR signaling reduces CHIP monocyte-mediated cardiac fibroblast activation. a** Analysis of phosphorylation of receptors and kinases in iHCF after incubation with supernatant from DNMT3A-silenced THP-1-derived cells for 30 min ($n = 3$ biologically independent experiments). Source data are provided as a Source Data file. **b** Quantification of EGFR phosphorylation ($n = 3$ biologically independent experiments) in iHCF after incubation with supernatant from DNMT3A-silenced THP-1-derived cells for 30 min. Source data are provided as a Source Data file. **c** Western blot for phospho-Akt (pAkt) and total Akt in iHCF stimulated with HB-EGF or with DNMT3A silenced monocytes ($n = 3$ biologically independent experiments). Source data are provided as a Source Data file. **d** Schematic representation of the experimental design used in (**e**–**h**). **e** Immunofluorescence analysis of HCF stimulated with DNMT3A silenced monocyte supernatants with or without gefitinib (5 µM) ($n = 4$ biologically independent experiments). **f** αSMA protein expression and cell numbers were determined by αSMA and DAPI staining in experiments shown in (**e**) ($n = 4$ biologically independent experiments). Source data are provided as a Source Data file. **g** Immunofluorescence analysis of cardiospheres stimulated with supernatant from DNMT3A silenced THP-1-cells with or without gefitinib (5 µM) ($n = 3$ biologically independent experiments). **h** αSMA protein expression and beating frequency of cardiospheres stimulated with supernatant from DNMT3A silenced THP-1-cells with

or without gefitinib ($n = 3$ for αSMA protein expression in biologically independent experiments, beating frequency in $n = 7$ for both groups without gefitinib and $n = 5$ for both groups with gefitinib in biologically independent experiments) in experiments shown in (**g**). Source data are provided as a Source Data file. **i** Immunofluorescence analysis of HCF stimulated with supernatants from DNMT3A-silenced THP-1-cells with or without HB-EGF neutralizing antibody (50 µg/ ml, 48 h) ($n = 4$ biologically independent experiments). **j** αSMA protein expression in experiments shown in (**i**) ($n = 4$ biologically independent experiments). Source data are provided as a Source Data file. Data are shown as mean ± SEM. **a**–**c**, **f**, **h**, **j** Normal distribution was assessed using the Shapiro–Wilk test. **a**-**c**, **f**, **h**, **j** Statistical analysis was performed using unpaired, two-sided Student's $t$ tests (**b**, **c**). $t = 2.783$, 4 degrees of freedom (**b**); $t = 2.503$, 4 degrees of freedom; $t = 4.004$, 4 degrees of freedom (**c**). Statistical analysis was performed using ordinary one-way ANOVA with Dunnett's correction for pairwise comparisons for data with a Gaussian distribution (**f**); ordinary one-way ANOVA with post hoc Tukey tests for data with a Gaussian distribution (panel left) and a non-parametric Kruskal-Wallis test with Dunn's multiple comparisons test for data without a Gaussian distribution (panel right) (**h**); ordinary one-way ANOVA with Šídák's multiple comparisons test for data with a Gaussian distribution (**j**).

from CHIP and No-CHIP carriers[15]. Quality control filters were applied as in the original publications[15,22]. To integrate snRNA-seq data of cardiac tissue and scRNA-seq datasets of PBMCs, we followed the standard Seurat integration vignette for different datasets (https://satijalab.org/seurat/articles/integration_introduction.html). In short, integration anchors were retrieved with the FindIntegrationsAnchors() function, followed by the IntegrateData() function to generate a combined object. Expression matrices were normalized and scaled with 'NormalizeData()', 'FindVariableFeatures()' and 'ScaleData()'. Using 'RunPCA()', we obtained the reduced dimensionality of the merged object. The resulting integration was visualized on a UMAP plot with 'RunUMAP()'. For crosstalk analysis CellChat (version 1.1.43)[21] was used. We followed the standard tutorial 'Comparison analysis of multiple datasets using CellChat' from the CellChat GitHub repository (https://github.com/sqjin/CellChat).

## Monocytes cell culture and transfection
The human monocytic cell line THP-1 was obtained from the German Collection of Microorganisms and Cell Cultures (#ACC16, DSMZ) and cultured in RPMI 1640 with GlutaMAX medium (#61870036, Thermo Fisher Scientific) supplemented with 10% heat-inactivated fetal bovine serum (FBS) (#10082147, Thermo Fisher Scientific), 10 mM HEPES (#H0887, Sigma Aldrich) and 50 U/mL Penicillin and Streptomycin (#15140122, Thermo Fisher Scientific) at 5% $CO_2$, 37 °C.

THP-1 monocytes were differentiated into THP-1-derived macrophages by stimulating $5×10^5$ THP-1 monocytes/well in 6-well plate with 100 ng/mL phorbol 12-myristate 13-acetate (PMA) (#P8139, Sigma Aldrich) for 48 h, followed by 24 h of culture in media without PMA. *DNMT3A* (HSS176225) was silenced with 50 nM Stealth RNAi siRNAs (#1299001, Invitrogen) by using siTran2.0 transfection reagent (#TT320001, Origene) and 50 nM scrambled siRNA with similar median GC content was used as siRNA control (#12935300, Invitrogen). Transfection was carried out according to the manufacturer's instructions. The media was changed 24 h post-transfection. Supernatants from transfected monocytes were collected 48 h after transfection and used for stimulation.

## Human cardiac fibroblasts cell culture and treatment
Primary human cardiac fibroblasts (HCF) (#C-12375, Promocell) or immortalized human cardiac fibroblasts (iHCF) (#P10453-IM, Innoprot) were used. HCF were cultured in Fibroblast Growth Medium 3 (#C-23130, Promocell) according to the manufacturer's protocol. HCF were seeded at a density of 5000 cells/cm² in 6-well, 12-well plates and 8-well µ-Slides (#80826, IBIDI) coated with human

fibronectin (#F0895, Sigma Aldrich, 0.1% solution; 1:1000 in water) for indirect co-culture, RNA extraction and qRT-PCR analysis. Immortalized iHCF were cultured in Dulbecco modified Eagles medium (DMEM) (#11965092, Thermo Fisher Scientific) supplemented with 10 % FBS (#16000044, Thermo Fisher Scientific) and 50 U/mL Penicillin and Streptomycin according to the manufacturer's protocol. HCF were seeded at a density of 17 500 cells/cm² in fibronectin-coated (#F0895, Sigma Aldrich, 0.1% solution; 1:1000 in water) 12-well plates and 8-well µ-Slides for indirect co-culture, RNA extraction and qRT-PCR analysis.

For stimulation with recombinant human HB-EGF (#100-47, PeproTech), fibroblasts were seeded in the respective densities one day prior to the stimulation. One day after stimulation, medium was replaced with fresh medium containing 100 ng/ml HB-EGF for 48 h.

## Human endothelial cell culture
Human umbilical vein endothelial cells (HUVECs, passage 2-3) were purchased from PromoCell (#C-12203 / C-12253) and cultured in endothelial cell basal medium (EBM, #CC-3121, Lonza) supplemented with EGM-SingleQuots (CC-4133, Lonza) and 10% FBS (FCS, #10270-106, Gibco) at 37 °C and 5% CO2. Cells were trypsinized with 0.05% trypsin (#25300062; Thermo Fisher Scientific) for 3 min at 37 °C.

## Human cardiomyocytes cell culture
Human cardiomyocytes ventricle type (HCM VT, 36044-15VT, Celprogen) were cultured in collagen-coated flasks containing cell specific serum free media (M3604415, Celprogen). Cells were cultured at 37 °C and 5 % CO2. Upon reaching the confluency, 80–90%, cells were split using TrypLE™ (12563011, ThermoFischer Scientific).

## Paracrine interaction of THP-1 cells with cardiac fibroblasts or cardiomyocytes
For indirect co-culture experiments, fibroblasts were incubated with the supernatant from transfected THP-1 macrophages. The supernatants were collected 48 h after transfection and were stored at −80 °C. THP-1-derived macrophages supernatant was mixed with the respective fibroblast cell medium at a ratio 1:1 (co-culture medium). Fibroblasts were incubated with this co-culture medium for 48 h and then were used for RNA isolation, immunofluorescent staining, and functional assays.

For the EGFR signaling inhibitor experiments, cells were treated with either 5 µM gefitinib (#SML1657, Sigma-Aldrich) or water as control. After 2 h, the medium was replaced by co-culture medium and

incubated for 48 h with or without addition of the inhibitor to the co-culture medium.

For the α-HB-EGF neutralizing antibody experiments, co-culture medium was incubated with either rabbit IgG control (#2729, Cell Signaling) or with rabbit anti-HBEGF (E5L5T) (#27450, Cell Signaling). After 2 h, the co-culture medium was applied on fibroblasts and incubated for 48 h prior to fixation and analysis.

For indirect co-culture experiments with cardiomyocytes, HCM VT were incubated with the supernatant from transfected THP-1 macrophages for 48 h and then fixed for the subsequent analysis.

## Western blot

For Western Blot analysis, protein from transfected THP-1-derived macrophages and human cardiac fibroblasts treated with macrophage supernatant or with recombinant human HB-EGF (100 ng/ml, 30 min, #100-47, PeproTech) was isolated. Briefly, cells were washed with ice-cold DPBS (#14190144, Thermo Fischer Scientific) and lysed in RIPA buffer (#R0278, Sigma-Aldrich) supplemented with protease inhibitor cocktail (#P8340, Sigma-Aldrich, 1:50) and phosphatase inhibitor (#P5726, Sigma-Aldrich, 1:100) for 30 min on ice.

Lysates were centrifuged at 12,000 g for 10 min at 4 °C. Protein concentrations were determined using the DC Protein Assay Kit II (#5000112, Bio-Rad). Proteins (20 μg) were reduced with Laemmli SDS (6×, #J61337, Alfa Aesar) and were separated on a Mini-PROTEAN TGX gel (4561094, Bio-Rad) (115 V, 50 min) and transferred onto a nitro-cellulose membrane using a semi-dry blot (25 V, 0,22 A, 50 min, Trans-Blot SD Semi-Dry Transfer Cell, #1703940, Biorad) according to the manufacturer's protocol. Membranes were blocked for 1 h in 5 % Blotto nonfat milk (#sc-2325, Santa Cruz), dissolved in Tris-buffered saline supplemented with Tween-20 (1x TBST, #sc-281695, Santa-Cruz). Primary antibodies were incubated overnight at 4 °C in 5% blocking solution (1:1000 rabbit anti-DNMT3A (E9P2F), #49768, Cell Signaling; 1:1000 rabbit anti-Akt (pan) (C67E7), #4691, Cell Signaling; 1:1000 rabbit anti-phospho-Akt (Ser473) (193H12), #4058, Cell Signaling; 1:1000 rabbit anti-HBEGF (E5L5T), #27450, Cell Signaling; 1:1000 rabbit anti-α-tubulin Antibody, #2144, Cell Signaling). On the next day, membranes were washed with 1x TBST prior to incubation with secondary antibodies. Secondary antibodies were diluted in 5% blocking solution and membranes were incubated for 1 h at room temperature (1:1000 donkey anti-rabbit IgG, HRP-linked Antibody, #7074S, Cell Signaling). Proteins were detected based on HRP substrate-based enhanced chemiluminescence (#WBKLS0500, Millipore), visualized using ChemiDoc Touch Imaging System (BioRad) and quantified using ImageLab (version 5.0).

## Ribonucleic acid analysis

For RNA isolation, cells were lysed with 350 μL RLT Plus buffer and RNA was isolated using the RNeasy Plus Mini Kit according to the manufacturer's protocol (#74034, Qiagen). RNA content and purity were assessed by spectrophotometry (NanoDrop Technologies). mRNA expression was quantified by qRT-PCR using total RNA, which was reversed transcribed by MuLV reverse transcriptase (#28025013, Thermo Fisher Scientific) and random hexamer primers (#SO142, Thermo Fisher Scientific). cDNA synthesis was performed according to the protocol: (Primer annealing) 65 °C × 5 min, (Primer elongation) 25 °C × 10 min, (Reverse transcription) 37 °C × 50 min, (Enzyme denaturation) 70 °C × 15 min. Expression levels of mRNA were detected by using Fast SYBR Green (#4385612, Applied Biosystems) and an Applied Biosystems Viia7 machine. Cycling conditions on the machine were as follows: (Hold) 95 °C x 30 sec, (PCR) 95 °C x 1 sec, 60 °C × 20 s repeated for 40 cycles, (Melt curve) 95 °C × 15 s, 60 °C × 60 s, 95 °C × 15 s. Relative gene expression was calculated with the QuantStudio Real-Time PCR software (version 1.3) using $2^{-\Delta Ct}$

($^{\Delta Ct}$= Ct target gene − Ct housekeeping gene). *RPLPO* was used as housekeeping gene.

Human primer sequences are listed below:

| | |
|---|---|
| *RPLPO-F* | ATCCGTCTCCACAGACAAGG |
| *RPLPO-R* | TCGACAATGGCAGCATCTAC |
| *DNMT3A-F* | TATTGATGAGCGCACAAGAGAGC |
| *DNMT3A-R* | GGGTGTTCCAGGGTAACATTGAG |
| *HBEGF-F* | TGTATCCACGGACCAGCTGCTA |
| *HBEGF-R* | TGCTCCTCCTTGTTTGGTGTGG |
| *AREG-F* | GCACCTGGAAGCAGTAACATGC |
| *AREG-R* | GGCAGCTATGGCTGCTAATGCA |
| *ACTA2-F* | CTATGCCTCTGGACGCACAACT |
| *ACTA2-R* | CAGATCCAGACGCATGATGGCA |
| *TGFB1-F* | TACCTGAACCCGTGTTGCTCTC |
| *TGFB1-R* | GTTGCTGAGGTATCGCCAGGAA |
| *COL1A1-F* | GATTCCCTGGACCTAAAGGTGC |
| *COL1A1-R* | AGCCTCTCCATCTTTGCCAGCA |
| *COL3A1-F* | TGGTCTGCAAGGAATGCCTGGA |
| *COL3A1-R* | TCTTTCCCTGGGACACCATCAG |
| *ADAM8-F* | TGCTGGAGGTGGTGAATCACGT |
| *ADAM8-R* | TCAGGAGGTTCTCCAGTGTGAC |
| *ADAM9-F* | CTTGCTGCGAAGGAAGTACCTG |
| *ADAM9-R* | CACTCACTGGTTTTTCCTCGGC |

## Immunofluorescence staining for cell culture

μ-Slide with 8 wells (#80826, IBIDI) were pre-coated with human fibronectin (#F0895, Sigma-Aldrich, 0.1% solution) diluted in water 1:1000 for 1 h at 37 °C. Human cardiac fibroblasts were seeded in slides in the respective medium, using HCF with $5 \times 10^3$ cells per well for αSMA staining, $2 \times 10^3$ cells/well for phospho-histone H3 staining, or iHCF with $20 \times 10^3$ cells per well. For 48 h after respective treatment, cells were washed two times with DBPS and fixed with 4% PFA (#28908, Thermo Fisher Scientific) for 10 min at room temperature. After two washing steps, each for 5 min, with DPBS, cells were permeabilized with 0.1 % Triton X-100 (#T8787, Thermo Fisher Scientific) in DPBS for 15 min. Cells were blocked with 5% donkey serum (#ab7475, abcam) in DPBS for 60 min at RT. Primary antibodies were incubated in the same blocking solution overnight at 4 °C. Cells were stained with phalloidin (1:100, #O7466, Thermo Fisher Scientific), mouse anti-α-smooth muscle actin (1:200, #C6168, Sigma-Aldrich), rabbit anti-phospho-histone H3 (Ser10; 1:200, #06-570, Sigma-Aldrich). Cells were washed 4 times with DPBS, each 5 min, and secondary antibodies in DPBS were incubated in for 1 h at RT. Following dyes and secondary antibodies were used: DAPI (1:1000, #D9542, Merck), donkey anti-rabbit-647 (1:200, #A-31573, Thermo Fisher Scientific). Cells were mounted with Fluoromount-G (#00-4958-0, Invitrogen). Images were taken using Leica STELLARIS confocal microscope. Z-Stack images were acquired with a resolution of 1024×1024 pixels, a speed of 200 frames per second, and a Z-Stack size of approximately 1 μm. Quantification was done using Volocity software version 6.5 (Quorum Technologies).

## Collagen gel contraction assay

Macrophage supernatant-treated HCF ($1 \times 10^5$ per well) were mixed with rat tail collagen type I (#354236, Corning) matrices on a 24-well plate. After 20 min of polymerization, gels were manually released from the edges of the well and were floating in culture medium only or culture medium mixed with respective supernatants. Gel surface area

was measured at 0 h and 72 h with NIS Elements (Nikon Eclipse TS100) and quantified with ImageJ Fiji 2.3.1. Gel contraction was calculated as 1/(gel surface area at 72 h/gel surface area at 0 h) and normalized to the contraction of untreated iHCF.

## Migration Assay

For analysis of cell migration, 2-well culture-inserts for self-insertion (#80209, IBIDI) were put into μ-Slide with 8 wells (#80826, IBIDI) and 7 × 10⁴ iHCF were seeded per division in the inserts. Cells were directly stimulated with mixed medium for 48 h. Images were taken at 0 h and 8 h after removing the culture-insert with a Nikon Eclipse TS100 microscope. The cell-free area was measured using ImageJ Fiji 2.3.1. Subsequently, the relative change of the migration was calculated as 1/(cell-free area at 24 h/cell-free area at 0 h).

## Cell isolation for cardiac tissue mimetics (cardiospheres)

Neonatal rat cells were isolated from Sprague Dawley P1 and P2 female and male rat pups, as described previously[22]. Pregnant female Sprague Dawley rats (>12 weeks old) were obtained from Janvier (Le Genest Saint-Isle, France). Female and male rat pups at P1 and P2 were sacrificed by cervical dislocation and hearts were transferred into Hank's buffered saline solution ($-Ca^{2+}/-Mg^{2+}$; #14025-050, Gibco) containing 0.2% 2,3-Butanedione monoxime (BDM; Sigma-Aldrich; #B0753-25G). Hearts were cut into small pieces and dissociated at 37 °C with the commercially available enzyme mix (Neonatal Heart Dissociation Kit, mouse and rat, Miltenyi Biotec GmbH; # 130-098-373) followed by tissue homogenization with the genteMACS™ Dissociator (Miltenyi Biotec GmbH; #130-093-235; program m_neoheart_01_01) in a C-tube (#130-093-237, Miltenyi Biotec GmbH). Cells were transferred to a 70 μm strainer (#352350; Falcon), centrifuged ($80 \times g$, 5 min) and were resuspended in plating medium (DMEM high glucose #1-26P02-K, M199 EBS #1-21F01-I; both from BioConcept, 10% horse serum (#16050130; Thermo Fisher Scientific), 5% fetal calf serum (#10270-106; Gibco), 2% L-glutamine (#25030149; Thermo Fisher Scientific) and penicillin/streptomycin (#11074440001; Roche) and incubated for 1 h and 40 min in 10 cm cell culture dishes (Greiner Bio-One GmbH) at 37 °C and 5% $CO_2$. Within the incubation time, the fibroblasts attach to the uncoated culture dish and can be detached by using 0.05 % trypsin (#25300062; Thermo Fisher Scientific) for 3 min at 37 °C. The supernatant contains the cardiomyocytes and was collected. Cells were counted with the Neubauer-Chamber (Carl Roth, #T729.1).

All rats were housed under standard conditions with controlled dark–light cycle, temperature and humidity in cages at the animal facility of University Hospital of the Johann Wolfgang Goethe University.

## Cardiac tissue mimetic (cardiospheres) formation

Isolated rat cardiomyocytes (CM), rat fibroblasts (FB) and HUVECs were used to form cardiac tissue mimetics (CTM) as previously described[32]. Briefly, 32,000 CMs and 6,400 FBs were cultured in plating medium as hanging drops at 37 °C and 5% CO2 for 4 days. The formed cellular spheroids were collected and cultivated in Ultra-Low Adhesion U-Bottom plates (#7007, Costar) in 50 μL plating medium per well. After 4 h of incubation, 10 000 HUVECs in 50 μL fully supplemented EBM (Lonza) were added to each well. After 24 h, the medium was changed to maintenance medium (DMEM high glucose #1-26P02-K, M199 EBS #1-21F01-I; both from BioConcept, 1% horse serum (#16050130; Thermo Fisher Scientific), 2% L-glutamine (#25030149; Thermo Fisher Scientific) and penicillin/streptomycin (#11074440001; Roche) and fully supplemented EBM (50%:50%). Mature CTMs were treated with 200 μM phenylephrine (short PE; Sigma-Aldrich; P6126-5G), every second day for 14 days and then treated with PE (200 μM) and supernatant from macrophages (50%:50%) or PE (200 μM) + 100 ng/ml recombinant human HB-EGF for 5 days with 3 stimulations. For the inhibitor experiments with gefitinib

(#SML1657, Sigma-Aldrich), cardiospheres were treated with PE containing medium and supernatant from macrophages (50%:50%), supplemented with 5 μM gefitinib.

PDGFRα positive areas, αSMA positive areas, COL1A1 positive areas, and vascularization were determined using the Leica STELLARIS confocal microscope and was quantified with the Volocity software version 6.5 (Quorum Technologies).

## Contractility measurement

Spontaneous cardiospheres contraction were determined by counting the number of beats per minute (bpm) of cardiospheres using a Zeiss Observer Z1 microscope and Axiovision 4.5 (Zeiss).

## Area measurement

Pictures of cardiospheres were taken using a Zeiss Observer Z1 microscope and Axiovision 4.5 (Zeiss). The area of cardiospheres was measured using ImageJ Fiji 2.3.1 (NIH).

## Cardiac tissue mimetics immunofluorescence staining

Whole CTMs were used for fluorescence immunohistochemistry. CTMs were collected and fixed with 4% ROTI Histofix (#P087.1, Carl-Roth GmbH & Co. KG) for 1 h at room temperature. After fixation, CTMs were permeabilized with 0.2% Triton X-100 (#T8787, Thermo Fisher Scientific) in DPBS for 1 h at room temperature followed by incubation in blocking solution (2% donkey serum (#ab7475, abcam), 3% BSA (#A7030, Sigma-Aldrich) in 0.2% Triton X-100 in DBPS) for 30 min at room temperature.

Primary antibodies and dyes (mouse anti-α-smooth muscle actin (1:40, #C6168, Sigma-Aldrich), goat anti-PDGFRα (1:20, #AF1062, R&D Systems), biotinylated Ulex Europaeus Agglutinin I (1:100, #B-1065-2, Vector Laboratories), rabbit anti-COL1A1 (1:20, 72026S, Cell Signaling)) were diluted in blocking solution and were incubated overnight at 4 °C. CTMs were washed on the next day with 0.2% Triton X-100 in DBPS three times for 20 min. Secondary antibodies and dyes (streptavidin 488 (#S11223, Thermo Fisher Scientific), donkey anti-goat (1:200 #A-21447, Thermo Fisher Scientific), donkey anti-rabbit-647 (1:200, # A-31573, Thermo Fisher Scientific), DAPI (1:1000, #D9542, Merck) were diluted in the blocking solution and incubated for 4 h at room temperature in the dark. After washing CTMs again three times with 2% Triton X-100 in DBPS, CTMs were mounted with Fluoromount-G (#00-4958-0, Invitrogen) and analyzed using the Leica STELLARIS confocal microscope with quantification performed in Volocity software version 6.5 (Quorum Technologies).

## Preparation and treatment of human myocardial slices

Left ventricular failing myocardium was obtained from the Clinic of Cardiac Surgery, University Hospital, Munich, Germany, and from the Clinic of Thoracic and Cardiovascular Surgery, Heart and Diabetes Center, Bad Oeynhausen, Germany.

Left ventricular failing myocardium was extracted from 2 male and 1 female patients, all suffering from dilatory cardiomyopathy, age 19 – 64 years. Sex and gender of participants was determined based on self-reports. All available tissues have been included irrespective of gender or sex.

Myocardial slice preparation and biomimetic culture chambers (BMCCs) have been described in previous studies[43]. After precision cutting in 4 °C, myocardial slices were mounted in BMCCs and were cultivated with mechanical preload (1000-1200 μN) and continuous electrical stimulation (0.5 Hz) in M199 medium supplemented with 1% Insulin-Transferrin-Selenium, 1% Penicillin & Streptomycin solution, 50 μM β-mercaptoethanol, and 20 nM hydrocortisone. BMCCs were connected with the rocking platforms (60 rpm, 15° tilt angle) in a standard incubator (37 °C, 5% $CO_2$, 20% $O_2$, 80% humidity). Two-thirds of the culture medium was exchanged every 2–3 days. After continuous cultivation for more than 3 weeks, myocardial slices

were treated with 100 ng/mL HB-EGF as final concentration for 9 days.

## Experimental mouse models, bone marrow transplantations and left anterior descending artery (LAD) ligation

DNMT3A$^{R882H}$ knock-in (KI) mice were generated by the Mouse Biology Program (MBP) of the University of California (Davis, CA) as previously described[34]. The DNMT3A$^{fl-R882H-fl}$ mice were crossed to B6.Cg-Tg(Mx1-cre)1Cgn/J mice (referred to as Mx1-Cre)[57] or to tamoxifen-inducible transgenic B6.129-Gt(ROSA)26Sor$^{tm1(cre/ERT2)Tyj}$/J mice (referred to as R26-Cre$^{ERT2}$ Cre recombinase−estrogen receptor T2)[58]. C57BL/6N (CD45.2+ or Ly5.2+) and B6.SJL-Ptprc$^a$Pepc$^b$/BoyJ (CD45.1+ or Ly5.1+) mice were obtained from Charles River Laboratory. Primary bone marrow cells of age-matched 16- to 28-week-old female and male Mx1-Cre + /DNMT3A$^{+/R882H}$(denoted as +/m) and Mx1-Cre-/DNMT3A$^{+/+}$(denoted as +/+) donor mice (Ly5.2+) were collected 14 days after the last pI:pC injection (Amersham; 400 μg/mouse intraperitoneally for 3 nonconsecutive days) to ensure that signaling activation and cytotoxic effects mediated by pI:pC were minimized. To validate excision efficiency, genomic DNA from blood or harvested cells was subjected to PCR.

12- to 16-week-old female recipient mice were lethally (7.5 Gy total body irradiation) irradiated using X-Ray Irradiation System MultuRad 160 (Faxitron Bioptics LLC, USA). Recipient mice were maintained on Cyprofloxacin drinking water (40/mg/kg, Fresenius Kabi Deutschland GmbH, Bad Homburg) for one week before and two weeks after transplantation and allowed to engraft for 6 weeks, before being used for analysis. For transplantation, donor bone marrow cells from primary DNMT3A$^{+/R882H}$ (Ly5.2 + ) or wild-type +/+ (Ly5.2 + ) mice were transplanted into lethally irradiated Ly5.1+ (B6.SJL-Ptprca Pepcb/BoyJ; Charles River Laboratory) recipients with a total of 2.5 ×10$^5$ cells per recipient. Peripheral blood engraftment was assessed by flow cytometry 6 weeks post-transplantation.

Six weeks after reconstitution of the bone marrow, myocardial infarction was induced by permanent ligation of the left anterior descending artery in transplanted recipients. In brief, anesthesia was induced with isoflurane (4%/800 ml O$_2$/min) and maintained by endotracheal ventilation (2–3%/800 ml O$_2$/min). Thoracotomy was performed in the fourth left intercostal space. The left ventricle was exposed, and the left coronary artery was permanently occluded. Chest and skin were closed, and anesthesia was terminated. Animals were extubated when breathing was restored. Initial myocardial injury was evaluated by measuring cardiac Troponin T levels in plasma 24 h after induction of myocardial infarction. To obtain hearts after four weeks, mice were sacrificed via cervical dislocation during isoflurane anesthesia and perfused with cold Hank's buffered saline solution (HBSS, 14175-053, Invitrogen).

For mouse single-cell nuclei (sn) and single-cell (sc) RNA-sequencing we used a transplantation model with tamoxifen-inducible Rosa26Cre$^{ERT2}$:DNMT3A-R882H (Cre recombinase - estrogen receptor T2; known as R26-Cre$^{ERT2}$ strain) intercross mice as donors. These mice do not require pI:pC treatment and are independent of intrinsic inflammatory IFN-response to induce DNMT3A-R882H expression within the hematopoietic compartment[34].

We generated complete "chimeras" mice by transplanting non-induced Rosa26Cre$^{ERT2}$:DNMT3A-R882H (Ly5.1+) hematopoietic cells (either (+/+) or (+/m), 16-28-week old, female and male) into 16-week-old female Ly5.1/2+ recipients. 6 weeks after BM cells transplantation, recipients were subjected to myocardial infarction as described above. Dissolved tamoxifen (Sigma-Aldrich) in corn oil was administered via intraperitoneal injection (1 mg/mouse/day) once every 24 h for a total of 10 consecutive days. The injections started 28 days after the induction of the myocardial infarction. The mice were used at day 74–76 post left anterior descending artery ligation through sacrifice via cervical dislocation during isoflurane anesthesia.

All mice were housed under standard conditions in individually ventilated cages with controlled dark–light cycle, temperature and humidity in cages at the animal facility of Heidelberg University Hospital according to national and institutional guidelines for animal care.

## Plasma preparation and measurement of cardiac Troponin T

Blood samples were obtained from the facial vein by a lancet 24 h post-LAD-ligation. Whole blood was centrifuged at 14,000 g for 10 min at 4 °C. Supernatants were stored until further analysis at −80 °C. For quantification of infarct size, high-sensitive Troponin T (hs-TnT) was measured using an automated Cobas Troponin T hs STAT Elecsys (Roche) as described previously[59].

## Collagen deposition analysis

To analyze fibrotic scar area and diffuse fibrosis, we checked collagen deposition in the heart by performing Sirius Red staining on paraffin sections. After deparaffinization and rehydration, the slides were stained with 0.1 % Picro Sirius Red solution prepared using Sirius Red F3BA (#1A280, Waldeck GmbH) in an aqueous solution of Picric Acid (#6744, Sigma Aldrich). Slides were mounted with Aquatex mounting medium (#HC440258, Millipore) and imaged using a Nikon Eclipse Ci microscope with a 2X (fibrotic scar) and 40X (diffuse fibrosis) objectives. The positive area was measured using ImageJ (NIH, version 1.53a) and normalized either to the heart area (scar analysis) or to the total tissue area (diffuse fibrosis).

## Immunohistochemistry

Hearts were harvested in cold PBS and fixed overnight in 4% PFA at 4 °C. Samples were embedded in paraffin, solidified and 4 μm thick sections were produced using a microtome. Slides containing the samples were melt for 30 min at 37 °C, and rehydrated by washings 2 × 10 min in 100% xylene, 5 min in 100% ethanol, 5 min in 95% ethanol, 5 min in 80% ethanol, 5 min in 70% ethanol, 5 min in 50% ethanol and 5 min in distilled water. Antigen retrieval was performed in 0.01 M citrate buffer of pH 6.0 for 90 s in a pressure cooker. Slides were then cooled down, and permeabilized with PBS containing 0.1% Triton X-100 by washing them three times for 5 min. Tissue was blocked for 1 h in freshly prepared blocking solution (3% BSA, 0.1% Triton X-100, 20 mM MgCl2, and 5% donkey serum in PBS). Primary antibodies were incubated overnight at 4 °C in a blocking solution. On the following day, the primary antibodies were washed three times 5 min in PBS containing 0.1% Triton X-100. Secondary antibodies, together with Hoechst, were incubated for 1 h at room temperature in PBS containing 0.1% Triton X-100. Slides were mounted with mounting medium and immunostainings were imaged in a Leica Stellaris confocal inverted microscope.

Primary antibodies: biotinylated Isolectin-B4 (B-1205.5; Vector Laboratories) (1:25), rabbit anti-CD68 (97778S, Cell signaling) (1:100). Secondary antibodies: Donkey anti-rabbit Alexa Fluor 488 (A21206, Invitrogen) (1:200), Streptavidin Alexa Fluor 647 (S32357; Invitrogen) (1:200). Nuclei were counterstained using Hoechst (33342; Sigma-Aldrich) (1:1000).

Stainings were quantified using Volocity Software 6.5.1 (Quorum Technologies).

## Nuclear isolation from frozen murine hearts

Cardiac tissues were thawed on ice and cut into small pieces. Minced tissue was pre-digested with a 5-ml enzyme solution of collagenase (2500 U, Thermo Fisher Scientific) in HBSS + /+ (Gibco) for 10 min at 37 °C in a water bath. After centrifugation at 500 g and 4 °C for 5 min, the supernatant was discarded, and nuclei were isolated after cell disruption with a glass dounce homogenizer (five strokes with a loose pestle and ten strokes with a tight pestle). After filtering (20-μm strainer, pluriSelect), the suspension was centrifuged at 1000 g and 4 °C for 6 min and resuspended in 500 μl staining buffer containing

1% BSA (Sigma-Aldrich), 5 nM MgCl2 (Sigma-Aldrich), 1 mM EDTA (Gibco), 1 mM EGTA (Gibco), 0.2 U μl−1 RNasin Plus Inhibitor (Promega) and 0.1 μg ml−1 Hoechst (Life Technologies) in Dulbecco´s phosphate buffered saline (DPBS). Hoechst-positive nuclei were separated from cell debris by using the FACSAria Fusion instrument (BD Biosciences) and sorted into a staining buffer without Hoechst at 4 °C.

## Single-cell RNA-sequencing library preparation

Cellular suspensions were loaded on a 10X Chromium Controller (10X Genomics) according to the manufacturer's protocol based on the 10X Genomics proprietary technology. Single-cell RNA-Seq libraries were prepared using Chromium Single Cell 3′ Reagent Kit, v3 (10X Genomics) according to the manufacturer's protocol. Briefly, the initial step consisted in performing an emulsion capture where individual cells were isolated into droplets together with gel beads coated with unique primers bearing 10X cell barcodes, UMI (unique molecular identifiers) and poly(dT) sequences. Reverse transcription reactions were engaged to generate barcoded full-length cDNA followed by the disruption of emulsions using the recovery agent and cDNA clean up with Dyna-Beads MyOne Silane Beads (ThermoFisher). Bulk cDNA was amplified using a Biometra Thermocycler TProfessional Basic Gradient with 96-Well Sample Block (98 °C for 3 min; cycled 14×: 98 °C for 15 s, 67 °C for 20 s, and 72 °C for 1 min; 72 °C for 1 min; held at 4 °C). Amplified cDNA product was cleaned with the SPRIselect Reagent Kit (Beckman Coulter). Indexed sequencing libraries were constructed using the reagents from the Chromium Single Cell 3′ v3 Reagent Kit, as follows: fragmentation, end repair and A-tailing; size selection with SPRIselect; adaptor ligation; post-ligation cleanup with SPRIselect; sample index PCR and cleanup with SPRI select beads. Library quantification and quality assessment were performed using Bioanalyzer Agilent 2100 using a High Sensitivity DNA chip (Agilent Genomics). Indexed libraries were equimolarly pooled and sequenced on an Illumina NovaSeq 6000.

## Analysis of single-cell nuclei RNA-sequencing data from murine hearts

For analysis of snRNA-sequencing data, we used Seurat (version 4.0.2). Quality control filters were applied as described above. Expression matrices of the Seurat object were normalized and scaled with 'NormalizeData()', 'FindVariableFeatures()' and 'ScaleData()'. Using 'RunPCA()', we obtained the reduced dimensionality of the object. The resulting dataset was visualized on a UMAP plot with 'RunUMAP() and gene expression was assessed with functions 'VlnPlot()' and'DotPlot()'.

## Cardiac magnetic resonance (CMR) imaging and post-processing

All participants underwent a standardized CMR imaging protocol at the Centre for Cardiovascular Imaging, University Frankfurt, on a 3 T scanner equipped with advanced cardiac software and a multi-channel coil (Skyra and Prisma, Siemens Healthineers, software version VE11) as previously described[37,38]. The study was approved by the Ethics Committee of the University Hospital of the Johann Wolfgang Goethe University in compliance with the internal standards of the German government, and procedures followed were in accordance with institutional guidelines (application 347/18) and the Declaration of Helsinki. Patients provided their written informed consent and no compensation was provided to participants. The sex and/or gender of participants were determined based on self-reports. Acquisition of cardiac function, volumes, mass, myocardial mapping and scar imaging was performed. Imaging parameters and scanning and shimming procedures for all sequences were standardized and mandatorily performed by all operators in all scans. Comparability and reproducibility of measurements were determined at each location. Slice thickness in all acquisitions was set uniformly at 8 mm. Cine imaging was performed using a balanced steady-state free precession sequence in combination with parallel imaging (GRAPPA) and retrospective gating during expiratory breath-hold (TE/TR/flip-angle: 1.7 ms/3.4 ms/30°; spatial resolution, 1.8 × 1.8 ×8 mm), as a short-axis (SAX) stack for assessment of cardiac volumes and function or single-slice long-axis views (two-chamber, three-chamber and four-chamber views). Cardiac volumes, function and mass were measured in line with standardized post-processing recommendations[37,38]. Cine images were employed for derivation of GLS analyses using Medis Suite MR version 2.1 (Medis Medical Imaging Systems). Myocardial mapping was acquired in a single midventricular slice and measured conservatively in the septal myocardium[8].

## Gene ontology analysis

Gene ontology analysis was performed using the Enrichr database (http://amp.pharm.mssm.edu/Enrichr/)[60,61].

## ELISA

Peripheral blood samples were taken from heart failure with reduced ejection fraction (HFrEF) patients at the Department of Cardiology, University Frankfurt, in the frame of the UCT-Project-Nr.: KardioBMB#2022-004. The study was approved by the Ethics Committee of the University Hospital of the Johann Wolfgang Goethe University in compliance with the internal standards of the German government, and procedures followed were in accordance with institutional guidelines (application 347/18) and the Declaration of Helsinki. Patients provided their written informed consent for the scientific use of blood samples and no compensation was provided to participants. Sex and/or gender of participants were determined based on self-reports. Aliquots of serum and whole blood samples were collected and stored at −80ºC. The serum was diluted 1:4 prior to the analysis. Diluted serum was then used to measure human HB-EGF with the DuoSet ELISA Kit (DY259B, R&D Systems).

## Next-generation sequencing

Whole blood from patients with heart failure with reduced ejection fraction (HFrEF) was used for DNA isolation and next-generation sequencing as previously described[10]. Briefly, next-generation sequencing was commercially performed by MLLDxGmbH, Munchen, Germany. DNA was isolated with the MagNaPure System (Roche Diagnostics, Mannheim, Germany) from mononuclear cells after lysis of erythrocytes. The patients' libraries were generated with the Nextera Flex for enrichment kit (Illumina, San Diego,CA, USA) and sequences for DNMT3A enriched with the IDT xGen hybridization capture of DNA libraries protocol and customized probes (IDT, Coralville, IA, USA). The libraries were sequenced on anIllumina NovaSeq 6000 with a mean coverage of 2147x and a minimum coverage of 400x. Reads were mapped to the reference genome (UCSC hg19) using Isaac aligner (v2.10.12) and a small somatic variant calling was performed with Pisces (v5.1.3.60).

## Human phospho-kinase array

The human Proteome Profiler Human Phospho-Kinase Array Kit (#ARY003C, R&D Systems) was used to assess receptor phosphorylation and kinase activation of HCF treated with macrophages supernatants for 30 min. Briefly, HCF was stimulated with mixed medium in a 6-well plate, lysed in 334 μl of a provided lysis buffer and processed according to the manufacturer's protocol.

## Graphical figures

Graphical figures were originally created with BioRender.com or adapted from BioRender.com templates with modifications to original content and/or design using (licence number: 28D5A348-0001).

## Statistical analysis

Data are biological replicates and are represented as mean and error bars indicating standard error of the mean (SEM). Data were statistically assessed for Gaussian distribution using Shapiro–Wilk, Kolmogorov–Smirnov and Anderson–Darling test. Statistical significance for data with a Gaussian distribution was calculated using two-sided, unpaired Student's $t$ tests for two-group comparison. For data not following a Gaussian distribution, statistical analysis was performed using Wilcoxon–Mann–Whitney test for two-group comparison or one sample $t$ test. For multiple comparisons, ordinary one-way ANOVA with a post hoc Tukey's, Holm-Sidak's or Dunett's multiple comparison (following Gaussian distribution) or a Kruskal–Wallis test with a post hoc Dunn's multiple comparison (not following Gaussian distribution) and two-way ANOVA with Fisher's LSD multiple comparisons posttest was used. Fisher's exact test was used in the analysis of contingency tables. For single-cell RNA Sequencing, statistical analysis of differential gene expression was performed with Seurat's FindMarkers or FindAllMarkers functions by using a bimodal likelihood estimator test suitable for scRNA-seq or snRNA-seq data. Data is always shown in p-values corrected for multiple testing (Bonferroni). All calculations were performed in GraphPad Prism 9.3.0.

## Reporting summary

Further information on research design is available in the Nature Portfolio Reporting Summary linked to this article.

# Data availability

The single-nuclei RNA-Seq data generated in this study have been deposited in the ArrayExpress Data Depository (https://www.ebi.ac.uk/biostudies/arrayexpress) with series accession number E-MTAB-13384. The following datasets from ArrayExpress Data Depository were reused in this manuscript: accession number E-MTAB-13016 (monocytes scRNA-seq dataset) and E-MTAB-13264 (human HFrEF snRNA-seq dataset). The used snRNA-seq dataset of the septum from healthy heart samples was taken from Heart Cell Atlas (version 2, https://www.heartcellatlas.org/#DataSources). Clinical data from Myoflame 19 study (EudraCT 2022-00162-12, NCT05619653) was reused in the publication. CellChat analysis package and related tools are accessible under https://github.com/sqjin/CellChat. The Enrichr Database used in the manuscript could be found under https://maayanlab.cloud/Enrichr/. Associated Source Data file is provided within the article. All other data included in the manuscript, source data (with statistical tests performed), and supplementary data are provided in the main article and/or at the referenced depositories or are available upon request per e-mail: dimmeler@em.uni-frankfurt.de. Source data are provided with this paper.

# Code availability

All R-script and generated code is available at GitHub under https://github.com/djhn75/Nat-Com-DNMT3A-Analysis.

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

## Acknowledgements

S.D. is supported by the Dr. Robert Schwiete Stiftung, the Clusterproject ENABLE funded by the Hessian Ministry for Science and the Arts, the German Research Foundation (DFG; SFB 1366, Project B04; SFB 1531 Project 456687919). A.M.Z. is supported by the ERC advanced grant CHIP-AVS Zeiher (Project number 101054899), and DZHK Cellular Heterogeneity. S.C. is supported by the DFG (SFB 1531, Project number 456687919; Project B10). V.P., E.N., I.H. are supported by the DZHK. Views and opinions expressed are however those of the authors only and do not necessarily reflect those of the funding agencies. Neither the European Union nor the granting authority can be held responsible for them. We thank Dr. Hendrik Milting for providing human myocardial tissue samples

## Author contributions

M.S., G.L., A.M.Z., and S.D. designed research; M.S., V.P., I.H., B.S., A.D., S.F.G, M.M.R, D.J., Z.S., A. Den. performed research; V.P, E.N., J.H. provided samples and CMR data of patients with heart failure; S..C. and K.K. provided serum of patients with HF and DNMT3A mutations; I.H., X.L., M.Sc., C.M.T., and F.L. provided samples of WT and DNMT3A CHIP mice; M.S., V.P., W.T.A., J.H., D.J., S.C., and S.D. analyzed data; M.S., G.L., A.M.Z., and S.D. wrote the paper.

## Funding

## Competing interests

A. Den. is a co-founder and shareholder of InVitroSys GmbH. A.M.Z. is unpaid consultant for TenSixteen Bio. All other authors have no patents or other financial conflicts, but they are supported by grants from the DFG, ERC and DZHK.

## Additional information

[1]Institute for Cardiovascular Regeneration, Goethe University Frankfurt, Theodor-Stern-Kai 7, 60590 Frankfurt am Main, Germany. [2]German Center of Cardiovascular Research (DZHK), Partner Site Rhine/Main, 60439 Frankfurt am Main, Germany. [3]Cardiopulmonary Institute (CPI), 60590 Frankfurt, Germany. [4]Department of Internal Medicine III, University Hospital Heidelberg, University of Heidelberg, Im Neuenheimer Feld 410, 69120 Heidelberg, Germany. [5]German Center of Cardiovascular Research (DZHK), Partner Site Heidelberg/Mannheim, 69120 Heidelberg, Germany. [6]Department of Medicine V, Hematology, Oncology and Rheumatology, University of Heidelberg, Im Neuenheimer Feld 410, 69120 Heidelberg, Germany. [7]Walter-Brendel-Centre of Experimental Medicine, Hospital of the Ludwig-Maximilians-University Munich, Marchioninistraße 68, 81377 München, Germany. [8]Department of Medicine, Cardiology, University Hospital Frankfurt, Theodor-Stern-Kai 7, 60590 Frankfurt am Main, Germany. [9]Institute of Experimental and Translational Cardiovascular Imaging, Centre for Cardiovascular Imaging, University Hospital Frankfurt, Theodor-Stern-Kai 7, 60590 Frankfurt am Main, Germany. [10]These authors contributed equally: Andreas M. Zeiher, Stefanie Dimmeler. ✉e-mail: dimmeler@em.uni-frankfurt.de

