## [Peer Review File · Nature Communications]

REVIEWER COMMENTS

Reviewer #1 (Remarks to the Author):

The study presented in this manuscript describes a novel mechanistic pathway in which DNMT3A mutations in clonal hematopoiesis of indeterminate potential (CHIP) can promote cardiac fibrosis in HFrEF. Analysis of clinical single nuclei RNA sequencing datasets identified cross-talk between DNMT3A-CHIP circulating monocytes and cardiac fibroblasts. DNMT3A-silenced monocytes activated fibroblasts and promoted myofibroblast differentiation in cell culture models. Mice and patients with DNMT3A CHIP show greater interstitial cardiac fibrosis. Finally, mechanistic studies identified EGF as a potential monocyte-to-fibroblast signaling pathway promoting myofibroblast differentiation and cardiac fibrosis. These novel findings provide evidence that CHIP mutations may promote poor cardiovascular outcomes through more causal mechanisms than the previously identified inflammation pathways.

As presented, the study shows strong evidence to causally connect DNMT3A CHIP with cardiac interstitial fibrosis that is characteristic in HFrEF. EGF signaling in DNMT3A CHIP monocytes may play a role in promoting this fibrosis, which would provide an exciting direct mechanistic connection between CHIP and cardiac myofibroblast activation in cardiac fibrosis. However, there are several issues pertaining to analysis rationale and alternative hypotheses that limit the overall strength of the findings. These are mainly regarding snRNAseq analysis, effects of DNMT3A-KO macrophages, and TGF- β signaling. Specific concerns are detailed below.

1. The retrospective single cell analysis compares circulating monocytes with human heart tissue in order to identify potential crosstalk pathways, but it is unclear whether the circulating monocytes can actually affect cardiac cell types. Were monocyte populations found in the human heart tissue snRNAseq dataset that are comparable to the circulating monocytes? Could these two populations overlap with integrated snRNAseq analysis? If so, the crosstalk connections identified here have much more direct significance, and this should be highlighted. If not, the crosstalk ligand-receptor pairs need to be further analyzed and annotated for pathways that can affect cardiac resident fibroblasts from circulating monocytes that are not recruited to the heart tissue. Many ligand-receptor pairs require direct contact or immediate proximity, but if the circulating monocytes are not found in the heart tissue samples then these connections are of questionable relevance. This is an important point.

2. It is unclear why the snRNAseq datasets of circulating CHIP and non-CHIP monocytes from HFrEF patients were analyzed with the snRNAseq dataset of healthy cardiac tissue and not a snRNAseq dataset from HFrEF patients. The authors should explain this rationale of comparing HFrEF samples to healthy heart samples or perform similar analyses using cardiac snRNAseq datasets from HFrEF patients as well.

3. There is clearly a causal relationship between DNMT3A-CHIP and cardiac interstitial fibrosis. However, the relative contribution of DNMT3A-CHIP monocytes to cardiac fibrosis, as compared to macrophages or inflammatory cytokines known to be increased in DNMT3A-CHIP leukocytes, is not clear from this study. Presumably, macrophages in the injured cardiac tissue could have a stronger effect on fibroblasts, as the macrophage-to-fibroblast communication has been detailed previously. Although there is cross-talk found between circulating monocytes and cardiac resident fibroblasts, are there fibrosis-related pathways between macrophages and fibroblasts? Are these pathways increased in HFrEF rather than the healthy heart? Or, are the pro-fibrotic signals higher from circulating monocytes than cardiac macrophages?

4. From the cardiosphere experiment, it seems possible that DNMT3A-silenced monocytes directly impair cardiac contraction. Is there evidence of this crosstalk within the snRNAseq CellChat analysis? Additionally, it would be beneficial to explain the cellular components of cardiospheres in the results paragraph introducing this experiment. It was unclear that cardiospheres contain fibroblasts.

5. Monocyte-mediated TGF- β signaling is dismissed and EGF signaling is investigated instead. What is the DNMT3A-silenced monocyte contribution to TGF- β signaling in HFrEF, and how does this affect fibroblast activation relative to EGF signaling?

Reviewer #2 (Remarks to the Author):

The study suggests a role for Dnmt3a clonal hematopoiesis driving mutations in the development of cardiac fibrosis. The authors use bioinformatic analysis of published data to provide a rationale for focusing on monocyte: fibroblast interactions. They show that in vitro dnmt3a CHIP mutations activate a fibrogenic profile in fibroblasts. In vivo, Dnmt3 mutation in hematopoietic cells accentuates interstitial fibrosis following MI. The effects are attributed to activation of a monocyte-driven EGFR axis.

General comment:

The study has substantive novel content, extending our understanding on a hot topic; the potential involvement of CHIP in cardiac fibrosis. Concerns regarding a) the selective effects on the remote non-infarcted myocardium, b) better documentation of the EGF ligand expression in vivo and c) the quality of the infarction experiments need to be addressed.

Major comments:

1. The infarction experiments (Figure 3) raise concerns. The representative images show no infarction in the NO CHIP group and a tiny segment of transmural fibrosis in the CHIP group. The scar size quantifications also suggest very small infarcts. Please show data consistent with an LAD territory infarct. This is important because the authors suggest no effects of the Dnmt3 mutation on infarct healing.

2. Sham experiments should be performed to exclude effects of transfer of mutant hematopoietic cells in the absence of injury.

3. The authors suggest that the effects of the mutation are limited to the remote myocardium. What is the basis for this selective effect? Monocytes infiltrate the infarct and activate reparative myofibroblasts. The selectivity for the remote zone is impossible to justify. Could the effects on interstitial fibrosis reflect perturbations in the healing infarct that the authors did not detect? (because of their limited analysis of the composition of the infarct). Such perturbations may affect filling pressures, thus causing secondary increases in interstitial fibrosis in the non-infarcted myocardium.

4. Some evidence supporting the proposed EGFR-mediated mechanism in vivo would strengthen the manuscript. Perhaps, the authors could show increased levels of EGFR ligands in myeloid cells in the remodeling infarcted heart.

5. Do THP1 monocytes recapitulate the cardiac macrophage population (resident and recruited)? Considering the heterogeneity of macrophages, this has to be better justified.

6. Figure 1: Was matrix synthesis and/or metabolism modulated in fibroblasts stimulated with the supernatant of the Dnmt3a cells?

Minor:

Considering that the samples are not from the same subjects, the bioinformatic support for the monocyte: fibroblast seems artificial. The authors are well-justified to hypothesize the role of monocyte/fibroblast interactions, on the basis of the very extensive evidence supporting roles of monocyte/macrophages in modulation of fibroblast phenotype.

Please briefly explain the rationale for the use of the Mx-Cre mice and the protocol for Cre activation in these animals. Also considering the hypothesis (which involved monocytes), why not use a myeloid cell Cre to generate cells for transfer?

Reviewer #3 (Remarks to the Author):

The manuscript from Shumliakivska et al investigate how DNMT3A clonal hematopoiesis-driver mutations can affect cardiac fibroblast activation, and identify monocyte HB-EGF and fibroblast EGFR as a potential axis regulating the increased fibroblast activation associated with DNMT3A CHIP or DNMT3A silencing. The paper is well written, and I enjoyed reading it. I do think that the manuscript needs to be strengthened in several parts, and I provide here below a series of comments that I hope can help the authors.

- The authors should definitely strengthen the first connection between DNMT3A CHIP and the cardiac fibroblasts at the beginning of the paper (Figure 1a-c), as then all the story develops from this premise. The authors show in Figure 1c that among the cardiac cell types, the DNMT3 circulating monocytes have the strongest crosstalk connection with the fibroblasts. But is this compared to the level of how NO-CHIP monocytes crosstalk to the cardiac fibroblasts? I think this is a critical point that the authors should explain better. How does Figure 1c look when assessing the cellular crosstalk connection between NO-CHIP monocytes and the other cardiac cells?

- Given how heterogenous is the cardiac fibroblast population, especially in disease, I wonder if the authors could go a bit more in depth in understanding which fibroblast cell states might interacting more with CHIP monocytes. After using the data from Litviňuková et al, it would be interesting to leverage a few of the many single cell transcriptional data that are publicly available and centered on the cardiac fibroblasts, to understand which fibroblast clusters might be more involved in a paracrine interaction with NO CHIP versus DNMT3A CHIP monocytes. Some examples could be PMID: 32130914 and PMID: 30912746. The exact same analysis in Figure 1c could be performed using the NO CHIP and the DNMT3A CHIP monocytes data from the author's previous work PMID: 33155517, but this time plotting the connections with the different fibroblast states/clusters.

- The quality of the collagen contraction gel images in Figure 1j is really not that good, hence very difficult to see differences between the 2 conditions. Also, I suggest the authors not only to measure at 24hr, but also at later time points as usually in this assay the gel contraction is more evident at later time points.

- It would help the readers if a schematic figure is added (similar to Fig 1f) to summarize how the experiment related to Figure 5d-i is performed.

- While the experiments with the EGFR kinase inhibitor Gefitinib strengthen the hypothesis the EGFR signaling pathway is involved in the increased fibroblast activation observed in the conditioned medium experiments with DNMT3 silencing in the monocytes, these don't really prove to which extent HB-EGF specifically is involved. Can the experiments shown in Figure 5d-i be repeated by incubating the conditioned medium with a HB-EGF neutralizing antibody prior moving the medium to the cardiac fibroblasts and the cardiospheres? These experiments would more specifically indicate the role of HB-EGF. Please see PMID: 23251664 or PMID: 20332144, just to highlight a few works where anti HB-EGF antibodies have been used.

- Why is cardiac function not shown for the experiment described in Figure 3a? Is the transplantation with the Ly5.2 D3A+/m worsening the function (EF or FS) as compared to Ly5.2 WT? Even if not the case, this is important to know.

- The experiment described in Figure 3a-c demonstrates that the DNMT3A CHIP increases interstitial fibrosis in an MI model, but the extent by which HB-EGF is involved is not clear. If the authors do find a neutralizing HB-EGF antibody that works in their hands, they should repeat the BMC transplantation experiments, and treat the mice with an anti HB-EGF antibody (this could be done over the period of the reconstitution or/and after MI), and see whether this decreases the fibrosis upon MI. If this is not possible (good antibodies not available or not effective in vivo in mouse), why not using an EGFR inhibitor that can neutralize mouse EGFR in vivo? I think these experiments would provide an in vivo

validation that HB-EGF or at least EGFR are strongly involved in the DNMT3A CHIP cardiac phenotype, hence really strengthen the study.

- Again, referred to Figure 3a-c: I was surprised to see that the authors did not compare the effect of the BMT with the Ly5.2 WT or D3A+/m at baseline, without cardiac stress. I think this is a very important point as in this model the upstream stressor is DNMT3A CHIP, and it would be important to understand how much fibroblast activation and fibrosis DNMT3A CHIP trigger in vivo even in the absence of an HF model.

- The authors only mention in Figure 3b that the scar size was not significantly affected, but when looking at interstitial fibrosis (Figure 3c) the difference is significant. I would at least attempt to speculate while this could be the case.

- I strongly suggest the authors to revisit how the statistical significance is shown in the figures as it sometimes confusing and it's not clear which groups are compared (for example please see Figure 5h).

- Figure 4a: I think it would make more sense to show the differences in signaling from the CHIP monocytes to specifically the fibroblasts, not all cardiac tissues. At this point of the manuscript, the authors have already established that their focus is understanding how CHIP monocytes can affect fibroblast and fibrosis, so showing the signaling pathway governing the connection between CHIP monocytes and all cardiac cell is not that relevant at this point.

- Along the same lines of my first comment: 4b should show the GO terms of the enriched interactions between CHIP monocytes/fibroblasts versus WT monocytes/fibroblasts.

Reviewer #4 (Remarks to the Author):

Shumliakivska et al. investigated the interaction between fibroblasts and CHIP-mutated monocytes in heart failure (HF) disease progression and fibrotic tissue formation. While the authors used wide range of functional assays to prove their hypothesis, there are crucial issues in the manuscript that need to be edited in experimental design, analysis of scRNA-seq data and manuscript writing.

Major comments

1. The introduction section is very short, and major issues are missing, for instance:

- What is known about monocytes, macrophages and stromal cells in HF models, and specifically in CHIP?
- Introduce the main methods/cellular functions which are discussed in the manuscript, such as the different scRNA-seq-based methods to infer cell-cell communications.

- What are the main results and observations of the study?

2. In the scRNA-seq data analysis, the authors claim they integrated data of circulating monocytes derived from patients with chronic heart failure with publicly available single-nuclei RNA-seq data of human heart tissue. The authors need to specify and explain more about the data-integration procedure (single-cell with single-nuclei). Which method for integration did they use? They should present QC figures in the supplementary data.

3. The scRNA-seq data should be thoroughly analyzed in the gene expression level.

In cell annotation – based on which gene programs the authors annotated the cell types/states. Are there monocytes and monocyte-derived macrophages in the myeloid cluster?

The cell-cell interaction transcriptome analysis (CellChat) should be presented also in the gene, and not only the cellular, level. The authors should present the top X ligands and receptors, which represent the hub signaling between monocytes and fibroblasts. Moreover, to show in supplementary the L-R genes related to other pairs.

4. Figure 3 – the examinations and tests that were done in the human tissues are not clear. What exactly was measured? How are these measurements correlated directly to the monocytes/macrophages and to their crosstalk with the fibroblasts?

5. In Figure 4 the authors present a crosstalk mechanism between the DNMT3A-silenced monocytes and the fibroblasts via HB-EGF-EGFR and AREG-EGFR.

a. The authors need to show that the expression of the ligands is highly expressed by the monocytes, but not the other cell types that they analyzed.

b. Moreover, is this signaling was also activated in the chimeric mice that the authors performed in Figure 3a?

6. The discussion section is superficial, and missing many known literature in the fields of: Fibroblasts, EGF signaling, immune cell function in heart failure and in fibrotic tissues, scRNA-seq and intercellular communication of myeloid cells and fibroblasts in heart tissue and other organs in tissue homeostasis and pathology, etc.

Minor comments:

1. The authors stated they used CellChat, however, in Figure 1a they wrote CellChatDB. While there is a pipeline named CellPhoneDB, this is not clear which computational pipeline was used by the authors in their analysis.

2. Experimental procedures of chimeric mice preparation should be explained in more details in the Results section. Moreover, what was the percentage of the chimeras? What is the portion of the Ly5.2 monocytes/macrophages out of Ly5.1 cells in the heart following LAD ligation?

Response to the reviewers

Response to Reviewer #1:

The study presented in this manuscript describes a novel mechanistic pathway in which DNMT3A mutations in clonal hematopoiesis of indeterminate potential (CHIP) can promote cardiac fibrosis in HFrEF. Analysis of clinical single nuclei RNA sequencing datasets identified cross-talk between DNMT3A-CHIP circulating monocytes and cardiac fibroblasts. DNMT3A-silenced monocytes activated fibroblasts and promoted myofibroblast differentiation in cell culture models. Mice and patients with DNMT3A CHIP show greater interstitial cardiac fibrosis. Finally, mechanistic studies identified EGF as a potential monocyte-to-fibroblast signaling pathway promoting myofibroblast differentiation and cardiac fibrosis. These novel findings provide evidence that CHIP mutations may promote poor cardiovascular outcomes through more causal mechanisms than the previously identified inflammation pathways.

As presented, the study shows strong evidence to causally connect DNMT3A CHIP with cardiac interstitial fibrosis that is characteristic in HFrEF. EGF signaling in DNMT3A CHIP monocytes may play a role in promoting this fibrosis, which would provide an exciting direct mechanistic connection between CHIP and cardiac myofibroblast activation in cardiac fibrosis. However, there are several issues pertaining to analysis rationale and alternative hypotheses that limit the overall strength of the findings. These are mainly regarding snRNAseq analysis, effects of DNMT3A-KO macrophages, and TGF- β signaling. Specific concerns are detailed below.

We thank the reviewer for the consideration of our manuscript and the insightful comments that have helped us to improve our study.

1. The retrospective single cell analysis compares circulating monocytes with human heart tissue in order to identify potential crosstalk pathways, but it is unclear whether the circulating monocytes can actually affect cardiac cell types. Were monocyte populations found in the human heart tissue snRNAseq dataset that are comparable to the circulating monocytes? Could these two populations overlap with integrated snRNAseq analysis? If so, the crosstalk connections identified here have much more direct significance, and this should be highlighted. If not, the crosstalk ligand-receptor pairs need to be further analyzed and annotated for pathways that can affect cardiac resident fibroblasts from circulating monocytes that are not recruited to the heart tissue. Many ligand-receptor pairs require direct contact or immediate proximity, but if the circulating monocytes are not found in the heart tissue samples then these connections are of questionable relevance. This is an important point.

ANSWER: *Integration of the human circulating monocytes with the human heart tissue revealed that circulating monocytes can be found in cluster 2, 6, 7, 8, 10, 12, 15, 17, 19 and 20, whereas cardiac tissue myeloid cells are detected in cluster 9 (Figure for reviewer 1a, b). However, we could not identify a cluster with monocytes intermingling from circulating and tissue resident cells. This might be due to the fact that circulating monocytes are well known to exhibit a distinct*

expression profile compared to monocyte population in tissues, which are either composed of long-term tissue resident macrophages or bone marrow-derived circulating monocytes, which have homed to the tissue and are subsequently polarized to macrophages. Since macrophages have a different gene expression profile as compared to monocytes, this might explain the assignment to different clusters.

We anticipate that circulating monocytes home to the cardiac tissue and then interact with the tissue-resident fibroblasts. Indeed, the CCR2 [chemokine (C-C motif) receptor 2], which mediates homing of bone marrow-derived circulating monocytes to the heart¹, is abundantly expressed in the circulating monocytes (**Figure for the reviewer 1c, d**). Monocytes from DNMT3A CHIP mutation carriers further show augmented homing factors, such as resistin, which increases monocyte adhesion to endothelial cells (**Figure for the reviewer 1e**)². Upon homing to the heart, secreted factor HB-EGF by monocytes/macrophages and EGFR receptor on fibroblast can contribute to the paracrine signaling cross-talk.

To further support this concept, we performed single nuclei RNA sequencing data of control and DNMT3A mutant mice hearts after myocardial infarction (**Figure for the reviewer 1f**). We found a 1.55-fold increased number of CCR2-positive macrophages ($p=0.09$) and a significantly higher expression of CCR2 (2.44-fold change, $p=0.0075$) in the macrophage cluster of DNMT3A-R882H mice compared to controls (**Figure for reviewer 1g, h**). HB-EGF expressing cells were similarly increased by 7.88-fold ($p=0.09$) (Figure for reviewer 1i, j).

Together, our data support the concept that DNMT3A mutations carrying monocytes show an increased homing to the heart and, thereby, can provide direct paracrine signals. These data are included in the revised manuscript (page 5) as follows:

“Immune cells of the cardiac tissue (named “immune cells”) were characterized by CD163, which is primarily expressed by tissue resident macrophages, whereas the cluster of circulating monocytes (named “monocytes”) showed higher expression of the monocyte markers CD14 and PTPRC, as well as CCR2, which mediates homing to the cardiac tissue²⁴ (Supplementary Figure 2d).”

on page 8 as follows:

“To determine the effect of the DNMT3A^{R882H} mutation on the cardiac tissue on a cellular level, we performed single nuclei RNA sequencing of cardiac tissue. After annotating all major cardiac cell types (Fig. 4g, Supplementary Figure 5a-d), we analyzed the expression of fibrosis-related genes in cardiac fibroblasts (Fig. 4h). Expression of pro-fibrotic and myofibroblast activation genes *Col3a1*, *Postn*, *Pdgfra* and others were upregulated in fibroblasts of DNMT3A^{R882H} mice (Fig. 4h, i, Supplementary Figure 5e). Additionally, we did observe an elevated expression of *Ccr2* and increased numbers of *Ccr2* positive cells in the macrophages subcluster specifically in DNMT3A^{R882H} mice (Fig. 4j, Supplementary Figure 5f-h), suggesting an enrichment of bone marrow-derived macrophages”.

And on page 10 as follows:

“Furthermore, the percentage of *Hbegf* positive cells in the macrophage cluster was increased by 7,87-fold ($p=0.0859$) in DNMT3A^{R882H} mice, whereas *Areg* expression was similar between the groups (Supplementary Fig. 7d-g).”

Figure for the reviewer 1. DNMT3A CHIP monocytes in healthy and diseased heart. a Uniform manifold approximation and projection (UMAP) plots after integration of monocytes from PBMC and cardiac data sets (healthy and HFREF cardiac tissue). **b.** UMAP plot depicting cell clusters. **c, d** Feature plot showing *CCR2* gene

expression in the individual cell clusters (c) and in the annotated cell type clusters (d). e. Violin plot showing *RETN* gene expression in monocytes from No-CHIP and DNMT3A CHIP patients. f. Uniform manifold approximation and projection (UMAP) plot representing different cell type clusters identified after integration WT and DNMT3A^{R883H} hearts. g,h. Percentage of cells (g) and mean expression (h) of *Ccr2* in WT and DNMT3A^{R882H} murine hearts after AMI. i,j. Percentage of cells (i) and mean expression (j) of *Hbegf* in WT and DNMT3A^{R882H} murine hearts after AMI. Data are shown as mean ± SEM. Normal distribution was tested using the Shapiro-Wilk test or the Kolmogorov-Smirnov test. Statistical comparison of two normally distributed groups was performed using unpaired, two-sided Student's t-test; for the data not following Gaussian distribution Wilcoxon-Mann-Whitney test was applied.

2. It is unclear why the snRNAseq datasets of circulating CHIP and non-CHIP monocytes from HFrEF patients were analyzed with the snRNAseq dataset of healthy cardiac tissue and not a snRNAseq dataset from HFrEF patients. The authors should explain this rationale of comparing HFrEF samples to healthy heart samples or perform similar analyses using cardiac snRNAseq datasets from HFrEF patients as well.

ANSWER: We thank the reviewer for this suggestion and integrated our circulating monocyte data with cardiac tissue samples of HFrEF patients. Overall, we observed a similar increase in outgoing interactions between monocytes of DNMT3A CHIP carriers and cardiac cell types in HFrEF tissue (**Figure for the reviewers 2a, b**). Cardiac fibroblasts were identified as a potential interaction partner of circulating monocytes, along with cardiomyocytes, endothelial cells and other cell types of HFrEF patients as well. Although the absolute number of interactions was lower in HFrEF versus healthy heart tissue (see y-axis of panels of the **Figure for the reviewers 2a versus b**, because we only had n=3 versus n=14 samples, respectively), differential number of predicted interactions of monocytes from DNMT3A CHIP carriers with cardiac cells were higher in HFrEF versus healthy tissue (**Figure for the reviewer 2c**).

Importantly, the predicted interactions of circulating monocytes with cardiac fibroblasts included HBEGF-EGFR and AREG-EGFR signaling pairs, which were uniquely upregulated between DNMT3A CHIP monocytes, but not detected between NO CHIP monocytes and cardiac fibroblasts (**Figure for the reviewer 2d, right panel**). These data are comparable to what we showed before for interactions of circulating DNMT3A CHIP monocytes with the fibroblasts of the healthy hearts (**Figure for the reviewers 2d, left panel**). The EGF pathway was among the specifically enriched interactions between monocytes of DNMT3A CHIP-carriers and both, healthy and HFrEF, human cardiac tissues (**Figure for the reviewers 2e, f**). When analyzing the specific interactions with individual cell types, EGF pathway interactions were specifically augmented between DNMT3A CHIP carrier-derived monocytes and fibroblasts and cardiomyocytes of the healthy cardiac tissue (**Figure for the reviewers 2g**). Altogether, these data indicate that HBEGF-EGFR cross-talks may also occur in established heart failure and may thereby aggravate the progression of heart failure.

These data are included in the revised manuscript (page 5) as follows:

"Pooled analysis of cellular interactions by CellChat²¹ demonstrated that monocytes derived from patients carrying DNMT3A CHIP-driver mutations showed highest numbers of outgoing signals between the monocyte cluster to healthy tissue-derived cardiac fibroblasts (Fig. 1e). Additionally, a higher number of interactions was predicted for monocytes derived from DNMT3A CHIP carriers compared to No-CHIP carriers to most cell types (Fig. 1e). When

comparing the outgoing monocyte interactions with the cardiac cells of the HFrEF tissue, we found a generally lower absolute number of interactions, which is explained by the lower number of patients in the HFrEF compared to healthy control samples (Fig. 1e, f). However, the patterns of interactions of DNMT3A CHIP versus No-CHIP with the HFrEF tissue-derived cells were similar (Fig. 1f). Again, DNMT3A CHIP-derived monocytes showed a higher number of predicted interactions compared to No-CHIP carriers (Fig. 1f). Next, we analyzed the specifically outgoing interactions, which were enriched in DNMT3A CHIP compared to No-CHIP-derived monocytes to cardiac cell types. The highest number of DNMT3A CHIP monocyte enriched interactions was observed between the monocytes and fibroblasts of the HFrEF samples (Fig. 1g).”

And on page 8 as follows:

”Next, we assessed the mechanism by which monocytes obtained from DNMT3A CHIP-driver mutation carriers stimulate fibroblast activation. Therefore, we predicted the receptor-ligand interactions between monocytes derived from DNMT3A CHIP-driver mutation carriers versus No-CHIP patients and cardiac tissue by using CellChat. A separate analysis of interactions between circulating monocytes and fibroblasts within the healthy heart revealed significant enrichment of natriuretic peptide receptor 2 (NPR2), epidermal growth factor (EGF), interleukin 1 (IL1), RESISTIN, insulin growth factor (IGF) and other signaling pathways in DNMT3A CHIP monocyte interactions (Fig. 6a). Enriched interactions within HFrEF heart tissue showed partially overlapping, but also distinct pathways including prolactin (PRL), EGF, activin (ACTIVIN), resistin (RESISTIN), and insulin-like growth factor (IGF) signaling between DNMT3A CHIP monocytes and cardiac fibroblasts (Fig. 6b). EGF signaling was defined by enriched ligand-receptor pairing of heparin-binding epidermal growth factor (EGF)-like growth factor (HB-EGF) and amphiregulin (AREG) expressed by monocytes of DNMT3A CHIP-carriers with the receptors EGFR, ERBB2 or ERBB4 in cardiac cell types in healthy and heart failure conditions (Fig. 6d). Moreover, analysis of the specific interactions between monocytes and fibroblasts revealed a specific enrichment of EGF-family member to EGFR cross talks in DNMT3A CHIP carriers (Fig. 6g).”

Figure for the reviewer 2. DNMT3A CHIP monocytes interact with cardiac fibroblasts in healthy and diseased heart **a.** Analysis of the integrated objects for cellular interactions by CellChat. Total number of outgoing paracrine signals from monocytes to all cells of the healthy heart. **b.** Analysis of the integrated objects for cellular interactions by CellChat. Total number of outgoing paracrine signals from monocytes to all cells of the HFrEF tissue. **c.** Differential number of outgoing signals enriched in DNMT3A CHIP-carrier monocytes to the individual clusters of cardiac cells of healthy and HFrEF tissues as indicated. **d.** Bubble plots representing upregulated ligand-receptor pairs in DNMT3A CHIP monocytes-to-fibroblasts signaling in healthy (left) and HFrEF (right) myocardium.

Color encodes communication probability, min. logFC for the interaction depicted is 0.1 and detection in minimum 10% of the cells. **e.** CellChat ligand receptor prediction showing signaling pathways mediating relative information flow (left) and total information flow (right) of DNMT3A CHIP patient-derived monocytes with fibroblasts of healthy cardiac tissue **f.** CellChat ligand receptor prediction showing signaling pathways mediating relative information flow (left) and total information flow (right) of DNMT3A CHIP patient-derived monocytes with fibroblasts of HFrEF cardiac tissue. **g.** Circular plots depicting EGFR signaling from No-CHIP and CHIP monocytes in healthy heart tissue. FB, Fibroblasts. CM, Cardiomyocytes. EC, Endothelial cells. PC, Pericytes. NC, Neurons. SMC, Smooth muscle cells.

3. There is clearly a causal relationship between DNMT3A-CHIP and cardiac interstitial fibrosis. However, the relative contribution of DNMT3A-CHIP monocytes to cardiac fibrosis, as compared to macrophages or inflammatory cytokines known to be increased in DNMT3A-CHIP leukocytes, is not clear from this study. Presumably, macrophages in the injured cardiac tissue could have a stronger effect on fibroblasts, as the macrophage-to-fibroblast communication has been detailed previously. Although there is cross-talk found between circulating monocytes and cardiac resident fibroblasts, are there fibrosis-related pathways between macrophages and fibroblasts? Are these pathways increased in HFrEF rather than the healthy heart? Or, are the pro-fibrotic signals higher from circulating monocytes than cardiac macrophages?

ANSWER: *Due to the low number of immune cells in the cardiac tissue, we were not able to subcluster macrophages from the immune cells cluster, so we performed the comparative analysis using all cardiac immune cells. Analysis of the number of incoming signals from cardiac immune cells to cardiac fibroblasts revealed that immune cells do interact with fibroblasts, this interaction is higher and stronger in HFrEF cardiac tissue in comparison to the healthy tissue (Figure for the reviewers 3a,b). Signaling pathways with highest information flow included fibrosis-related pathways as TGF β , EGF, PDGF, COLLAGEN and IL-6 in resident immune cells- and monocytes-to-fibroblasts signaling from healthy and diseased cardiac tissues (Figure for the reviewers 3c-g). TGF β and EGF pathways were higher abundant in circulating monocytes than in resident immune cells, with EGF signaling only being specifically up-regulated in CHIP carriers (Figure for the reviewers 3c, d). Additionally, EGF signaling was higher in the HFrEF cardiac tissue than in healthy heart (Figure for the reviewers 3d). On the other side, PDGF pathway was mediating resident immune cells-to-fibroblasts signaling in healthy and diseased states (Figure for the reviewers 3e). Interestingly, IL6 signaling was mediating both cardiac immune cells- and monocytes-to-fibroblasts signaling, but was detected only in the healthy cardiac tissue (Figure for the reviewers 3f). In the diseased state (HFrEF) signalling flow was strongly dominated by COLLAGEN pathway and it was detected mainly from resident immune cells.*

Taken together, these data suggest that the circulating monocytes-fibroblast cross-talks might be contributing to fibrosis particularly via TGF β and EGF signaling, whereas PDGF and COLLAGEN signaling are predicted to dominate the interaction between tissue-resident macrophages and fibroblasts.

4. From the cardiosphere experiment, it seems possible that DNMT3A-silenced monocytes directly impair cardiac contraction. Is there evidence of this crosstalk within the snRNAseq CellChat analysis? Additionally, it would be beneficial to explain the cellular components of cardiospheres in the results paragraph introducing this experiment. It was unclear that cardiospheres contain fibroblasts.

ANSWER: We thank the reviewer for this suggestion. Indeed, CellChat analysis revealed a strong interaction of CHIP monocytes with cardiomyocytes via the EGF pathway as well (**Figure for the reviewer 4a**). Since cardiomyocytes did express high levels of one of the EGF receptors, ERBB4,

but not EGFR, ERBB2, CD4, in healthy and HFrEF tissues (**Figure for the reviewer 4b, c**), we hypothesized that HBEGF might affect cardiomyocytes. Therefore, we analyzed the effects of HB-EGF on cardiomyocytes contractility using human heart slices³. As shown in **Figure for the reviewer 4d**, HB-EGF treatment significantly reduced contractility of the heart slices. To determine whether these effects might be mediated by a direct cardiomyocyte toxicity, we additionally evaluated the number of cardiomyocytes after stimulation with supernatants from siCtrl and siDNMT3A activated monocytes (**Figure for the reviewer 4e**). However, we did not find a change in cardiomyocyte numbers in response to conditioned medium of DNMT3A-silenced monocytes. Therefore, we conclude that HB-EGF might induce a reduction of cardiomyocyte contractility. These data are shown in the revised manuscript on pages 7, 9-10.

“Since we observed an effect on contractility by DNMT3A silenced monocyte supernatants, and CHIP monocytes also are predicted to interact with cardiomyocytes, we additionally tested the effect of the conditioned media on cardiomyocytes, using a recently established human ventricular cardiomyocyte cell line³³. However, we did not find cytotoxic effects in this setting (Supplementary Fig. 3d).”

“Of the EGF receptor family members, the EGF receptor was highest expressed in cardiac fibroblasts, both in the healthy and HFrEF hearts (Fig. 6i, Supplementary Fig. 7b-c).”

“HB-EGF treatment additionally reduced the beating frequency of cardiospheres without altering their size (Fig. 7 f, j). The effect on contractility was further assessed using human heart slices⁴³ showing a decline in contractility after HB-EGF treatment (Supplementary Fig. 9a, b).”

Additionally, we provide more details regarding the cellular components of cardiospheres in the revised manuscript as follows:

“To gain further insights in how monocytes derived from patients harboring DNMT3A CHIP-driver mutations affect fibroblasts and other cardiac cells in a more physiological environment, we used 3D cardiac tissue mimetics (cardiospheres)³², which comprise cardiomyocytes, fibroblasts and endothelial cells (Fig. 3a).”

“To gain insights into the functional consequences of the increased HB-EGF expression, we tested the effects of recombinant HB-EGF on cardiac fibroblast monocultures and cardiospheres containing cardiac fibroblasts.”

Figure for the reviewer 4. DNMT3A CHIP monocytes interact with cardiomyocytes in healthy and diseased heart **a**. Circular plots depicting EGFR signaling from No-CHIP and CHIP monocytes in healthy heart tissue. **b, c**. Violin plots depicting *ERBB4* and *ERBB2* gene expression in all the different cardiac cell types **b/c**. Expression of respective genes in single nuclei RNA sequencing from healthy (b) and HFrEF (c) tissue. **d** Contractility of human heart slices after treatment with HB-EGF (100 ng/ml, every 2. day) over 9 days. Statistical comparison was performed using two-way ANOVA with Fisher's LSD multiple comparisons post-test (n=3 per group). **e** Analysis of cardiomyocyte cell area treated with DNMT3A-silenced THP-1 monocytes supernatants (n=3). Data are shown as mean \pm SEM. Normal distribution was tested with the Shapiro-Wilk test or the Kolmogorov-Smirnov test. Statistical comparison, unless stated otherwise, of two normally distributed groups was performed using either one-sample t-test or unpaired, two-sided Student's t-test.

5. Monocyte-mediated TGF- β signaling is dismissed and EGF signaling is investigated instead. What is the DNMT3A-silenced monocyte contribution to TGF- β signaling in HFrEF, and how does this affect fibroblast activation relative to EGF signaling?

ANSWER: We apologize for dismissing to further analyze and discuss the implication of TGF β . Although TGF β was among the highest predicted interactions between monocytes and various other cell types of the heart in the original manuscript, we did not find TGF β in the specifically enriched interactions between monocytes of DNMT3A CHIP carriers versus controls in our revised analysis (**Figure for the reviewer 5a, b**). Additional analysis of TGF β pathways further revealed no difference between No-CHIP and DNMT3A CHIP carriers (**Figure for reviewer 5c**). In contrast, EGF showed low interaction number in NO CHIP monocytes, but was up-regulated in DNMT3A CHIP monocyte interactions to both fibroblasts and cardiomyocytes (**Figure for reviewer 5d**). Therefore, we focused our studies on the investigation of the EGF-axis in mediating DNMT3A CHIP monocytes to fibroblast and cardiomyocyte interactions.

CHIP patient-derived monocytes with fibroblasts of healthy cardiac tissue. Depicted in blue – significant interactions. **b** CellChat ligand receptor prediction showing signaling pathways mediating relative information flow of DNMT3A CHIP patient-derived monocytes with fibroblasts of HFrEF cardiac tissue. Depicted in blue – significant interactions **c** Circular plots depicting TGFB signaling from No-CHIP and CHIP monocytes in healthy heart tissue. **d** Circular plots depicting EGFR signaling from No-CHIP and CHIP monocytes in healthy heart tissue.

Response to Reviewer #2:

The study suggests a role for Dnmt3a clonal hematopoiesis driving mutations in the development of cardiac fibrosis. The authors use bioinformatic analysis of published data to provide a rationale for focusing on monocyte:fibroblast interactions. They show that in vitro dnmt3a CHIP mutations activate a fibrogenic profile in fibroblasts. In vivo, Dnmt3 mutation in hematopoietic cells accentuates interstitial fibrosis following MI. The effects are attributed to activation of a monocyte-driven EGFR axis.

General comment:

The study has substantive novel content, extending our understanding on a hot topic; the potential involvement of CHIP in cardiac fibrosis. Concerns regarding a) the selective effects on the remote non-infarcted myocardium, b) better documentation of the EGF ligand expression in vivo and c) the quality of the infarction experiments need to be addressed.

We thank the reviewer for the positive remarks. By addressing the specific comments, we believe that we have improved the manuscript.

Major comments:

1. The infarction experiments (Figure 3) raise concerns. The representative images show no infarction in the NO CHIP group and a tiny segment of transmural fibrosis in the CHIP group. The scar size quantitation's also suggest very small infarcts. Please show data consistent with a LAD territory infarct. This is important because the authors suggest no effects of the Dnmt3 mutation on infarct healing.

ANSWER: *We thank the reviewer for pointing out this important aspect and apologize for our poor selection of examples shown. Overall, scar size varies between 2% and 16% of total heart section for WT mice (recipients transplanted with WT cells after) and 7% and 15% for DNMT3A-R882H mice (recipients transplanted with DNMT3A-R882H cells) in our original analysis. To confirm our findings, we repeated the analysis with additional sections, which were obtained from the same mice but towards base. This analysis revealed similar scar sizes of 3% to 10% and 4% to 7% in WT compared to DNMT3A-R882H mice. Consistent with the first analysis, we did not detect significant differences in the infarct size between the group, but again observed an increase in interstitial fibrosis in remote septum, which is in line with the analysis in the initial submission (**Figure for reviewer 6a, b**). Moreover, we did not find differences in troponin T levels between DNMT3A-R882H versus control mice (**Figure for the reviewer 6c**). Consistently, ejection fraction was similar between groups at day 0 and day 7 after acute myocardial infarction and only deteriorated at later time point (**Figure for reviewer 6d**). Both analyses indicate that the initial injury does not differ between the two groups, but that the subsequent healing and remodeling response is altered. We included this new analysis in the revised manuscript and provide representative images in the figures of the revised manuscript. These data are discussed on page 7.*

“Similar cardiac troponin T levels 24 hours post LAD-ligation indicate comparable infarct sizes in wild-type mice carrying a human DNMT3A^{R882H} mutation (DNMT3A^{R882H}) in the bone marrow-derived hematopoietic and the control group (WT) (Fig.4b). Histopathological analysis of cardiac tissue after four weeks showed an increase in cardiac interstitial fibrosis in the remote zone of myocardial tissue DNMT3A^{R882H} mice in comparison WT mice, whereas the infarct size was not changed (Fig. 4c-f).”

Figure for the reviewer 6. DNMT3A CHIP promotes diffuse cardiac fibrosis in mice with DNMT3A^{R882H} bone marrow cells **a, b.** Picosirius red staining of murine cardiac cross sections **a.** Infarct fibrotic scar in WT and DNMT3A^{R882H} mice **b.** Remote zone in WT and DNMT3A^{R882H} mice **c** ELISA-based quantification of cardiac troponin in serum of wild-type mice transplanted with wild-type bone marrow cells (WT) and wild-type mice transplanted with DNMT3A^{R882H} bone marrow cells (DNMT3A^{R882H}) after AMI (n=5). **d** Left ventricle ejection fraction determined with echocardiography of WT and DNMT3A^{R882H} on day 0 and day 28 after AMI. Data are shown as mean \pm SEM. Normal distribution was tested with the Shapiro-Wilk test or the Kolmogorov-Smirnov test. Statistical comparison of two normally distributed groups was performed using unpaired, two-sided Student's t-test.

2. Sham experiments should be performed to exclude effects of transfer of mutant hematopoietic cells in the absence of injury.

ANSWER: We thank reviewer for pointing out this item. A sham experiment was not carried out parallel to the AMI experiments, since we have to reduce the number of mice to a minimum according to the ethical committee. However, to address the question of the reviewer, we analyzed untreated WT and DNMT3A R882H mice and assessed diffuse fibrosis in the septum in the absence of injury. Briefly, pl:pC-treated unfractionated BM cells from MxCre+:DNMT3A R882H (+/m) and wild-type MxCre-:Dnmt3a +/+ (+/+) mice were transplanted into young, 15 weeks old WT recipients. Under basal conditions without myocardial infarction, we found a 1.42-fold increase in fibrosis, which however, was not statistically significant (**Figure for reviewers 7a, b**).

These data are included and discussed in the revised manuscript (page 7) as follows:

“Cardiac fibrosis was also increased by 1.42-fold in DNMT3A^{R882H} carrying mice at baseline, but this increase was not statistically significant (Supplementary Fig. 4a).”

Figure for the reviewer 7. DNMT3A CHIP has no effect on diffuse cardiac fibrosis in non-injured mice with DNMT3A^{R882H} bone marrow cells. **a** Representative brightfield images of diffuse interstitial cardiac fibrosis in the septal zone of WT and DNMT3A^{R882H} mice. **b** Quantification of the diffuse cardiac fibrosis (n=3). Data are shown as mean ± SEM. Normal distribution was tested with the Shapiro-Wilk test or the Kolmogorov-Smirnov test. Statistical comparison of two normally distributed groups was performed using unpaired, two-sided Student's t-test.

3. The authors suggest that the effects of the mutation are limited to the remote myocardium. What is the basis for this selective effect? Monocytes infiltrate the infarct and activate reparative myofibroblasts. The selectivity for the remote zone is impossible to justify. Could the effects on interstitial fibrosis reflect perturbations in the healing infarct that the authors did not detect? (because of their limited analysis of the composition of the infarct). Such perturbations may affect filling pressures, thus causing secondary increases in interstitial fibrosis in the non-infarcted myocardium.

ANSWER: *Fibrosis was indeed mainly affected in the remote zone in the experimental mouse models and in the patient study. Our primary hypothesis is that DNMT3A R882H monocytes drive exaggerated infiltration of monocytes, also in the remote zone of the infarcted myocardium, which contributes to excessive fibrosis. Indeed, we have seen increased numbers of CD68+ macrophages in the remote zone of DNMT3A R882H mice subjected to AMI but not in control mice (Figure for reviewer 8a, b). Our data suggest that the release of HB-EGF and inflammatory cytokines by the DNMT3A CHIP cells, which home to the heart, may further promote fibroblast activation.*

We have included the additional data (page 7-8, page 12) and have discussed the potential mechanistic explanation in the revised manuscript as follows:

“Mutant DNMT3A^{R882H} carrying mice additionally showed an increase in CD68 positive inflammatory cells in the remote zone after infarction, whereas no change was detected in non-injured mice at baseline (Supplementary Fig. 4b, c). These data suggest that mimicking DNMT3A^{R882H} in mice augments the number of inflammatory cells in the remote zone and induces diffuse interstitial cardiac fibrosis particularly in response to injury.”

“The activation of cardiac fibroblasts may lead to the instigation and further progression of diffuse cardiac fibrosis in patients with heart failure carrying DNMT3A CHIP-driver mutations. Since the EGF receptor is expressed in fibroblasts of the healthy hearts as well, our bioinformatics analysis predicts that an interaction would also occur in uninjured hearts. However, healthy DNMT3A^{R882H} mice did only show a trend, but no significant increase in cardiac fibrosis. As infiltration of circulating monocytes is augmented by endothelial activation and tissue inflammation, the additional infarction-induced myocardial damage may amplify the recruitment of bone marrow derived cells into the heart. We indeed observed an increase in *Ccr2* expression in the macrophage cluster, which is indicative of recruited bone marrow-derived macrophages²⁴, and further detected more CD68 positive immune cells in the remote zone of the infarcted DNMT3A^{R882H} mutant mice. Since the release of inflammatory mediators can activate cardiac fibroblasts, the recruited monocytes may secondarily drive fibrosis not only via HB-EGF, but also by releasing interleukins and other cytokines well-known to activate fibroblasts⁴⁷. Likely, DNMT3A CHIP mutations amplify heart failure by pleiotropic effects on inflammation and fibrosis.”

4. Some evidence supporting the proposed EGFR-mediated mechanism in vivo would strengthen the manuscript. Perhaps, the authors could show increased levels of EGFR ligands in myeloid cells in the remodeling infarcted heart.

ANSWER: To address the question of the reviewer, we performed snRNA-Seq analysis of the control and DNMT3A mutant mice hearts after myocardial infarction (**Figure for reviewer 9a-c**). We indeed found a 7.88-fold increase in *Hbegf* expressing cells ($p=0.0859$) and 3.43-fold increased expression ($p=0.21$) in cardiac immune cells in DNMT3A CHIP carriers but not in controls, whereas the expression of *Areg* (*Ar*) was not augmented (**Figure for reviewer 9d-f**). The analysis is limited by the relative low expression levels of *Hbegf* and *Ar* (high number of drop outs) and the low numbers of cells (WT-1 n=63, WT-2 n=31, WT-3 n=85, DNMT3A-1 n=124, DNMT3A-2 n=91, DNMT3A-3 n=143 cells). We however feel that the data support our hypothesis and added these data in the revised manuscript (page 8-9) as follows:

“To determine the effect of the DNMT3A^{R882H} mutation on the cardiac tissue on a cellular level, we performed single nuclei RNA sequencing of cardiac tissue. After annotating all major cardiac cell types (Fig. 4g, Supplementary Figure 5a-d), we analyzed the expression of fibrosis-related genes in cardiac fibroblasts (Fig. 4h). Expression of pro-fibrotic and myofibroblast activation genes *Col3a1*, *Postn*, *Pdgfra* and others were upregulated in fibroblasts of DNMT3A^{R882H} mice (Fig. 4h, i, Supplementary Figure 5e). Additionally, we did observe an elevated expression of *Ccr2* and increased numbers of *Ccr2* positive cells in the macrophages subcluster specifically in DNMT3A^{R882H} mice (Fig. 4j, Supplementary Figure 5f-h), suggesting an enrichment of bone marrow-derived macrophages.”

“Furthermore, the percentage of *Hbegf* positive cells in the macrophage cluster was increased by 7,87-fold ($p=0.0859$) in *DNMT3A^{R882H}* mice, whereas *Areg* expression was similar between the groups (Supplementary Fig. 7d-g).

Since EGFR signaling is a well-known mediator of cardiac fibroblast activation^{40,41}, these data suggest that monocytes obtained from *DNMT3A* CHIP-driver mutation carriers may release factors of the EGF family to induce cardiac myofibroblast activation, which express high levels of EGFR in the human heart.”

Figure for the reviewer 9. EGF signaling in mice with *DNMT3A^{R882H}* bone marrow cells after AMI. a-c Representative uniform manifold approximation and projection (UMAP) plots of scRNA-seq sequenced WT and *DNMT3A^{R882H}* murine hearts after AMI (day 75) showing **a** individual samples **b** clusters **c** different cell clusters identified after annotation. CM, Cardiomyocytes; EC, Endothelial cells; FB Fibroblasts; PC, Pericytes **d** Violin plot representing *Hbegf* and *Ar* (*Areg*) gene expression in the immune cell cluster of hearts derived from WT and *DNMT3A^{R882H}* mice after AMI. **e** Percentage of cells expressing *Hbegf* and *Hbegf* mean expression in the immune cell cluster of hearts from WT and *DNMT3A^{R882H}* mice after AMI. **f** Percentage of cells expressing *Ar* and *Ar* mean

expression in immune cell clusters in hearts of WT and DNMT3A^{R882H} mice after AMI. Data are shown as mean \pm SEM. Normal distribution was tested with the Shapiro-Wilk test or the Kolmogorov-Smirnov test. Statistical comparison of two normally distributed groups was performed using unpaired, two-sided Student's t-test.

5. Do THP1 monocytes recapitulate the cardiac macrophage population (resident and recruited)? Considering the heterogeneity of macrophages, this has to be better justified.

ANSWER: Although THP-1 cells are a well-established model and have been widely used to assess leukemia/clonal hematopoiesis in top tier journals^{2,4}, we agree with the reviewer, that there are limitations of this cell model. Therefore, we confirmed our key findings in mice and humans; e.g. we demonstrate that silencing of DNMT3A leads to a consistent regulation of genes as observed in DNMT3A R882H carrying mice and DNMT3A CHIP mutations carrying humans. The functional effects on fibrosis have further been validated in vivo. Therefore, we believe that the THP-1 cells may be a useful model to gain first insights into the effects of CHIP mutant genes.

6. Figure 1: Was matrix synthesis and/or metabolism modulated in fibroblasts stimulated with the supernatant of the Dnmt3a cells?

ANSWER: We thank the reviewer for this idea and detected the expression of matrix proteins in fibroblasts after treatment with DNMT3A-silenced monocyte supernatants. Indeed, we found a significant increase of COL1A1, COL3A1 and VIM (**Figure for reviewer 10**).

These data are included in the revised manuscript (page 6):

"Treatment with the conditioned medium of DNMT3A-silenced monocytes further augmented collagen expression (Supplementary Fig. 3a)."

Figure for the reviewer 10. Matrix synthesis in human cardiac fibroblasts (HCF) after incubation with supernatants of DNMT3A-silenced activated monocytes. a Relative *COL1A1* (a), *COL3A1* (b), *VIM* (c) gene expression normalized to *RPLPO* mRNA expression in human cardiac fibroblasts (HCF) after incubation with supernatants from PMA-activated THP-1 cells for 48 hours after siRNA silencing of *DNMT3A* relative or negative control (n=3). Data are shown as mean ± SEM. Normal distribution was tested with the Shapiro-Wilk test or the Kolmogorov-Smirnov test. All samples are non-normally distributed. P value was calculated by one sample t-test.

Minor:

Considering that the samples are not from the same subjects, the bioinformatic support for the monocyte:fibroblast seems artificial. The authors are well-justified to hypothesize the role of monocyte/fibroblast interactions, on the basis of the very extensive evidence supporting roles of monocyte/macrophages in modulation of fibroblast phenotype.

Answer: *We thank the reviewer for acknowledging our effort. We have additionally raised the limitations of our bioinformatic analysis in the discussion section of our revised manuscript as follows (page 13):*

“Limitations: Although our study provides novel insights into the impact of DNMT3A CHIP on cardiac pathologies in mice and humans, the causal contributions of the HB-EGF-EGF receptor interactions to cardiac fibrosis has only been documented by using in vitro models and cardiac tissue mimetics. Further studies have to address whether inhibition of the EGF pathway reduces DNMT3A CHIP induced cardiac fibrosis in vivo.

Second, our *in silico* prediction analysis is limited to the investigation of circulating monocytes with human cardiac tissue. This interaction would not directly occur, since monocytes only can interact with the cardiac fibroblast and cardiomyocytes after entering the heart. The analysis of cardiac tissue biopsies from DNMT3A CHIP mutant carriers would provide a better model. However, such an analysis is limited by the low number of immune cells in biopsies and difficulties in discriminating between the mutant and wild type cells. A second limitation of our *in silico* data is the combination of single cell and single nuclei RNA sequencing data. However, since the No-CHIP and DNMT3A CHIP data show similar quality, a comparative analysis of the interactions of the two groups with the cardiac tissue-derived cells appears valid. We also confirmed the predicted findings with different cell culture models, tissue mimetics and mutant mice in vivo.

Taken together, our data, which suggest an augmented fibroblast activation and diffuse fibrosis in DNMT3A CHIP-driver mutation carriers, may also have therapeutic implications. Assessment of DNMT3A mutation may identify patients at high risk for negative remodeling and cardiac fibrosis allowing for a targeted treatment with new anti-fibrotic regimens or strategies that specifically interfere with the proposed downstream pathways.”

Please briefly explain the rationale for the use of the Mx-Cre mice and the protocol for Cre activation in these animals. Also considering the hypothesis (which involved monocytes), why not use a myeloid cell Cre to generate cells for transfer?

Answer: *In order to decipher the role of DNMT3A R882H mutation in heart failure in general (other manuscript in preparation by our collaborators), we concentrated our analysis on Cre-*

inducible system in the hematopoietic stem cells. Since DNMT3A R882H mutations are not only affecting monocytes but the entire spectrum of immune cells, we used an ubiquitous interferon-inducible Mx1-Cre intercross, which is still the most commonly used “deleter strain” in experimental hematology⁵. For all experiments, primary WT and DNMT3A R882H mouse hematopoietic cells were harvested at least 14 days - one month after the last pl:pC injection (400µg/mouse intraperitoneally for 3 nonconsecutive days) to ensure that signaling activation and cytotoxic effects mediated by pl:pC were minimized. In other experiments, we used tamoxifen-inducible Rosa26Cre ERT2 :DNMT3A-R882H (Cre recombinase - estrogen receptor T2; known as R26-Cre ERT2 strain) intercross mice as donors. These mice do not require pl:pC-treatment and are independent of intrinsic IFN-response to induce DNMT3A-R882H expression within the hematopoietic compartment⁶. After inducing Cre expression, we perform bone marrow transplantation to ensure that only the bone marrow and bone marrow derived cells harbor the CHIP inducing mutation. This recapitulates the essential features of human CHIP.

Response to Reviewer #3:

The manuscript from Shumliakivska et al investigate how DNMT3A clonal hematopoiesis-driver mutations can affect cardiac fibroblast activation, and identify monocyte HB-EGF and fibroblast EGFR as a potential axis regulating the increased fibroblast activation associated with DNMT3A CHIP or DNMT3A silencing. The paper is well written, and I enjoyed reading it. I do think that the manuscript needs to be strengthened in several parts, and I provide here below a series of comments that I hope can help the authors.

- The authors should definitely strengthen the first connection between DNMT3A CHIP and the cardiac fibroblasts at the beginning of the paper (Figure 1a-c), as then all the story develops from this premise. The authors show in Figure 1c that among the cardiac cell types, the DNMT3A circulating monocytes have the strongest crosstalk connection with the fibroblasts. But is this compared to the level of how NO-CHIP monocytes crosstalk to the cardiac fibroblasts? I think this is a critical point that the authors should explain better. How does Figure 1c look when assessing the cellular crossalk connection between NO-CHIP monocytes and the other cardiac cells?

ANSWER: *We thank the reviewer for this constructive critique and have improved the description of our findings. We have demonstrated No-CHIP and CHIP monocytes crosstalk to the cardiac fibroblasts and other cell types in healthy cardiac tissue and HFrEF tissue in the new integrated object (Figure for the reviewer 11a, b). We additionally have added an analysis of the interactions of the circulating DNMT3A CHIP monocytes with cardiac tissue of HFrEF patients to our study. The improved and new data are shown in Figure for the reviewer 11c-f below and are described in the revised manuscript as follows (page 5):*

“Pooled analysis of cellular interactions by CellChat²¹ demonstrated that monocytes derived from patients carrying DNMT3A CHIP-driver mutations showed highest numbers of outgoing signals between the monocyte cluster to healthy tissue-derived cardiac fibroblasts (Fig. 1e). Additionally, a higher number of interactions was predicted for monocytes derived from DNMT3A CHIP carriers compared to No-CHIP carriers to most cell types (Fig. 1e). When comparing the outgoing monocyte interactions with the cardiac cells of the HFrEF tissue, we found a generally lower absolute number of interactions, which is explained by the lower number of patients in the HFrEF compared to healthy control samples (Fig. 1e, f). However, the patterns of interactions of DNMT3A CHIP versus No-CHIP with the HFrEF tissue-derived cells were similar (Fig. 1f). Again, DNMT3A CHIP-derived monocytes showed a higher number of predicted interactions compared to No-CHIP carriers (Fig. 1f). Next, we analyzed the specifically outgoing interactions, which were enriched in DNMT3A CHIP compared to No-CHIP-derived monocytes to cardiac cell types. The highest number of DNMT3A CHIP monocyte enriched interactions was observed between the monocytes and fibroblasts of the HFrEF samples (Fig. 1g).”

And on page 8 as follows:

“Next, we assessed the mechanism by which monocytes obtained from DNMT3A CHIP-driver mutation carriers stimulate fibroblast activation. Therefore, we predicted the receptor-ligand interactions between monocytes derived from DNMT3A CHIP-driver mutation carriers versus No-CHIP patients and cardiac tissue by using CellChat. A separate analysis of interactions between circulating monocytes and fibroblasts within the healthy heart revealed significant enrichment of natriuretic peptide receptor 2 (NPR2), epidermal growth factor (EGF), interleukin 1 (IL1), RESISTIN, insulin growth factor (IGF) and other signaling pathways in DNMT3A CHIP monocyte interactions (Fig. 6a). Enriched interactions within HFrEF heart tissue showed partially overlapping, but also distinct pathways including prolactin (PRL), EGF, activin (ACTIVIN), resistin (RESISTIN), and insulin-like growth factor (IGF) signaling between DNMT3A CHIP monocytes and cardiac fibroblasts (Fig. 6b). EGF signaling was defined by enriched ligand-receptor pairing of heparin-binding epidermal growth factor (EGF)-like growth factor (HB-EGF) and amphiregulin (AREG) expressed by monocytes of DNMT3A CHIP-carriers with the receptors EGFR, ERBB2 or ERBB4 in cardiac cell types in healthy and heart failure conditions (Fig. 6d).”

Figure for the reviewer 11. DNMT3A CHIP monocytes interact with cardiac fibroblasts in healthy and diseased heart **a.** Analysis of the integrated objects for cellular interactions by CellChat. Total number of outgoing paracrine signals from monocytes to all cells of the healthy heart is shown. **b.** Analysis of the integrated objects for cellular interactions by CellChat. Total number of outgoing paracrine signals from monocytes to all cells of the HFrEF tissue. **c.** Differential number of outgoing signals enriched in DNMT3A CHIP-carrier monocytes to the individual clusters of cardiac cells of healthy and HFrEF tissues as indicated. **d.** Bubble plots representing upregulated ligand-receptor pairs in DNMT3A CHIP monocytes-to-fibroblasts signaling in healthy (left) and HFrEF (right) myocardium. Color encodes communication probability, min. logFC for the interaction depicted is 0.1 and detection in minimum 10% of the cells. **e.** CellChat ligand receptor prediction showing signaling pathways mediating relative information flow (left) and total information flow (right) of DNMT3A CHIP patient-derived monocytes with fibroblasts of healthy cardiac tissue **f.** CellChat ligand receptor prediction showing signaling pathways mediating relative information flow (left) and total information flow (right) of DNMT3A CHIP patient-derived monocytes with fibroblasts of HFrEF cardiac tissue. FB, Fibroblasts. CM, Cardiomyocytes. EC, Endothelial cells. PC, Pericytes. NC, Neurons. SMC, Smooth muscle cells.

- Given how heterogenous is the cardiac fibroblast population, especially in disease, I wonder if the authors could go a bit more in depth in understanding which fibroblast cell states might interacting more with CHIP monocytes. After using the data from Litviňuková at al, it would be interesting to leverage a few of the many single cell transcriptional data that are publicly available and centered on the cardiac fibroblasts, to understand which fibroblast clusters might be more involved in a paracrine interaction with NO CHIP versus DNMT3A CHIP monocytes. Some examples could be PMID: 32130914 and PMID: 30912746. The exact same analysis in Figure 1c could be performed using the NO CHIP and the DNMT3A CHIP monocytes data from the author's previous work PMID: 33155517, but this time plotting the connections with the different fibroblast states/clusters.

ANSWER: We thank the reviewer for this very helpful suggestion. We have used our new Seurat object, which comprises cardiac tissue from healthy controls⁷ and our own three HFrEF patients and have subclustered the fibroblast populations. We found 5 fibroblast populations (**Figure for the reviewer 12a-d**). The FB2 fibroblast subcluster was preferentially detected in HFrEF samples. Analysis of the interactions between the monocytes cluster and the 5 fibroblast subclusters revealed interactions of the monocytes with 4 out of the five cluster. HBEGF/AREG-EGFR interactions were detected between monocytes and all four clusters, showing a slightly higher communication probability for clusters FB2 and FB3.

These findings are described in the revised manuscript as follows (page 12):

"Cardiac fibroblasts comprise a heterogeneous population of cells, particularly after myocardial infarction^{30,45,46}. The observed interaction does not appear to be specific to one of the subpopulations, as most fibroblast clusters in the human heart samples showed an interaction with the DNMT3A CHIP-carrying monocytes (Supplementary Fig. 10). The activation of cardiac fibroblasts may lead to the instigation and further progression of diffuse cardiac fibrosis in patients with heart failure carrying DNMT3A CHIP-driver mutations. Since the EGF receptor is expressed in fibroblasts of the healthy hearts as well, our bioinformatics analysis predicts that an interaction would also occur in uninjured hearts. However, healthy DNMT3A^{R882H} mice did only show a trend, but no significant increase in cardiac fibrosis. As infiltration of circulating monocytes is augmented by endothelial activation and tissue inflammation, the additional

infarction-induced myocardial damage may amplify the recruitment of bone marrow derived cells into the heart. We indeed observed an increase in Ccr2 expression in the macrophage cluster, which is indicative of recruited bone marrow-derived macrophages²⁴, and further detected more CD68 positive immune cells in the remote zone of the infarcted DNMT3A^{R882H} mutant mice. Since the release of inflammatory mediators can activate cardiac fibroblasts, the recruited monocytes may secondarily drive fibrosis not only via HB-EGF, but also by releasing interleukins and other cytokines well-known to activate fibroblasts⁴⁷. Likely, DNMT3A CHIP mutations amplify heart failure by pleiotropic effects on inflammation and fibrosis.”

Figure for the reviewer 12. Heterogeneity of human cardiac fibroblasts in the integrated object

a-d. Uniform manifold approximation and projection (UMAP) plots showing integration of DNMT3A CHIP and No-CHIP monocytes and subclusters of cardiac fibroblasts from healthy and HFrEF hearts **a.** Represents the different origin of the cells **b.** Different cellular clusters **c.** Cell annotation **d.** Subcluster annotation **e.** Bubble plots representing CHIP-upregulated ligand-receptor pairs in monocytes-to-fibroblasts subclusters. Color encodes

communication probability, min. logFC for the interaction depicted is 0.1 and detection in minimum 10% of the cells.

- The quality of the collagen contraction gel images in Figure 1j is really not that good, hence very difficult to see differences between the 2 conditions. Also, I suggest the authors not only to measure at 24hr, but also at later time points as usually in this assay the gel contraction is more evident at later time points.

ANSWER: We have addressed the reviewers concern and provide the requested data (as shown in Figure for the reviewer 13a, b below and on page 6 of the revised manuscript).

Figure for the reviewer 13. a,b Collagen gel contraction analysis in stimulated HCF (n=3) after treatment with supernatants. All data are shown as mean \pm SEM. Normal distribution was tested with Shapiro-Wilk test or Kolmogorov-Smirnov test. All groups are normally distributed. P values were calculated with unpaired, two-sided Student's t-test.

- It would help the readers if a schematic figure is added (similar to Fig 1f) to summarize how the experiment related to Figure 5d-i is performed.

ANSWER: We have addressed this constructive comment and added a new schematic figure in the revised manuscript (Figure for the reviewer 14 and Figure 6 in the manuscript).

Figure for the reviewer 14. Schematic representation of the experimental design used to study monocytes-driven HBEGF-EGFR signaling .

- While the experiments with the EGFR kinase inhibitor Gefitinib strengthen the hypothesis the EGFR signaling pathway is involved in the increased fibroblast activation observed in the conditioned medium experiments with DNMT3 silencing in the monocytes, these don't really prove to which extent HB-EGF specifically is involved. Can the experiments shown in Figure 5d-I be repeated by incubating the conditioned medium with a HB-EGF neutralizing antibody prior moving the medium to the cardiac fibroblasts and the cardiospheres? These experiments would more specifically indicate the role of HB-EGF. Please see PMID: 23251664 or PMID: 20332144, just to highlight a few works where anti HB-EGF antibodies have been used.

ANSWER: We thank the reviewer for this very helpful advice and addressed the reviewers concern as suggested. Indeed, neutralizing anti-HB-EGF antibodies prevented the paracrine activation of cardiac fibroblasts by conditioned medium from DNMT3A silenced monocytes (**Figure for reviewer 15a**).

These data have been added to the revised manuscript (page 11) as follows:

“Finally, blocking HB-EGF by neutralizing antibodies reduced paracrine activation of fibroblasts by DNMT3A-silenced monocytes (Fig. 8i, j). Taken together, these results demonstrate that the secretome of DNMT3A silenced monocytes induces fibroblasts activation partially through EGFR signaling.”

Figure for the reviewer 15. Immunofluorescence analysis of α SMA protein expression in HCF stimulated with supernatants from DNMT3A-silenced THP1-cells with or without HB-EGF neutralizing antibody (50 μ g/ml, 48 h) (n=4). Data are shown as mean \pm SEM. Normal distribution was tested with the Shapiro-Wilk test or the Kolmogorov-Smirnov test. All samples are normally distributed. P value was calculated by one-way ANOVA test followed by Dunnett's multiple comparison test.

- Why is cardiac function not shown for the experiment described in Figure 3a? Is the transplantation with the Ly5.2 D3A+/m worsening the function (EF or FS) as compared to Ly5.2 WT? Even if not the case, this is important to know.

ANSWER: Analysis of the cardiac function by echocardiography 4 weeks following surgery displayed impaired left ventricular ejection fraction in DNMT3A R882H chimeric recipients compared to controls in the enlarged study cohort from different investigation (N=3) (**Figure for**

the reviewer 16a, see below). Cardiac function in the mice of which tissue was used in our study cohort (N=5 mice pro group) revealed a consistent decrease of ejection fraction after 4 weeks too (**Figure for the reviewer 16b**). However, troponin levels 24 h after infarct-induction in our mice cohort were not different between the groups (**Figure for the reviewer 16c**). Cardiac function is not shown in this manuscript as the data are not approved for public use by our collaborators.

- The experiment described in Figure 3a-c demonstrates that the DNMT3A CHIP increases interstitial fibrosis in an MI model, but the extent by which HB-EGF is involved is not clear. If the authors do find a neutralizing HB-EGF antibody that works in their hands, they should repeat the BMC transplantation experiments, and treat the mice with an anti HB-EGF antibody (this could be done over the period of the reconstitution or/and after MI), and see whether this decreases the fibrosis upon MI. If this is not possible (good antibodies not available or not effective in vivo in mouse), why not using an EGFR inhibitor that can neutralize mouse EGFR in vivo? I think these experiments would provide an in vivo validation that HB-EGF or at least EGFR are strongly involved in the DNMT3A CHIP cardiac phenotype, hence really strengthen the study.

ANSWER: Due to increased demands regarding the approval of animal experiments (it takes > 3 months to get approval) and the shutdown of the colony, we unfortunately could not perform the requested experiments. However, we performed further in vitro experiments to provide more evidence for the causal involvement of the proposed model. Specifically, we show that the neutralizing HB-EGF antibody also reduced fibroblast activation induced by DNMT3A silenced monocytes (**Figure for the reviewer 17a**). We also performed single nuclei RNA sequencing

analysis to confirm the regulated pathways involving *Hbegf* in vivo (**Figure for the reviewer 17b-c**). Additionally, we have demonstrated that we have increased number of homed blood-derived *Ccr2*⁺ macrophages in DNMT3A R882H mice and *Ccr2* is mostly expressed in clusters coming from DNMT3A R882H mice (**Figure for the reviewer 17d-g**). Increased number of *Ccr2*⁺ macrophages can explain the elevated levels of *Hbegf* in the heart. Finally, we demonstrate an increase of prototypical fibrosis markers in the cluster of fibroblasts in the DNMT3A R882H but not WT mice (**Figure for the reviewer 17h-i**).

We hope that we convinced the reviewer that our data sufficiently support that EGF derived from DNMT3A monocytes can regulate cardiac fibroblasts. We do acknowledge the lack of in vivo studies to confirm a potential therapeutic benefit of EGF inhibition in the limitation paragraph of the revised manuscript.

The new data are provided in the revised manuscript (page 8) and are described as follows.

“To determine the effect of the DNMT3A^{R882H} mutation on the cardiac tissue on a cellular level, we performed single nuclei RNA sequencing of cardiac tissue. After annotating all major cardiac cell types (Fig. 4g, Supplementary Figure 5a-d), we analyzed the expression of fibrosis-related genes in cardiac fibroblasts (Fig. 4h). Expression of pro-fibrotic and myofibroblast activation genes *Col3a1*, *Postn*, *Pdgfra* and others were upregulated in fibroblasts of DNMT3A^{R882H} mice (Fig. 4h, i, Supplementary Figure 5e). Additionally, we did observe an elevated expression of *Ccr2* and increased numbers of *Ccr2* positive cells in the macrophages subcluster specifically in DNMT3A^{R882H} mice (Fig. 4j, Supplementary Figure 5f-h), suggesting an enrichment of bone marrow-derived macrophages.”

On page 10 as follows:

“Furthermore, the percentage of *Hbegf* positive cells in the macrophage cluster was increased by 7,87-fold (p=0.0859) in DNMT3A^{R882H} mice, whereas *Areg* expression was similar between the groups (Supplementary Fig. 7d-g).”

And on page 12 as follows:

“Finally, to determine a potential causal involvement of EGF signaling in monocyte-fibroblast crosstalk, we inhibited EGF-receptor signaling in cardiac fibroblasts by the small-molecule EGFR kinase inhibitor gefitinib⁴⁴ or neutralizing antibodies directed against HB-EGF (Figure 8d). While gefitinib did not affect the basal expression of α SMA, it prevented the induction of α SMA by supernatants of DNMT3A-silenced monocytes (Fig. 8e, f), without showing cytotoxic effects in human cardiac fibroblasts (Fig. 8f). Moreover, EGFR inhibition with gefitinib in cardiospheres inhibited the induction of α SMA and abrogated reduced contraction of cardiac tissue mimetics induced by the treatment with the supernatant of DNMT3A-silenced monocytes (Fig. 8g, h). Finally, blocking HB-EGF by neutralizing antibodies reduced paracrine activation of fibroblasts by DNMT3A-silenced monocytes (Fig. 8i, j). Taken together, these results demonstrate that the secretome of DNMT3A silenced monocytes induces fibroblasts activation partially through EGFR signaling.”

Figure for the reviewer 17. EGF signaling in human cardiac fibroblasts and in mice with DNMT3A^{R882H} bone marrow cells after AMI. **a** Immunofluorescence analysis of α SMA protein expression in HCF stimulated with supernatants from DNMT3A-silenced THP1-cells with or without HB-EGF neutralizing antibody (50 μ g/ml, 48 h) (n=4). **b** Representative uniform manifold approximation and projection (UMAP) plots of snRNA-seq sequenced WT and DNMT3A^{R882H} murine hearts after AMI (day 75) showing different cell clusters identified after annotation. CM, Cardiomyocytes; EC, Endothelial cells; FB Fibroblasts; PC, Pericytes **c** Percentage of cells expressing *Hbegf* and *Hbegf* mean expression in the immune cell cluster of hearts from WT and DNMT3A^{R882H} mice after AMI. **d** Mean expression of *Ccr2* in WT and DNMT3A^{R882H} murine hearts after AMI. **e** UMAP plot of subclustered cardiac macrophages from WT and DNMT3A^{R882H} murine hearts after AMI (day 75) showing different cell clusters. **f**

Feature plots depicting gene expression of *Ccr2* in subclustered cardiac macrophages from WT and DNMT3A^{R882H} murine hearts after AMI. **g** Violin plot showing gene expression of *Ccr2* in subclusters of cardiac macrophages from WT and DNMT3A^{R882H} murine hearts after AMI. **h**. Dot plot depicting expression of fibrosis-related genes in the cardiac fibroblast cluster in the sequenced WT and DNMT3A^{R882H} murine hearts after AMI. **i**. Mean expression of *Col3a1*, *Postn*, *Pdgfra* in WT and DNMT3A^{R882H} murine hearts after AMI. Data are shown as mean ± SEM. Normal distribution was tested using the Shapiro-Wilk test or the Kolmogorov-Smirnov test. Statistical comparison of two normally distributed groups was performed using unpaired, two-sided Student's t-test; for the data not following Gaussian distribution Wilcoxon-Mann-Whitney test was applied.

- Again, referred to Figure 3a-c: I was surprised to see that the authors did not compare the effect of the BMT with the Ly5.2 WT or D3A+/m at baseline, without cardiac stress. I think this is a very important point as in this model the upstream stressor is DNMT3A CHIP, and it would be important to understand how much fibroblast activation and fibrosis DNMT3A CHIP trigger in vivo even in the absence of an HF model. cells in the absence of injury.

ANSWER: We thank reviewer for another valid point. A sham experiment was not carried out parallel to the AMI experiments, since we have to reduce the number of mice to a minimum according to the ethical committee. However, to address the question of the reviewer, we analyzed untreated WT and DNMT3A R882H mice and assessed diffuse fibrosis in the septum in the absence of injury. Briefly, p:Ipc-treated unfractionated BM cells from MxCre+:DNMT3A R882H (+/m) and wild-type MxCre-:Dnmt3a +/+ (+/+) mice transplanted into young, 15 weeks old WT recipients. Under basal conditions without myocardial infarction, we found a 1.42-fold increase in fibrosis, which however, was not statistically significant (**Figure for the reviewer 18a,b**).

These data are included and discussed in the revised manuscript (page 7) as follows:

“Cardiac fibrosis was also increased by 1.42-fold in DNMT3A^{R882H} carrying mice at baseline, but this increase was not statistically significant (Supplementary Fig. 4a).”

- The authors only mention in Figure 3b that the scar size was not significantly affected, but when looking at interstitial fibrosis (Figure 3c) the difference is significant. I would at least attempt to speculate while this could be the case.

ANSWERS: *First of all, we did confirm the lack of effect on scar size by showing that troponin release was not different between the groups (Figure for reviewer 19a). Furthermore, ejection fraction was similar at day 0 and only deteriorated later in the DNMT3A-R882H group during the remodeling phase at day 28 (Figure for reviewer 19b). Consistent with a decline in cardiac function at later stages, fibrosis was significantly augmented in the remote zone of DNMT3A-R882H mice (Figure for reviewer 19c,d).*

Our primary hypothesis is that DNMT3A CHIP monocytes drive exaggerated infiltration of monocytes, particularly in the remote zone of the infarcted myocardium, which contributes to excessive fibrosis. Indeed, we have seen increased number of CD68+ macrophages in the remote zone of DNMT3A CHIP mice subjected to AMI (Figure for reviewer 19e).

We have included the additional data (page 7) and have discussed the potential mechanistic explanation in the revised manuscript as follows:

“Similar cardiac troponin T levels 24 hours post LAD-ligation indicate comparable infarct sizes in wild-type mice carrying a human DNMT3A^{R882H} mutation (DNMT3A^{R882H}) in the bone marrow-derived hematopoietic and the control group (WT) (Fig.4b). Histopathological analysis of cardiac tissue after four weeks showed an increase in cardiac interstitial fibrosis in the remote zone of myocardial tissue DNMT3A^{R882H} mice in comparison WT mice, whereas the infarct size was not changed (Fig. 4c-f). Cardiac fibrosis was also increased by 1.42-fold in DNMT3A^{R882H} carrying mice at baseline, but this increase was not statistically significant (Supplementary Fig. 4a). Mutant DNMT3A^{R882H} carrying mice additionally showed an increase in CD68 positive inflammatory cells in the remote zone after infarction, whereas no change was detected in non-injured mice at baseline (Supplementary Fig. 4b, c). These data suggest that mimicking DNMT3A^{R882H} in mice augments the number of inflammatory cells in the remote zone and induces diffuse interstitial cardiac fibrosis particularly in response to injury.”

And on page 12 of the discussion:

“Cardiac fibroblasts comprise a heterogeneous population of cells, particularly after myocardial infarction^{30,45,46}. The observed interaction does not appear to be specific to one of the subpopulations, as most fibroblast clusters in the human heart samples showed an interaction with the DNMT3A CHIP-carrying monocytes (Supplementary Fig. 10). The activation of cardiac fibroblasts may lead to the instigation and further progression of diffuse cardiac fibrosis in patients with heart failure carrying DNMT3A CHIP-driver mutations. Since the EGF receptor is expressed in fibroblasts of the healthy hearts as well, our bioinformatics analysis predicts that an interaction would also occur in uninjured hearts. However, healthy DNMT3A^{R882H} mice did only show a trend, but no significant increase in cardiac fibrosis. As infiltration of circulating monocytes is augmented by endothelial activation and tissue inflammation, the additional

infarction-induced myocardial damage may amplify the recruitment of bone marrow derived cells into the heart. We indeed observed an increase in *Ccr2* expression in the macrophage cluster, which is indicative of recruited bone marrow-derived macrophages²⁴, and further detected more CD68 positive immune cells in the remote zone of the infarcted DNMT3A^{R882H} mutant mice. Since the release of inflammatory mediators can activate cardiac fibroblasts, the recruited monocytes may secondarily drive fibrosis not only via HB-EGF, but also by releasing interleukins and other cytokines well-known to activate fibroblasts⁴⁷. Likely, DNMT3A CHIP mutations amplify heart failure by pleiotropic effects on inflammation and fibrosis. ”

Figure for the reviewer 19. DNMT3A CHIP affects diffuse cardiac fibrosis and immune cell infiltration in mice with DNMT3A^{R882H} bone marrow cells after AMI. a ELISA-based quantification of cardiac Troponin in serum of wild-type mice transplanted with wild-type bone marrow cells (WT) and wild-type mice transplanted with DNMT3A^{R882H} bone marrow cells (DNMT3A^{R882H}) after AMI (n=5). **b** Left ventricle ejection fraction determined with echocardiography of WT and DNMT3A^{R882H} on day 0 and day 28 after AMI. **c** Representative brightfield images of picosirius red staining of fibrotic scar and diffuse interstitial cardiac fibrosis in the septal zone of WT and DNMT3A^{R882H} mice. **d** Quantification of the fibrotic scar area and diffuse cardiac fibrosis (n=5). **e** Immunofluorescence analysis of CD68+ immune cell infiltration in the remote zone of WT and DNMT3A^{R882H} mice after AMI (n=3). Data are shown as mean ± SEM. Normal distribution was tested with the Shapiro-Wilk test or the Kolmogorov-Smirnov test. Statistical comparison of two normally distributed groups was performed using unpaired, two-sided Student's t-test.

- I strongly suggest the authors to revisit how the statistical significance is shown in the figures as it sometimes confusing and it's not clear which groups are compared (for example please see Figure 5h).

ANSWERS: We have improved the illustration of the statistical analysis.

- Figure 4a: I think it would make more sense to show the differences in signaling from the CHIP monocytes to specifically the fibroblasts, not all cardiac tissues. At this point of the manuscript, the authors have already established that their focus is understanding how CHIP monocytes can affect fibroblast and fibrosis, so showing the signaling pathway governing the connection between CHIP monocytes and all cardiac cell is not that relevant at this point.

ANSWERS: We thank the reviewer for this helpful suggestion and have improved the illustration of the data. As shown in the **Figure for the reviewer 20a-d** (and revised manuscript on page 8-9), we focused our attention on the monocyte-fibroblast interactions. Additionally, we provide the data about signals between monocytes and all cells in healthy cardiac tissue and in HFrEF tissue for the comparison (**Figure for the reviewer 20e-g**).

Figure for the reviewer 20. EGF signaling contributes to CHIP monocyte-mediated cardiac fibroblast activation. **a** CellChat ligand receptor prediction showing signalling pathways mediating relative information flow (left) and total information flow (right) of DNMT3A CHIP patient-derived monocytes with fibroblasts of healthy cardiac tissue **b**. CellChat ligand receptor prediction showing signalling pathways mediating relative information flow (left) and total information flow (right) of DNMT3A CHIP patient-derived monocytes with fibroblasts of HFrEF cardiac tissue. **c**. Venn diagram depicting common upregulated signalling pathways in monocytes of DNMT3A CHIP carriers with healthy and HFrEF fibroblasts. **d**. Bubble plots representing upregulated ligand-receptor pairs in DNMT3A CHIP monocytes-to-fibroblasts signalling in healthy (left) and HFrEF (right) myocardium. Color encodes communication probability, min. logFC for the interaction depicted is 0.1 and detection in minimum 10% of the cells. **e** CellChat ligand receptor prediction showing signalling pathways mediating relative information flow (left) and total information flow (right) of DNMT3A CHIP patient-derived monocytes with all cell types of healthy cardiac tissue **b**. CellChat ligand receptor prediction showing signalling pathways mediating relative information flow (left) and total information flow (right) of DNMT3A CHIP patient-derived monocytes with all cell types of HFrEF cardiac tissue. **c**. Venn diagram depicting common upregulated signalling pathways in monocytes of DNMT3A CHIP carriers with healthy and HFrEF cardiac tissue.

The data are described on pages 8-9 as follows:

“Next, we assessed the mechanism by which monocytes obtained from DNMT3A CHIP-driver mutation carriers stimulate fibroblast activation. Therefore, we predicted the receptor-ligand interactions between monocytes derived from DNMT3A CHIP-driver mutation carriers versus No-CHIP patients and cardiac tissue by using CellChat. A separate analysis of interactions between circulating monocytes and fibroblasts within the healthy heart revealed significant enrichment of natriuretic peptide receptor 2 (NPR2), epidermal growth factor (EGF), interleukin 1 (IL1), RESISTIN, insulin growth factor (IGF) and other signaling pathways in DNMT3A CHIP monocyte interactions (Fig. 6a). Enriched interactions within HFrEF heart tissue showed partially overlapping, but also distinct pathways including prolactin (PRL), EGF, activin (ACTIVIN), resistin (RESISTIN), and insulin-like growth factor (IGF) signaling between DNMT3A CHIP monocytes and cardiac fibroblasts (Fig. 6b). Common enriched pathways for DNMT3A CHIP condition in both healthy and diseased heart

tissue include EGF, RESISTIN and IGF, with EGF dominating absolute information flow (Fig. 6c). Since EGF was predicted to mediate interactions of CHIP monocytes with healthy and disease heart and showed the highest information flow, we focused our attention to this pathway. EGF signaling was defined by enriched ligand-receptor pairing of heparin-binding epidermal growth factor (EGF)-like growth factor (HB-EGF) and amphiregulin (AREG) expressed by monocytes of DNMT3A CHIP-carriers with the receptors EGFR, ERBB2 or ERBB4 in cardiac cell types in healthy and heart failure conditions (Fig. 6d). Reactome analysis of interactions specific for monocytes obtained from DNMT3A CHIP-driver mutation carriers to cardiac fibroblasts revealed an enrichment of EGFR interactions with known EGF downstream signaling pathways like phospholipase C-gamma, GRB2, SHC1 activity (Fig. 6e, f). Moreover, analysis of the specific interactions between monocytes and fibroblasts revealed a specific enrichment of EGF-family member to EGFR cross talks in DNMT3A CHIP carriers (Fig. 6g). “

- Along the same lines of my first comment: 4b should show the GO terms of the enriched interactions between CHIP monocytes/fibroblasts versus WT monocytes/fibroblasts.

ANSWER: We thank the reviewer for this suggestion and provided the requested analysis (see **Figure for the reviewer 21** below and Figure 6 of the revised manuscript).

The data are described on page 9 follows:

“Reactome analysis of interactions specific for monocytes obtained from DNMT3A CHIP-driver mutation carriers to cardiac fibroblasts revealed an enrichment of EGFR interactions with known EGF downstream signaling pathways like phospholipase C-gamma, GRB2, SHC1 activity (Fig. 6e, f).”

Figure for the reviewer 21. Reactome 2022 processes upregulated in healthy and HFrEF tissue with CHIP monocytes. a Gene ontology analysis of differentially expressed genes (logFC for the interaction is 0.1 and detection in minimum 10% of the cells, p-value < 0.05) using the Enrichr data base. Representation of the ten most significant functional categories in healthy (**a**) and HFrEF (**b**) interactions represented by Enrichr combined score that considers P value and Z score

Response to Reviewer #4:

Shumliakivska et al. investigated the interaction between fibroblasts and CHIP-mutated monocytes in heart failure (HF) disease progression and fibrotic tissue formation. While the authors used wide range of functional assays to prove their hypothesis, there are crucial issues in the manuscript that need to be edited in experimental design, analysis of scRNA-seq data and manuscript writing.

Major comments

1. The introduction section is very short, and major issues are missing, for instance:
 - What is known about monocytes, macrophages and stromal cells in HF models, and specifically in CHIP?
 - Introduce the main methods/cellular functions which are discussed in the manuscript, such as the different scRNA-seq-based methods to infer cell-cell communications.
 - What are the main results and observations of the study?

ANSWER: *We thank the reviewer for this constructive critique and have extended and improved the introduction section (see page 3-4) of the revised manuscript.*

2. In the scRNA-seq data analysis, the authors claim they integrated data of circulating monocytes derived from patients with chronic heart failure with publicly available single-nuclei RNA-seq data of human heart tissue. The authors need to specify and explain more about the data-integration procedure (single-cell with single-nuclei). Which method for integration did they use? They should present QC figures in the supplementary data.

ANSWER: *The original data set was created by directly merging monocytes from scRNA sequencing dataset and cardiac cell types from snRNA-sequencing dataset. In order to improve the data integration process, we used an alternative approach and performed additional quality control analysis:*

Using Seurat (version 4.0.2), we subtracted the monocyte clusters via the subset() function of the published PBMC dataset from CHIP and NO-CHIP carriers². Quality control filters were applied as in the original publications^{2,7}. To integrate snRNA-seq data of cardiac tissue and scRNA-seq datasets of PBMCs, we followed the standard Seurat integration vignette for different datasets (https://satijalab.org/seurat/articles/integration_introduction.html). In short, integration anchors were retrieved with the FindIntegrationsAnchors() function, followed by the IntegrateData() function to generate a combined object. Expression matrices were normalized and scaled with 'NormalizeData()', 'FindVariableFeatures()' and 'ScaleData()'. Using 'RunPCA()', we obtained the reduced dimensionality of the merged object. For the more details please refer to the updated Methods sections.

*Quality controls demonstrate that the number of expressed features, the UMI counts and the percent mitochondria content is similar across the circulating monocytes and the tissue-derived cells (**Figure for reviewer 22**). Of note, all indices are higher in the circulating cells, because these data are derived from single cell RNA sequencing as compared to nuclei RNA sequencing, which*

was used for the sequencing of the tissue to allow sequencing of cardiomyocytes. Since we compare No-CHIP with CHIP carriers, which show very consistent quality, the general difference between the sources are not expected to influence the results. However, this aspect is discussed in the limitation section of the revised manuscript (page 13-14) as shown below. The codes will be uploaded upon acceptance at Github.

“Limitations: Although our study provides novel insights into the impact of DNMT3A CHIP on cardiac pathologies in mice and humans, the causal contributions of the HB-EGF-EGF receptor interactions to cardiac fibrosis has only been documented by using in vitro models and cardiac tissue mimetics. Further studies have to address whether inhibition of the EGF pathway reduces DNMT3A CHIP induced cardiac fibrosis in vivo.

Second, our in silico prediction analysis is limited to the investigation of circulating monocytes with human cardiac tissue. This interaction would not directly occur, since monocytes only can interact with the cardiac fibroblast and cardiomyocytes after entering the heart. The analysis of cardiac tissue biopsies from DNMT3A CHIP mutant carriers would provide a better model. However, such an analysis is limited by the low number of immune cells in biopsies and difficulties in discriminating between the mutant and wild type cells. A second limitation of our in silico data is the combination of single cell and single nuclei RNA sequencing data. However, since the No-CHIP and DNMT3A CHIP data show similar quality, a comparative analysis of the interactions of the two groups with the cardiac tissue-derived cells appears valid. We also confirmed the predicted findings with different cell culture models, tissue mimetics and mutant mice in vivo.”

3. The scRNA-seq data should be thoroughly analyzed in the gene expression level. In cell annotation – based on which gene programs the authors annotated the cell types/states. Are there monocytes and monocyte-derived macrophages in the myeloid cluster? The cell-cell interaction transcriptome analysis (CellChat) should be presented also in the gene, and not only the cellular, level. The authors should present the top X ligands and receptors, which represent the hub signaling between monocytes and fibroblasts. Moreover, to show in supplementary the L-R genes related to other pairs.

ANSWER to first part of the question: Regarding the first question, cell annotation of the integrated snRNA-Seq and scRNA-Seq datasets was performed based on the expression of characteristic marker genes as previously described^{2,8}. Clusters were annotated by the FindAllMarkers function (**Figure for reviewer 23a**) and by detection of established and published cell type defining markers (**Figure for reviewer 23b**). Briefly, cardiomyocytes expressed genes encoding troponin (TNNT2), ryanodine receptor type 2 (RYR2), endothelial cells expressed genes encoding vascular endothelial (VE)-cadherin (CDH5) and CD31 (PECAM1), fibroblasts were characterized by decorin (DCN) expression, pericytes expressed platelet derived growth factor receptor beta (PDGFRB), and smooth muscle cells were identified by expression of myosin heavy chain 11 (MYH11). Immune cells of the cardiac tissue (named “immune cells”) were characterized by CD163, which is primarily expressed by tissue resident macrophages, whereas the cluster of circulating monocytes (named “monocytes”) showed higher expression of the monocyte markers CD14 and PTPRC, as well as CCR2, which mediates homing to the cardiac tissue. One cluster contained neuronal markers (such as neurexins, NRXN) so they were annotated as ‘neuronal-like’ cells (Neuro) (**Figure for reviewer 23b**). We included the description of the cell annotation strategy in the updated methods section and provide the annotation profile as Figure 1 and Supplementary Figure 2.

ANSWER to the second part of the question: As for cell-cell interaction transcriptome analysis representation, we additionally included a bubble plot depicting significantly regulated and unique ligand receptor pairs in monocytes-to-fibroblasts crosstalk (**Figure for reviewer 24** and Figure 6 of the revised manuscript). Additionally, we show the expression of the ligand and receptors pairs as violin plots (**Figure for reviewer 24b**). These raw data demonstrate that the expression of HBEGF (in CHIP monocytes) and EGFR (on fibroblasts) is most promising and, therefore, was further investigated.

Figure for the reviewer 24. HBEGF-EGFR signaling axis contributes to CHIP monocyte-mediated cardiac fibroblast activation. **a** Bubble plots representing upregulated ligand-receptor pairs in DNMT3A CHIP monocytes-to-fibroblasts signalling in healthy (left) and HFREF (right) myocardium. Color encodes communication probability, min. logFC for the interaction depicted is 0.1 and detection in minimum 10% of the cells. **b** Violin plot depicting expression of the top 6 ligand and receptors pairs driving CHIP monocytes-to-fibroblasts interactions in healthy heart tissue.

4. Figure 3 – the examinations and tests that were done in the human tissues are not clear. What exactly was measured? How are these measurements correlated directly to the monocytes/macrophages and to their crosstalk with the fibroblasts?

ANSWER: *We have improved the description of the experiments as follows (page 8):*

“To further investigate the impact of harboring DNMT3A CHIP-driver mutations on diffuse cardiac fibrosis in humans, we determined cardiac fibrosis using cardiac magnetic resonance imaging with myocardial mapping^{36–38} in patients with heart failure (Fig. 5a, Supplementary Table 2). 6 of the patients were harboring DNMT3A CHIP-driver mutations (Fig. 5b-d, Supplementary Fig. 6a), but did not differ with respect to sex, age or co-morbidities from No-CHIP patients (Supplementary Table 3). We performed native T1 mapping, which specifically determines excessive extracellular matrix deposition associated with diffuse fibrosis³⁹. In addition, we measured T2, which primarily detects myocardial edema^{36,38}. Carriers of DNMT3A CHIP-driver mutations had significantly increased native T1, but no differences in native T2 measurements, indicating the presence of diffuse myocardial fibrosis in these patients (Fig. 5 e-g).”

5. In Figure 4 the authors present a crosstalk mechanism between the DNMT3A-silenced monocytes and the fibroblasts via HB-EGF-EGFR and AREG-EGFR.

a. The authors need to show that the expression of the ligands is highly expressed by the monocytes, but not the other cell types that they analyzed.

b. Moreover, is this signaling was also activated in the chimeric mice that the authors performed in Figure 3a?

ANSWER: *We demonstrate that the ligands are highly expressed in monocytes of the human data sets (Figure for the reviewer 25a). Of the cardiac cells, only cardiomyocytes showed some expression of the EGF ligands. Moreover, we performed snRNA seq of the mouse hearts and showed that Hbegf is predominantly expressed in the DNMT3A R882H mutant mice (Figure for the reviewer 25b-e). Hbegf expression was 3,5-fold higher in DNMT3A R882H mice compared to control mice ($p=0.21$). The expression of Ar(Areg) was not augmented (Figure for the reviewer 25f-h).*

Finally, we confirmed that HBEGF specifically mediates the paracrine interactions of DNMT3A silenced monocytes with fibroblasts, as demonstrated by the inhibition with neutralizing antibodies (Figure for the reviewer 25i).

These data are included in the revised manuscript in pages 9 and 10.

The expression of both, HBEGF and AREG, was significantly higher in circulating monocytes of heart failure patients carrying DNMT3A CHIP-driver mutations compared to No-CHIP carriers (Fig. 6h, Supplementary Fig. 7a). Of the EGF receptor family members, the EGF receptor was highest expressed in cardiac fibroblasts, both in the healthy and HFrEF hearts (Fig. 6i, Supplementary Fig. 7b-c). Moreover, HB-EGF protein was increased in plasma samples obtained from patients with HF

harboring DNMT3A CHIP-driver mutations compared to No-CHIP HF patients as assessed by ELISA (Fig. 6j; Supplementary Table 3). Furthermore, the percentage of *Hbegf* positive cells in the macrophage cluster was increased by 7,87-fold ($p=0.0859$) in DNMT3A^{R882H} mice, whereas *Areg* expression was similar between the groups (Supplementary Fig. 7d-g).

Figure for the reviewer 25. HB-EGF-EGFR and AREG-EGFR signalling in mice with DNMT3A^{R882H} bone marrow cells and in human cardiac fibroblasts. a Violin plots demonstrating *HBEGF* and *AREG* expression in different cell types of the integrated scRNA-seq and scnRNA-seq object. **b** Representative uniform manifold approximation and projection (UMAP) plots of snRNA-seq sequenced WT and DNMT3A^{R882H} murine hearts after AMI (day 75) showing different cell clusters identified after annotation. CM, Cardiomyocytes; EC, Endothelial cells; FB Fibroblasts; PC, Pericytes **c**. Violin plot representing *Hbegf* gene expression in the immune cell cluster of hearts derived from WT and DNMT3A^{R882H} mice after AMI. **d, e**. Percentage of cells expressing *Hbegf* (**d**) and *Hbegf* mean expression (**e**) in the immune cell cluster of hearts from WT and DNMT3A^{R882H} mice after AMI. **f**. Violin plot representing *Ar* (Areg)

gene expression in the immune cell cluster from WT and DNMT3A^{R882H} mice after AMI **g,h**. Percentage of cells expressing *Ar* (**g**) and *Ar* mean expression (**h**) in immune cell clusters in hearts of WT and DNMT3A^{R882H} mice after AMI.

6. The discussion section is superficial, and missing many known literature in the fields of: Fibroblasts, EGF signaling, immune cell function in heart failure and in fibrotic tissues, scRNA-seq and intercellular communication of myeloid cells and fibroblasts in heart tissue and other organs in tissue homeostasis and pathology, etc.

ANSWER: *We have addressed this comment and have improved the discussion section of the manuscript (see new discussion section on page 12-14)*

Minor comments:

1. The authors stated they used CellChat, however, in Figure 1a they wrote CellChatDB. While there is a pipeline named CellPhoneDB, this is not clear which computational pipeline was used by the authors in their analysis.

ANSWER: *We thank the reviewer for spotting this mistake. We have corrected the text accordingly.*

2. Experimental procedures of chimeric mice preparation should be explained in more details in the Results section. Moreover, what was the percentage of the chimeras? What is the portion of the Ly5.2 monocytes/macrophages out of Ly5.1 cells in the heart following LAD ligation?

ANSWER: *We have updated the description of the experimental procedure and included the percentage of engraftment in the **Figure for reviewer 26a** below. We do not know how many of the engrafted cells home to the heart, but our snRNA sequencing analysis showed increased levels of CCR2 positive macrophages (which were shown to represent bone marrow-derived cells in the heart⁹) suggesting that DNMT3A-R882H mutant cells show an increased homing and integration as compared to controls (**Figure for reviewer 25b**).*

Method section:

Age-matched *Mx-Cre⁺/Dnmt3a* and *Mx-Cre⁻/Dnmt3a* donor mice (Ly5.2) were treated with pl:pC (400µg/mouse intraperitoneally [i.p.]) for 3 non-consecutive days. Donor bone marrow cells (2x10⁶ Ly5.2 cells) were intravenously injected into lethally irradiated (7.5Gy total body irradiation) B6.SJL recipients (Ly5.1). Recipient mice were maintained on antibiotic-containing drinking water (ciprofloxacin 50 mg/kg) 5 days pre-lethally irradiation and 2 weeks post irradiation and transplantation. Peripheral blood engraftment was assessed by flow cytometry 6 weeks after reconstitution. Levels of donor chimerism were differentiated by staining with anti-mouse CD45.1 BUV737 and anti-mouse CD45.2 BV711 antibodies. Dead cells were excluded by 7-AAD staining. The percentage of donor engraftment was above 90% Ly5.2-positive cells.

References

1. Shahid, F., Lip, G. Y. H. & Shantsila, E. Role of monocytes in heart failure and atrial fibrillation. *J. Am. Heart Assoc.* **7**, 1–17 (2018).
2. Abplanalp, W. T. *et al.* Clonal Hematopoiesis-Driver DNMT3A Mutations Alter Immune Cells in Heart Failure. *Circ. Res.* **128**, 216–228 (2021).
3. Fischer, C. *et al.* Long-term functional and structural preservation of precision-cut human myocardium under continuous electromechanical stimulation in vitro. *Nat. Commun.* **10**, 1–12 (2019).
4. Sano, S. *et al.* JAK2V617F-Mediated Clonal Hematopoiesis Accelerates Pathological Remodeling in Murine Heart Failure. *JACC Basic to Transl. Sci.* **4**, 684–697 (2019).
5. Kühn, R., Schwenk, F., Aguet, M. & Rajewsky, K. Inducible gene targeting in mice. *Science (80-)*. **269**, 1427–1429 (1995).
6. Scheller, M. *et al.* Hotspot DNMT3A mutations in clonal hematopoiesis and acute myeloid leukemia sensitize cells to azacytidine via viral mimicry response. *Nat. Cancer* **2**,

- 527–544 (2021).
7. Litviňuková, M. *et al.* Cells of the adult human heart. *Nature* **588**, 466–472 (2020).
 8. Nicin, L. *et al.* A human cell atlas of the pressure-induced hypertrophic heart. *Nat. Cardiovasc. Res.* **1**, 174–185 (2022).
 9. Lavine, K. J. *et al.* Distinct macrophage lineages contribute to disparate patterns of cardiac recovery and remodeling in the neonatal and adult heart. *Proc. Natl. Acad. Sci. U. S. A.* **111**, 16029–16034 (2014).

REVIEWERS' COMMENTS

Reviewer #1 (Remarks to the Author):

This study is a strong and impactful investigation into causal connections between DNMT3A CHIP and cardiac interstitial fibrosis through direct monocyte to fibroblast signaling that promotes myofibroblast activation, which is characteristic of HFrEF. In response to the first comments, the authors have additionally applied their clinical findings to the context of HFrEF patients, further characterized cell-cell communication in myocardial tissue by analyzing their CellChat analyses for resident cardiac immune cells to fibroblasts and circulating monocytes to cardiomyocytes, and rationalized their focus on EGF signaling rather than TGF signaling. Additionally, the authors directly tested and validated their correlative findings in a new causal mouse experiment presented in the revised manuscript, wherein the authors used adoptive transfer to investigate DNMT3A-mutant CHIP on cardiac remodeling after myocardial infarction with scRNAseq to validate their findings in clinical datasets and cell culture assays. The only limitation to this mouse experiment is that it models ischemia-induced heart failure, and the clinical findings are generalized to any patient with HFrEF, but this is a reasonable model for HFrEF and strongly supports the causal relationship presented in the conclusions of this manuscript.

Reviewer #2 (Remarks to the Author):

The authors have addressed my concerns. I have no further recommendations.

Reviewer #3 (Remarks to the Author):

I thank the authors for having addressed all the points I have raised in my previous comments.

Reviewer #4 (Remarks to the Author):

The authors referred to all the comments I suggested and improved the manuscript.

I suggest the authors will edit the text, since there are sentences which are not so clear along the manuscript.

Response to the reviewers

We thank the reviewers and the editor for their positive comments that have helped us improving our manuscript.

Reviewer #1 (Remarks to the Author):

This study is a strong and impactful investigation into causal connections between DNMT3A CHIP and cardiac interstitial fibrosis through direct monocyte to fibroblast signaling that promotes myofibroblast activation, which is characteristic of HFrEF. In response to the first comments, the authors have additionally applied their clinical findings to the context of HFrEF patients, further characterized cell-cell communication in myocardial tissue by analyzing their CellChat analyses for resident cardiac immune cells to fibroblasts and circulating monocytes to cardiomyocytes, and rationalized their focus on EGF signaling rather than TGF signaling. Additionally, the authors directly tested and validated their correlative findings in a new causal mouse experiment presented in the revised manuscript, wherein the authors used adoptive transfer to investigate DNMT3A-mutant CHIP on cardiac remodeling after myocardial infarction with scRNAseq to validate their findings in clinical datasets and cell culture assays. The only limitation to this mouse experiment is that it models ischemia-induced heart failure, and the clinical findings are generalized to any patient with HFrEF, but this is a reasonable model for HFrEF and strongly supports the causal relationship presented in the conclusions of this manuscript.

We have further developed the limitation section of the manuscript in order to address the concern of the reviewer. The new paragraph reads as follows:

Another limitation of our study is that the murine heart failure analysis was conducted in a model of ischemia-induced heart failure, while the clinical data originate from HFrEF patients, potentially with different underlying etiologies. Nevertheless, this mouse model is well-established for HFrEF and supports the casual relationship between DNMT3A CHIP-driver mutation and diffuse cardiac fibrosis.

Reviewer #2 (Remarks to the Author):

The authors have addressed my concerns. I have no further recommendations.

Thank you very much.

Reviewer #3 (Remarks to the Author):

I thank the authors for having addressed all the points I have raised in my previous comments.

Thank you very much.

Reviewer #4 (Remarks to the Author):

The authors referred to all the comments I suggested and improved the manuscript. I suggest the authors will edit the text, since there are sentences which are not so clear along the manuscript.

As suggested by the reviewer, we have edited the text to improve the clarity of the manuscript.